# The Role of Baselines in Policy Gradient Optimization

**Jincheng Mei** [1*]       **Wesley Chung** [2]       **Valentin Thomas** [3]       **Bo Dai** [1]
**Csaba Szepesvári** [4, 5, *]       **Dale Schuurmans** [1, 5]

[1]Google Research, Brain Team    [2]Mila, McGill University    [3]Mila, University of Montreal
[4]DeepMind    [5]Amii, University of Alberta

{jcmei,bodai,szepi,schuurmans}@google.com   {wesley.chung2,vltn.thomas}@gmail.com

## Abstract

We study the effect of baselines in on-policy stochastic policy gradient optimization, and close the gap between the theory and practice of policy optimization methods. Our first contribution is to show that the *state value* baseline allows on-policy stochastic *natural* policy gradient (NPG) to converge to a globally optimal policy at an $O(1/t)$ rate, which was not previously known. The analysis relies on two novel findings: the expected progress of the NPG update satisfies a stochastic version of the non-uniform Łojasiewicz (NŁ) inequality, and with probability 1 the state value baseline prevents the optimal action's probability from vanishing, thus ensuring sufficient exploration. Importantly, these results provide a new understanding of the role of baselines in stochastic policy gradient: by showing that the variance of natural policy gradient estimates remains unbounded with or without a baseline, we find that variance reduction *cannot* explain their utility in this setting. Instead, the analysis reveals that the primary effect of the value baseline is to **reduce the aggressiveness of the updates** rather than their variance. That is, we demonstrate that a finite variance is *not necessary* for almost sure convergence of stochastic NPG, while controlling update aggressiveness is both necessary and sufficient. Additional experimental results verify these theoretical findings.

## 1   Introduction

The policy gradient (PG) [29] is a key concept in reinforcement learning (RL), lying at the foundation of policy-based and actor-critic methods, and responsible for some of the most prominent practical achievements in RL [27, 28, 11]. However, progress in the theoretical understanding of PG methods is recent, and a number of the techniques used in practice still lack rigorous support, particularly in the online stochastic regime where an action is sampled from the current policy at each iteration. We study stochastic policy optimization in more detail to close this gap between theory and practice.

In stochastic policy optimization, the two most common techniques for improving the basic algorithm are to include on-policy importance sampling (IS) and subtract a baseline. Including on-policy IS provides unbiased gradient estimates, but introduces high variance when an action's sampling probability is close to $0$. Meanwhile, subtracting a baseline remains a heuristic [26] that has strong empirical but limited theoretical support. One possible benefit of a baseline is that it provides variance reduction [10], which has motivated work on designing alternative baselines that further reduce variance [30, 4, 20, 31]. However, other work [7] has shown that variance reduction is not necessarily aligned with policy learning quality. To date, it has remained unclear how a baseline impacts the quality of the ultimate solution found by policy gradient optimization. We resolve this question in this work.

---

[*]Correspondence to: Jincheng Mei and Csaba Szepesvári

36th Conference on Neural Information Processing Systems (NeurIPS 2022).

Recent progress in the theory of deterministic PG has shown that, given exact gradients, softmax policy gradient is able to converge to a globally optimal policy at a $O(1/t)$ rate [24]. Unfortunately, despite this guarantee, the constants in this rate can be extremely large [19] due to initialization sensitivity and poor performance at escaping sub-optimal plateaus [23]. Therefore, in the exact gradient setting, several techniques have been considered for mitigating the weaknesses of softmax PG, leading to better constants [2] or even exponentially faster rates of $O(e^{-c \cdot t})$ for $c > 0$. Such improvements include adding entropy regularization [24, 6], normalizing the gradients [22], or applying natural policy gradient (NPG) [6, 14, 21].

However, in the on-policy *stochastic* optimization case, recent studies [21, 7] show that naively applying the above techniques, such as normalization or NPG, leads to unexpectedly *worse* performance than stochastic PG. That is, techniques that accelerate convergence in the exact policy gradient setting become *unsound* in the stochastic gradient setting, by inducing a non-zero probability of failure (i.e., failing to converge to a globally optimal solution) [21]. Such failures occur even when stochastic PG can still converge to a global optimum in probability. Previous work has indicated that one key reason behind the failure of these acceleration strategies arises from their "over-committal behaviour" in the stochastic setting, which occurs independently of the variance of the gradient estimates [7]. That is, baseline techniques with higher variance can still better avoid over-committal behaviour (i.e., premature convergence) and ultimately achieve better policy optimization [7].

To resolve this issue, we develop a deeper understanding of the role of baselines in stochastic policy optimization based on the following contributions. First, we establish a new result that combining on-policy IS with a value function baseline and natural policy gradient (NPG) can achieve almost sure convergence to a globally optimal policy at a $O(1/t)$ rate. This result is based on two novel findings: **(i)** At any iteration $t$, the conditional expected progress of the algorithm's next iterate obeys a stochastic non-uniform Łojasiewicz (NŁ) inequality. **(ii)** The use of the state value baseline (with appropriate learning rate control) almost surely prevents the probability of the optimal action from vanishing. These findings show that a key role of the value baseline is to automatically ensure "sufficient exploration" during on-policy stochastic optimization. Next, we provide a detailed understanding of how baselines modulate the circular interaction between stochastic action sampling and updating. Although a baseline has no effect on exact gradients, it can play a major role in stochastic gradients. In this respect, we first show that the PG estimator variance is unbounded with or without a baseline, hence variance reduction cannot be the primary effect. Instead, our analysis reveals that the key role the baseline plays in ensuring global convergence is to reduce the aggressiveness of updates. That is, finite variance of the gradient estimates is not necessary for ensuring global convergence, while properly controlling update aggressiveness is both necessary and sufficient.

The remainder of the paper is organized as follows. Section 2 provides the main results that establish the almost sure $O(1/t)$ convergence rate of stochastic NPG with on-policy IS and state value baseline to a globally optimal policy. Section 3 then develops the new understanding of the role of the baseline by going beyond standard variance reduction arguments. Section 4 provides some simulations to verify the results, and Section 5 concludes the paper with a brief discussion.

## 2 On-policy Stochastic Natural Policy Gradient

We first consider a one-state Markov Decision Process (MDP) defined by a finite action space $[K] \coloneqq \{1, 2, \ldots, K\}$ where the true mean reward vector is $r \in [0, 1]^K$. The policy optimization problem is to maximize the expected reward,

$$\max_{\theta: [K] \to \mathbb{R}} \mathbb{E}_{a \sim \pi_\theta(\cdot)} [r(a)], \tag{1}$$

where the policy $\pi_\theta$ is parameterized by $\theta$ using the standard softmax parameterization,

$$\pi_\theta(a) = \frac{\exp\{\theta(a)\}}{\sum_{a' \in [K]} \exp\{\theta(a')\}}, \quad \text{for all } a \in [K]. \tag{2}$$

Our focus in this paper is on on-policy optimization, where at each iteration $t \geq 1$ the current policy $\pi_{\theta_t}$ is used to sample one action and perform one update.

For the sampled action $a_t$, a noisy reward observation $x_t(a_t) \in \mathbb{R}$ is drawn from an unknown distribution with expected value $r(a_t)$. We make the following assumption that the observed reward $x_t(a)$ is sampled from a bounded distribution: $x_t(a) \in [-R_{\max}, R_{\max}]$ with probability one.

**Assumption 1** (Bounded sampled reward). *For each action $a \in [K]$, the true mean reward $r(a)$ is the expectation of a bounded reward distribution, i.e.,*

$$r(a) = \int_{-R_{\max}}^{R_{\max}} x \cdot P_a(x)\mu(dx) \tag{3}$$

*where $\mu$ is a finite measure over $[-R_{\max}, R_{\max}]$, and $P_a(x) \geq 0$ is the probability density function with respect to $\mu$, and $R_{\max} > 0$ is the reward range. We let $R_a$ denote the reward distribution for action $a$ defined by the density $P_a$ and base measure $\mu$.*

Then, given a sampled reward observation $x_t(a) \sim R_a$, an unbiased estimate of the expected reward vector $r$ can be formed by on-policy importance sampling (IS).

**Definition 1** (On-policy importance sampling (IS)). *At iteration $t$, sample one action $a_t \sim \pi_{\theta_t}(\cdot)$ and observe one reward sample $x_t(a_t) \sim R_{a_t}$. Let $x_t(a) = 0$ for all $a \neq a_t$. Then the IS reward estimate is constructed as $\hat{r}_t(a) = \frac{\mathbb{I}\{a=a_t\}}{\pi_{\theta_t}(a)} \cdot x_t(a)$ for all $a \in [K]$.*

If the true mean reward $r(a_t)$ is observed for sampled actions $a_t$, we have the simplified IS estimator.

**Definition 2** (Simplified on-policy importance sampling (IS)). *At iteration $t$, sample one action $a_t \sim \pi_{\theta_t}(\cdot)$. The IS reward estimate is then constructed as $\hat{r}_t(a) = \frac{\mathbb{I}\{a_t=a\}}{\pi_{\theta_t}(a)} \cdot r(a)$ for all $a \in [K]$.*

Definition 2 will be used for illustrating ideas and new understandings in Section 3, while the main results in Section 2 are based on Definition 1.

## 2.1 Failure Without a Baseline

First, to establish context, we review an existing negative result for the representative algorithm, natural policy gradient (NPG) [13], which for the softmax parameterization is defined as follows.

**Update 1** (NPG with on-policy stochastic gradient). *$\theta_{t+1} \leftarrow \theta_t + \eta \cdot \hat{r}_t$, where $\pi_\theta(a)$ is by Eq. (2).*

It is known that NPG behaves problematically with on-policy IS, even if the true mean reward $r(a_t)$ is observed. In particular, NPG converges to a sub-optimal deterministic policy with a constant positive probability in this case, as shown by [7, 21].

**Proposition 1** (Theorem 3 of [21]). *Using Update 1, where $\hat{r}_t$ is from Definition 2, and $r \in (0, 1]^K$, we have, with positive probability, $\sum_{a \neq a^*} \pi_{\theta_t}(a) \to 1$ as $t \to \infty$.*

Essentially Proposition 1 asserts that Update 1 is too aggressive: if sub-optimal actions are sampled $t$ times successively, their probabilities will become exponentially close to 1; i.e., $1 - \sum_{a \neq a^*} \pi_{\theta_t}(a) \in O(e^{-c \cdot t})$. It follows that $\prod_{t=1}^{\infty} \sum_{a \neq a^*} \pi_{\theta_t}(a) > 0$; that is, the on-policy sampling process $a_t \sim \pi_{\theta_t}(\cdot)$ has a non-zero probability of sampling sub-optimal actions forever, which implies that there is a positive probability that $\pi_{\theta_t}$ fails to converge to an optimal deterministic policy.

## 2.2 Global Convergence with a Value Baseline

Despite the above failure, we now prove that subtracting a value baseline rectifies the problem for NPG. Consider the modified update that includes a baseline.

**Update 2** (NPG, on-policy stochastic gradient with value baseline). *$\theta_{t+1} \leftarrow \theta_t + \eta \cdot (\hat{r}_t - \hat{b}_t)$, where $\pi_\theta(a)$ is by Eq. (2), $\hat{b}_t(a) = \left( \frac{\mathbb{I}\{a_t=a\}}{\pi_{\theta_t}(a)} - 1 \right) \cdot b_t$ for all $a \in [K]$, and $b_t := \pi_{\theta_t}^\top r$.*

Since $\mathrm{softmax}(\theta) = \mathrm{softmax}(\theta + c \cdot \mathbf{1})$ for all $c \in \mathbb{R}$, Update 2 is equivalent to the following update if $\hat{r}_t$ is by Definition 1. Given the same $\pi_{\theta_t}$, Updates 2 and 3 produce the same next policy $\pi_{\theta_{t+1}}$.

**Update 3.** *$\theta_{t+1}(a) \leftarrow \theta_t(a) + \eta \cdot \frac{\mathbb{I}\{a_t=a\}}{\pi_{\theta_t}(a)} \cdot (x_t(a) - \pi_{\theta_t}^\top r)$, i.e., $\theta_{t+1}(a_t) \leftarrow \theta_t(a_t) + \eta \cdot \frac{x_t(a_t) - \pi_{\theta_t}^\top r}{\pi_{\theta_t}(a_t)}$, and $\theta_{t+1}(a) \leftarrow \theta_t(a)$ for all $a \neq a_t$.*

Unfortunately, the variance of this update is not uniformly bounded whenever $\pi_{\theta_t}(a)$ is close to 0 for at least one action $a \in [K]$ (Proposition 3), therefore standard stochastic gradient analysis

for bounded variance estimators [25, 33, 17, 32] cannot be applied. Instead, we develop two new techniques to establish global convergence results, both of which rely heavily on using baselines.

Lemma 1 provides the first key technique, which we refer to as the stochastic NŁ inequality.

**Lemma 1** (Stochastic non-uniform Łojasiewciz (NŁ)). *Suppose Assumption 1 holds. Let $r \in [0,1]^K$, $a^* \coloneqq \arg\max_{a \in [K]} r(a)$, and $\Delta \coloneqq r(a^*) - \max_{a \neq a^*} r(a)$. Using Update 2 with on-policy sampling $a_t \sim \pi_{\theta_t}(\cdot)$ and IS estimator $\hat{r}_t$,*

**(1)** *if $\hat{r}_t$ is from Definition 2, then with constant learning rate $\eta > 0$, we have, for all $t \geq 1$,*

$$\pi_{\theta_{t+1}}^\top r - \pi_{\theta_t}^\top r \geq 0, \qquad almost\ surely\ (a.s.), \qquad and \tag{4}$$

$$\mathbb{E}_t[\pi_{\theta_{t+1}}^\top r] - \pi_{\theta_t}^\top r \geq \frac{\eta}{1+\eta} \cdot \pi_{\theta_t}(a^*) \cdot \left(r(a^*) - \pi_{\theta_t}^\top r\right)^2, \tag{5}$$

*where $\mathbb{E}_t[\cdot]$ is on randomness from on-policy sampling $a_t \sim \pi_{\theta_t}(\cdot)$.*

**(2)** *if $\hat{r}_t$ is from Definition 1, then with learning rate,*

$$\eta = \frac{\pi_{\theta_t}(a_t) \cdot \left|r(a_t) - \pi_{\theta_t}^\top r\right|}{8 \cdot R_{\max}^2}, \tag{6}$$

*we have, for all $t \geq 1$,*

$$\mathbb{E}_t[\pi_{\theta_{t+1}}^\top r] - \pi_{\theta_t}^\top r \geq \frac{1}{16 \cdot R_{\max}^2} \cdot \sum_{i=1}^K \pi_{\theta_t}(i)^2 \cdot \left|r(i) - \pi_{\theta_t}^\top r\right|^3 \tag{7}$$

$$\geq \frac{1}{16 \cdot R_{\max}^2} \cdot \frac{\Delta}{K-1} \cdot \pi_{\theta_t}(a^*)^2 \cdot \left(r(a^*) - \pi_{\theta_t}^\top r\right)^2, \tag{8}$$

*where $\mathbb{E}_t[\cdot]$ is on randomness from on-policy sampling $a_t \sim \pi_{\theta_t}(\cdot)$ and reward sampling $x \sim R_{a_t}$.*

**Remark 1.** *We have $\eta \in O(1/t)$ in Eq. (6) after knowing the convergence rate later.*

We refer to $\pi_{\theta_t}(a^*)^2$ in Eq. (8) the **stochastic NŁ coefficient**. Lemma 1 is a stochastic generalization of the NŁ inequality, which has been widely used in proving global convergence of softmax PG variants [24, 23, 22, 21, 34]. It is stochastic since Eq. (7) contains an expectation. It is non-uniform because Eq. (8) depends on $\theta_t$, which cannot be uniformly lower bounded away from $0$ across the entire domain of $\theta \in \mathbb{R}^K$ (that is, one can always find $\theta$ such that $\pi_\theta(a^*)$ is arbitrarily close to $0$).

The key idea of Lemma 1 is as follows. If $\hat{r}_t$ is from Definition 2, then by algebra we have,

$$\mathbb{E}_t[\pi_{\theta_{t+1}}^\top r] - \pi_{\theta_t}^\top r = \sum_{i=1}^K \pi_{\theta_t}(i) \cdot \frac{\left[\exp\left\{\eta \cdot \frac{r(i) - \pi_{\theta_t}^\top r}{\pi_{\theta_t}(i)}\right\} - 1\right] \cdot \left(r(i) - \pi_{\theta_t}^\top r\right)}{\exp\left\{\eta \cdot \frac{r(i) - \pi_{\theta_t}^\top r}{\pi_{\theta_t}(i)}\right\} + \frac{1 - \pi_{\theta_t}(i)}{\pi_{\theta_t}(i)}}. \tag{9}$$

Since $(e^{c \cdot y} - 1) \cdot y \geq 0$ for all $y \in \mathbb{R}$ and $c > 0$, Eq. (9) is non-negative (letting $y \coloneqq r(i) - \pi_{\theta_t}^\top r$ and $c \coloneqq \eta/\pi_{\theta_t}(i)$). However, this is not true if $\hat{r}_t$ is from Definition 1, where we have,

$$\mathbb{E}_t[\pi_{\theta_{t+1}}^\top r] - \pi_{\theta_t}^\top r = \sum_{i=1}^K \pi_{\theta_t}(i) \cdot \int_{-R_{\max}}^{R_{\max}} \frac{\left[\exp\left\{\eta \cdot \frac{x - \pi_{\theta_t}^\top r}{\pi_{\theta_t}(i)}\right\} - 1\right] \cdot \left(r(i) - \pi_{\theta_t}^\top r\right)}{\exp\left\{\eta \cdot \frac{x - \pi_{\theta_t}^\top r}{\pi_{\theta_t}(i)}\right\} + \frac{1 - \pi_{\theta_t}(i)}{\pi_{\theta_t}(i)}} \cdot P_i(x)\mu(dx). \tag{10}$$

Note that $(e^{c \cdot y'} - 1) \cdot y < 0$ if $y' \cdot y < 0$ and $c > 0$ (letting $y' \coloneqq x - \pi_{\theta_t}^\top r$, $y \coloneqq r(i) - \pi_{\theta_t}^\top r$, and $c \coloneqq \eta/\pi_{\theta_t}(i)$). For a "good" action ($r(i) - \pi_{\theta_t}^\top r > 0$), if unfortunately its sampled reward is "bad" ($x - \pi_{\theta_t}^\top r < 0$), then the update will make negative progress. Similar things happen for a "bad" action ($r(i) - \pi_{\theta_t}^\top r < 0$) with "good" sampled reward ($x - \pi_{\theta_t}^\top r > 0$). It is then necessary to use $\eta$ like Eq. (6), to control the non-linear sigmoid-like functions in the progress by piecewise linear functions (Lemma 15) to get non-negative **expected** progresses. According to Eq. (8), we have

$$\mathbb{E}_t[\pi_{\theta_{t+1}}^\top r] - \pi_{\theta_t}^\top r \geq 0, \tag{11}$$

which implies that Update 2 achieves non-negative progress *in expectation*. Combining Lemma 1 with Doob's supermartingale convergence theorem then leads to the following result.

**Corollary 1.** *The sequence $\{\pi_{\theta_t}^\top r\}_{t \geq 1}$ converges with probability one.*

Corollary 1 asserts that, the random sequence $\pi_{\theta_t}^\top r$ produced by Update 2 asymptotically approaches some finite value (since $\pi_\theta^\top r \in [0, 1]$), ruling out the possibility of divergence (oscillating forever). However, this does not necessarily imply that $\pi_{\theta_t}^\top r \to r(a^*)$ as $t \to \infty$. A subtlety arises in bounding the stochastic NŁ coefficient in Eq. (7) away from 0, which requires a second key technique.

**Lemma 2** (Non-vanishing stochastic NŁ coefficient / "automatic exploration"). *Using Update 2 with conditions in Lemma 1 and $\hat{r}_t$ from Definition 1, for an arbitrary initialization $\theta_1 \in \mathbb{R}^K$, we have,*

$$c := \inf_{t \geq 1} \pi_{\theta_t}(a^*) > 0, \qquad \text{almost surely (a.s.).} \tag{12}$$

Lemmas 1 and 2 together guarantee that $\pi_{\theta_t}^\top r \to r(a^*)$ as $t \to \infty$. In fact, using the "variance-like" expected progress (Eq. (7)), Corollary 1 implies that $\pi_{\theta_t}$ approaches a "generalized one-hot policy" as $t \to \infty$. Lemma 2 then argues by contradiction that $\pi_{\theta_t}$ cannot approach a sub-optimal "generalized one-hot policy" as $t \to \infty$, which will imply that the optimal action's probability must approach 1 and achieve Eq. (12). Proof details in the appendix and intuitions in Section 3 reveal that Update 2 achieves a form of "automatic exploration" by using a baseline, i.e., maintaining $\pi_{\theta_t}(a)$ decay no faster than $O(1/t)$, such that every action will be sampled infinitely many times in a long run. Finally, combining Lemmas 1 and 2, we establish not only asymptotic convergence of NPG to a global optimum, but also a global convergence rate of $O(1/t)$ in terms of the sub-optimality gap.

**Theorem 1** (Almost sure global convergence rate). *Using Update 2 with on-policy sampling $a_t \sim \pi_{\theta_t}(\cdot)$, the IS estimator $\hat{r}_t$ in Definition 1, $\eta$ in Eq. (6), and any initialization $\theta_1 \in \mathbb{R}^K$, we have,*

$$\mathbb{E}[(\pi^* - \pi_{\theta_t})^\top r] \leq \frac{16 \cdot R_{\max}^2}{\Delta \cdot \mathbb{E}[c^2]} \cdot \frac{K-1}{t}, \qquad \text{and} \tag{13}$$

$$\limsup_{t \geq 1} \left\{ \frac{\Delta \cdot c^2}{16 \cdot R_{\max}^2} \cdot \frac{t}{K-1} \cdot (\pi^* - \pi_{\theta_t})^\top r \right\} < \infty, \qquad \text{a.s.,} \tag{14}$$

*where $\pi^* := \arg\max_{\pi \in \Delta(K)} \pi^\top r$ is the optimal policy, $R_{\max}$ is the sampled reward range from Assumption 1, $\Delta := r(a^*) - \max_{a \neq a^*} r(a)$ is the reward gap of $r$, and $c > 0$ is from Lemma 2.*

## 2.3 General MDPs

Next, we generalize these results to finite Markov decision processes (MDPs). Given a finite set $\mathcal{X}$, let $\Delta(\mathcal{X})$ denote the set of all probability distributions on $\mathcal{X}$. A finite MDP is defined as a tuple $\mathcal{M} := (\mathcal{S}, \mathcal{A}, r, \mathcal{P}, \gamma)$, where $\mathcal{S}$ and $\mathcal{A}$ are finite state and action spaces, respectively. $r : \mathcal{S} \times \mathcal{A} \to \mathbb{R}$ is the expected reward function, $\mathcal{P} : \mathcal{S} \times \mathcal{A} \to \Delta(\mathcal{S})$ is the probability transition function, and $\gamma \in [0, 1)$ is the discount factor. We also extend Assumption 1 to every $(s, a) \in \mathcal{S} \times \mathcal{A}$ and assume there is a reward distribution $R_{s,a}$ with expectation $r(s, a)$, uniformly bounded within $[-R_{\max}, R_{\max}]$. Given a policy $\pi : \mathcal{S} \to \Delta(\mathcal{A})$, at each time $t \geq 0$, an agent is given a state $s_t \in \mathcal{S}$, takes an action $a_t \sim \pi(\cdot|s_t)$, then receives a scalar reward observation $x(s_t, a_t) \sim R_{s_t, a_t}$ and a next-state $s_{t+1} \sim \mathcal{P}(\cdot|s_t, a_t)$. The value function of $\pi$ at state $s$ is defined as

$$V^\pi(s) := \mathbb{E}_{\substack{a_t \sim \pi(\cdot|s_t), \\ s_{t+1} \sim \mathcal{P}(\cdot|s_t, a_t)}} \left[ \sum_{t=0}^\infty \gamma^t r(s_t, a_t) \,\middle|\, s_0 = s \right]. \tag{15}$$

The policy optimization problem for a general MDP is to maximize the expected value of the policy,

$$\max_{\theta : \mathcal{S} \times \mathcal{A} \to \mathbb{R}} V^{\pi_\theta}(\rho) := \max_{\theta : \mathcal{S} \times \mathcal{A} \to \mathbb{R}} \mathbb{E}_{s \sim \rho(\cdot)} [V^{\pi_\theta}(s)], \tag{16}$$

where $\rho \in \Delta(\mathcal{S})$ is an initial state distribution, and $\pi_\theta(\cdot|s) = \text{softmax}(\theta(s, \cdot))$,

$$\pi_\theta(a|s) = \frac{\exp\{\theta(s, a)\}}{\sum_{a' \in \mathcal{A}} \exp\{\theta(s, a')\}}, \text{ for all } (s, a) \in \mathcal{S} \times \mathcal{A}. \tag{17}$$

Given a policy $\pi$, its state-action value is defined as $Q^\pi(s, a) := r(s, a) + \gamma \cdot \sum_{s'} \mathcal{P}(s'|s, a) \cdot V^\pi(s')$, and its advantage function is defined as $A^\pi(s, a) := Q^\pi(s, a) - V^\pi(s)$, for $(s, a) \in \mathcal{S} \times \mathcal{A}$. The state distribution of $\pi$ is defined as $d_{s_0}^\pi(s) := (1 - \gamma) \cdot \sum_{t=0}^\infty \gamma^t \cdot \Pr(s_t = s|s_0, \pi, \mathcal{P})$. We also denote $d_\rho^\pi(s) := \mathbb{E}_{s_0 \sim \rho(\cdot)} [d_{s_0}^\pi(s)]$. Given $\rho$, there exists an optimal policy $\pi^*$ such that $V^{\pi^*}(\rho) = \max_{\pi : \mathcal{S} \to \Delta(\mathcal{A})} V^\pi(\rho)$. We denote $V^*(\rho) := V^{\pi^*}(\rho)$ for conciseness.

For a general MDP, we assume the initial state distribution $\mu$ is "sufficiently exploratory" [2, 24, 18].

**Assumption 2** (Sufficient exploration). *The initial state distribution satisfies* $\min_s \mu(s) > 0$.

At iteration $t$, the NPG method uses the current state distribution to sample one state $s_t \sim d_\mu^{\pi_{\theta_t}}(\cdot)$, then uses on-policy sampling to sample one action $a_t \sim \pi_{\theta_t}(\cdot|s)$. For the sampled state action pair $(s_t, a_t) \in \mathcal{S} \times \mathcal{A}$, the state-action value $Q^{\pi_{\theta_t}}(s_t, a_t)$ is then used to perform update. The current state value function $V^{\pi_{\theta_t}}(s_t)$ is used as the baseline, as shown in Algorithm 1.

---

**Algorithm 1** NPG, on-policy stochastic natural gradient

**Input:** Learning rate $\eta > 0$.
**Output:** Policies $\pi_{\theta_t} = \text{softmax}(\theta_t)$.
Initialize parameter $\theta_1(s, a)$ for all $(s, a) \in \mathcal{S} \times \mathcal{A}$.
**while** $t \geq 1$ **do**
    Sample $s_t \sim d_\mu^{\pi_{\theta_t}}(\cdot)$, and $a_t \sim \pi_{\theta_t}(\cdot|s_t)$.
    $\theta_{t+1}(s_t, a_t) \leftarrow \theta_t(s_t, a_t) + \eta \cdot \frac{Q^{\pi_{\theta_t}}(s_t, a_t) - V^{\pi_{\theta_t}}(s_t)}{\pi_{\theta_t}(a_t|s_t)}$.
**end while**

---

According to the performance difference lemma, we have,

$$V^{\pi_{\theta_{t+1}}}(\mu) - V^{\pi_{\theta_t}}(\mu) = \frac{1}{1-\gamma} \cdot \sum_s d_\mu^{\pi_{\theta_{t+1}}}(s) \cdot \sum_a \left( \pi_{\theta_{t+1}}(a|s) - \pi_{\theta_t}(a|s) \right) \cdot Q^{\pi_{\theta_t}}(s, a), \qquad (18)$$

where the inner summation over actions is similar to $\left( \pi_{\theta_{t+1}} - \pi_{\theta_t} \right)^\top r$ in one-state MDPs. This connection allows us to generalize Lemma 1 to the following result.

**Lemma 3** (Stochastic NŁ). *Using Algorithm 1 with constant $\eta > 0$, we have, for all $t \geq 1$,*

$$V^{\pi_{\theta_{t+1}}}(s_0) - V^{\pi_{\theta_t}}(s_0) \geq 0, \qquad a.s., \qquad \forall s_0 \in \mathcal{S}, \qquad and \qquad (19)$$

$$\mathbb{E}_t[V^{\pi_{\theta_{t+1}}}(\mu)] - V^{\pi_{\theta_t}}(\mu) \geq \frac{\eta \cdot (1-\gamma)^4 \cdot \min_s \mu(s)}{1+\eta} \cdot \left\| \frac{d_\mu^{\pi^*}}{\mu} \right\|_\infty^{-1} \cdot \frac{\min_s \pi_{\theta_t}(a^*(s)|s)^2}{S} \cdot \left( V^{\pi^*}(\mu) - V^{\pi_{\theta_t}}(\mu) \right)^2. \quad (20)$$

*where $\mathbb{E}_t[\cdot]$ is on randomness from state sampling $s_t \sim d_\mu^{\pi_{\theta_t}}(\cdot)$, on-policy sampling $a_t \sim \pi_{\theta_t}(\cdot|s_t)$, and $a^*(s)$ is the action selected by the optimal policy $\pi^*$ under state $s$.*

Next, similar to Lemma 2, we can develop a set of contradictions that establish the following result.

**Lemma 4** (Non-vanishing stochastic NŁ coefficient / "automatic exploration"). *Using Algorithm 1 with the conditions in Lemma 3, with arbitrary initialization $\theta_1 \in \mathbb{R}^{\mathcal{S} \times \mathcal{A}}$, we have,*

$$c := \inf_{t \geq 1, s \in \mathcal{S}} \pi_{\theta_t}(a^*(s)|s) > 0, \qquad a.s. \qquad (21)$$

By combining Lemmas 3 and 4, we obtain the following result that generalizes Theorem 1.

**Theorem 2** (Almost sure global convergence rate). *Using Algorithm 1 with any initialization $\theta_1 \in \mathbb{R}^K$, under the same assumptions as Lemmas 3, there exists a $C > 0$ such that for all $t \geq 1$,*

$$\mathbb{E}[V^*(\mu) - V^{\pi_{\theta_t}}(\mu)] \leq \frac{1+\eta}{\eta \cdot (1-\gamma)^4 \cdot \min_s \mu(s)} \cdot \left\| \frac{d_\mu^{\pi^*}}{\mu} \right\|_\infty \cdot \frac{S}{\mathbb{E}[c^2]} \cdot \frac{1}{t}, \qquad and \qquad (22)$$

$$\limsup_{t \geq 1} \left\{ \frac{\eta \cdot (1-\gamma)^4 \cdot \min_s \mu(s)}{1+\eta} \cdot \left\| \frac{d_\mu^{\pi^*}}{\mu} \right\|_\infty^{-1} \cdot \frac{c^2 \cdot t}{S} \cdot (V^*(\mu) - V^{\pi_{\theta_t}}(\mu)) \right\} < \infty, \qquad a.s., \qquad (23)$$

*where $\pi^*$ is the global optimal policy, $S$ is the state number, $\min_s \mu(s) > 0$ by Assumption 2, and $c := \inf_{t \geq 1, s \in \mathcal{S}} \pi_{\theta_t}(a^*(s)|s) > 0$ is from Lemma 4.*

## 3 Understanding Baselines in On-policy Stochastic Policy Optimization

Section 2 shows that using a value function baseline in on-policy stochastic NPG can ensure convergence to a globally optimal policy. However, the mechanism behind this finding requires further elucidation. Preliminary studies [7, 21] have observed that subtracting a baseline can reduce the committal behavior of PG-based estimators, suggesting that this effect might be more important than variance reduction. A mathematical characterization of "committal behavior" is from using the following concept of "committal rate" [21].

**Definition 3** (Committal Rate, Definition 2 of [21]). *Fix $r \in (0,1]^K$ and $\theta_1 \in \mathbb{R}^K$. Consider a policy optimization algorithm $\mathcal{A}$. Let action $a$ be the sampled action **forever** after initialization and let $\theta_t$ be produced by $\mathcal{A}$ on the first $t$ observations. The committal rate of algorithm $\mathcal{A}$ on action $a$ (given $r$ and $\theta_1$) is,*

$$\kappa(\mathcal{A}, a) = \sup \left\{ \alpha \geq 0 : \limsup_{t \to \infty} t^\alpha \cdot [1 - \pi_{\theta_t}(a)] < \infty \right\}. \tag{24}$$

The larger the committal rate $\kappa$ is, the more aggressive one update is. In this section, we provide a new, deeper understanding of how a baseline improves the convergence behaviour of a stochastic PG based method using Definition 3. However, [21] only studied the deterministic reward setting i.e., $\hat{r}_t$ is from Definition 2. We follow the same settings in this section.

### 3.1 Baselines Do Not Control Update Variance in NPG

We begin from the well known result that value baselines have no effect on exact policy gradients.

**Proposition 2** (Unbiasedness of NPG). *For NPG with and without a state value baseline, corresponding to Updates 1 and 2 respectively, we have $\mathbb{E}_{a_t \sim \pi_{\theta_t}(\cdot)} [\hat{r}_t] = \mathbb{E}_{a_t \sim \pi_{\theta_t}(\cdot)} [\hat{r}_t - \hat{b}_t] = r$.*

According to Proposition 2, Updates 1 and 2 become identical if the exact policy gradient is available, hence both enjoy an $O(e^{-c \cdot t})$ convergence rate to a global optimum ($c > 0$) [14, 21]. Therefore, a state value baseline can only have an effect if the policy gradient has to be estimated from a stochastic sample. However, we find that the variance of the NPG updates remains unbounded in the stochastic setting, regardless of whether a state value baseline is used.

**Proposition 3** (Unboundedness of NPG). *For NPG without a baseline, Update 1, we have $\mathbb{E}_{a_t \sim \pi_{\theta_t}(\cdot)} \|\hat{r}_t\|_2^2 = \sum_{a \in [K]} \frac{r(a)^2}{\pi_{\theta_t}(a)}$. For NPG with a state value baseline, Update 2, we have $\mathbb{E}_{a_t \sim \pi_{\theta_t}(\cdot)} \|\hat{r}_t - \hat{b}_t\|_2^2 = \sum_{a \in [K]} \frac{(r(a) - \pi_{\theta_t}^\top r)^2}{\pi_{\theta_t}(a)} - K \cdot (\pi_{\theta_t}^\top r)^2 + 2 \cdot (\pi_{\theta_t}^\top r) \cdot (r^\top \mathbf{1})$.*

According to Proposition 3, whenever $\pi_{\theta_t}$ nears a one-hot probability distribution over $[K]$ (which it must converge to), there exists at least one action $a \in [K]$ such that both $\frac{r(a)^2}{\pi_{\theta_t}(a)}$ and $\frac{(r(a) - \pi_{\theta_t}^\top r)^2}{\pi_{\theta_t}(a)}$ become unbounded, implying an unbounded scale for both Updates 1 and 2. Yet we know from Proposition 1 that not using a baseline fails with positive probability, while from Theorem 1 subtracting a state value baseline ensures almost sure convergence to a global optimum. The fact that the variance of both updates is unbounded suggests that it is difficult to draw conclusions on the effect of the baseline from a variance reduction perspective alone. An alternative analysis is required to explain the fundamental difference between Updates 1 and 2.

### 3.2 Coupled Sampling and Updating

In on-policy stochastic policy optimization, sampling and updating are coupled as shown in Figure 1. At iteration $t$, the data collected depends on the current policy, since on-policy sampling is used $a_t \sim \pi_{\theta_t}(\cdot)$, while the policy is updated from the observations collected based on $a_t$. This coupling introduces complexity in the optimization process as well as in the analysis. However, this coupling is also fundamental to understanding the circular interaction created by any on-policy stochastic optimization method. That is, on-policy stochastic optimization faces an exploration-exploitation dilemma: a learning algorithm can improve the policy and increase the probability of choosing actions that yield higher rewards (exploitation), but it must not do so too aggressively lest it fail to identify possibly higher-reward actions (exploration). Striking a proper balance between exploration and exploitation is key to achieving good convergence properties. Different levels of update aggression create different circular effects between sampling and updating, which is central to determining almost sure convergence to a global optimum.

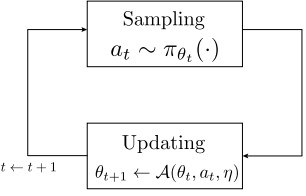

Figure 1: Coupled on-policy sampling and updating [21, Figure 2].

### 3.3 The "Vicious Circle" of Being Too Aggressive

First we illustrate a negative effect, the "vicious circle" of being too aggressive.

**Lemma 5** (Bad sampling). *Let $\pi_{\theta_t}(a) \in (0, 1)$ be the probability of sampling action $a$ using online sampling $a_t \sim \pi_{\theta_t}(\cdot)$, for all $t \geq 1$. If $1 - \pi_{\theta_t}(a) \in O(1/t^{1+\epsilon})$, where $\epsilon > 0$, then $\prod_{t=1}^{\infty} \pi_{\theta_t}(a) > 0$.*

Note that Lemma 5 characterizes sampling behaviour under general conditions that do not otherwise depend on specific updates. However, according to Lemma 5, if an action's probability approaches 1 strictly faster than $O(1/t)$, by whatever means, it becomes possible to not sample any other action forever, which creates a "lack of exploration" phenomenon as it is known in RL. In particular, on-policy stochastic NPG without a baseline can produce such a sequence of $\{\pi_{\theta_t}(a)\}_{t \geq 1}$.

**Lemma 6** (NPG aggressiveness). *Fix sampling $a_t = a$ for all $t \geq 1$, using Update 1 with constant learning rate $\eta > 0$, where $\hat{r}_t$ is from Definition 2, we have $1 - \pi_{\theta_t}(a) \in O(e^{-c \cdot t})$ for all $t \geq 1$, where $c > 0$.*

According to Definition 3, we have $\kappa(\text{NPG}, a) = \infty$, meaning that NPG without baseline is very aggressive. Note that Lemma 6 only characterizes the aggressiveness of Update 1 with the sampling fixed to be $a_t = a$ for all $t \geq 1$. Lemmas 5 and 6 together describe the "vicious circle" between sampling and updating that can be created by overly aggressive updates. First, in on-policy sampling, there will always be a non-zero probability of "bad luck"; that is, with positive probability a set of sub-optimal actions can be sequentially sampled for multiple steps. Second, an overly aggressive update will only exaggerate the weakness of the sampling procedure by increasing the sampled sub-optimal actions' probabilities rapidly (Lemma 6). Third, this exaggeration can worsen data collection for subsequent updating by further increasing the prevalence of sub-optimal actions. Such a vicious circular interaction between sampling and updating can happen repeatedly, and its self-reinforcing nature can create a non-zero probability that the cycle occurs forever (Lemma 5), resulting in convergence to a sub-optimal deterministic policy (a stationary point for both sampling and updating).

### 3.4 The "Virtuous Circle" of Not Being Too Aggressive

Next, we demonstrate a positive effect, the "virtuous circle" of not being too aggressive.

**Lemma 7** (Good sampling). *Let $\pi_{\theta_t}(a) \in (0, 1)$ and $a_t \sim \pi_{\theta_t}(\cdot)$, for all $t \geq 1$. If $\sum_{t=1}^{\infty} (1 - \pi_{\theta_t}(a)) = \infty$ (e.g., $1 - \pi_{\theta_t}(a) \in \Omega(1/t)$), then $\prod_{t=1}^{\infty} \pi_{\theta_t}(a) = 0$.*

As in Lemma 5, Lemma 7 only characterizes the effect of sampling behaviour under general conditions that do not otherwise depend on specific updates. Here we see that if an action's probability approaches 1 no faster than $O(1/t)$, it is no longer possible to avoid sampling any other action forever; that is, sufficiently slow modification of the sampling probabilities forces persistent exploration such that every action is sampled within some finite time with probability 1. In particular, subtracting a value baseline in on-policy stochastic NPG produces such a sequence $\{\pi_{\theta_t}(a)\}_{t \geq 1}$.

**Lemma 8** (Value baselines reduce NPG aggressiveness). *Fix sampling $a_t = a$ for all $t \geq 1$. Then using Update 2 with a constant learning rate $\eta > 0$ and $\hat{r}_t$ from Definition 2 obtains $1 - \pi_{\theta_t}(a) \in \Omega(1/t)$ for all $t \geq 1$.*

According to Definition 3, with value baselines, we have $\kappa(\text{NPG}, a) = 1$, meaning that the aggressiveness of NPG update is reduced. As in Lemma 6, Lemma 8 only characterizes the conservativeness of Update 2 with fixed sampling of $a_t = a$ for all $t \geq 1$. Lemmas 7 and 8 now describe a "virtuous circle" between sampling and updating that is created by using not too aggressive updates. First, even in a worst case situation (e.g., an adversarial initialization), where a sub-optimal action has a dominant probability $\pi_{\theta_t}(a) \approx 1$, under on-policy sampling all actions will eventually be sampled. Second, conservative updating will mitigate the effect of the extreme sampler by not increasing the sub-optimal action's probability too rapidly (Lemma 8). Third, sustained diversity in sampling will eventually draw a better action than the current dominating sub-optimal action (Lemma 7). Finally, once better actions are sampled, the update will improve subsequent sampling by decreasing the probability of the dominating sub-optimal action. In particular, this is achieved by increasing value baselines to be larger than the dominating sub-optimal action's true mean reward, such that the dominating sub-optimal action will start losing probabilities. This virtuous circular interaction between sampling and updating ensures sufficient exploration, which prevents the iteration from converging to a sub-optimal deterministic policy.

## 3.5 How a State Value Baseline Reduces Update Aggressiveness

Based on Lemmas 5 and 7, the boundary between "too aggressive" and "not too aggressive" is precisely $\Theta(1/t)$. We now explain how a state value baseline in NPG will control update aggressiveness. First, without a baseline, sampling a sub-optimal action $a \in [K]$ for $t$ times makes its parameter behave as $\theta_t(a) \in \Theta(t)$, since $r(a) \in \Theta(1)$. On the other hand, other action parameters will behave as $\theta_t(a') \in \Theta(1)$ if they are only sampled a constant number of times. Under the softmax parameterization Eq. (2), this will imply that $1 - \pi_{\theta_t}(a) \in O(e^{-c \cdot t})$, which is far too aggressive. Second, using a state value baseline, under repeated sampling the parameter increase for a sub-optimal action $a \in [K]$ will be damped. In particular, whenever the policy is close to deterministic, say $\pi_{\theta_t}(a) \approx 1$, we also have $\pi_{\theta_t}^\top r \approx r(a)$. Therefore, since

$$r(a) - \pi_{\theta_t}^\top r = \sum_{a' \neq a} \pi_{\theta_t}(a') \cdot (r(a) - r(a')) \leq 1 - \pi_{\theta_t}(a), \tag{25}$$

the closer $1 - \pi_{\theta_t}(a)$ is to 0, the smaller $r(a) - \pi_{\theta_t}^\top r$ will be. This means even if $a$ is sampled repeatedly for $t$ times, we obtain $\theta_t(a) \in O(\log t)$ and $1 - \pi_{\theta_t}(a) \in \Omega(1/t)$ (Lemma 8). Thus, the effect of baseline is to modify the sampling to lie exactly on the boundary of being good enough. From this argument the key role of the value baseline is to reduce update aggressiveness to achieve a particular effect on long-term sampling, rather than simply reduce variance. It also shows how using an appropriately un-aggressive update is both necessary (Lemma 5) and sufficient (Lemma 7) to achieve almost sure convergence to a global optimum in on-policy stochastic policy optimization.

# 4 Simulations

We conducted simulations to verify the two main results above: asymptotic convergence toward globally optimal policy $\pi^*$ in Lemma 2, and the $O(1/t)$ convergence rate in Theorem 1.

## 4.1 Asymptotic Convergence

We first consider a one-state MDP with $K = 20$ actions and true mean reward vector $r \in (0, 1)^K$, where the optimal action is $a^* = 1$ with true mean reward $r(1) \approx 0.97$ and best sub-optimal action's true mean reward $r(2) \approx 0.95$. The sampled reward is observed with a large noise, e.g., $x \approx -2.03$ and $x \approx 3.97$ with both 0.5 probability for the optimal action, such that $r(1) \approx 0.5 \cdot (-2.03) + 0.5 \cdot 3.97$. Details about $r$ and the reward distributions can be found in the appendix.

To verify asymptotic convergence to a globally optimal policy in Lemma 2, we consider the iteration behaviors of Update 2 under an adversarial initialization, where $\pi_{\theta_1}(2) \approx 0.88$, i.e., a sub-optimal action starts with a dominating probability. This is the worst case scenario for Lemma 2, where the optimal action only has a small chance to be sampled, while the sampled reward noise is very large.

As shown in Figure 2a, the expected reward $\pi_{\theta_t}^\top r$ quickly approaches and remains stuck around $r(2) \approx 0.95$ initially, as expected. However, after about $8 \times 10^6$ iterations, the policy $\pi_{\theta_t}$ finally escapes the sub-optimal plateau and approaches the optimal reward $r(1) \approx 0.97$. This simulation result is consistent with Lemma 2, i.e., for an arbitrary initialization, the introduction of a value baseline eventually makes $\pi_{\theta_t}$ approach a globally optimal policy within finite time, while additionally the optimal action's probability never vanishes, $\inf_{t \geq 1} \pi_{\theta_t}(a^*) > 0$, as shown in Figure 2b.

## 4.2 Convergence Rate

We run Update 2 with a uniform initialization, i.e., $\pi_{\theta_1}(a) = 1/K$ for all $a \in [K]$, and calculate averaged sub-optimality gap $(\pi^* - \pi_{\theta_t})^\top r$ across 20 independent runs, using deterministic reward settings where $\hat{r}_t$ is from Definition 2. As shown in Figure 2c, where both axes are in $\log$ scale, the slope is approximately $-1$, indicating that $\log (\pi^* - \pi_{\theta_t})^\top r = -\log t + C$, or equivalently $(\pi^* - \pi_{\theta_t})^\top r = C'/t$, which is consistent with Theorem 1.

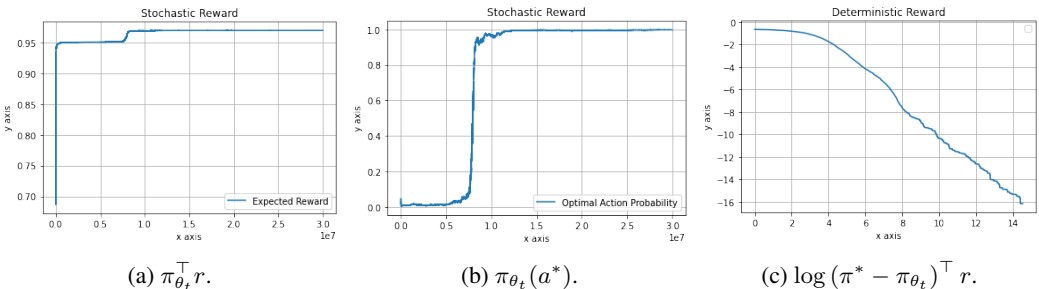

(a) $\pi_{\theta_t}^\top r$.       (b) $\pi_{\theta_t}(a^*)$.       (c) $\log \left(\pi^* - \pi_{\theta_t}\right)^\top r$.

Figure 2: Adversarial initialization (a) and (b); uniform initialization (c).

## 5 Conclusion

This work clarifies some of the longstanding mysteries those have separated the theory and practice of policy gradient optimization. The major finding is a state value baseline reduces the aggressiveness of the on-policy stochastic NPG update, which turns out to be necessary and sufficient for achieving almost sure convergence to a global optimum. The deeper understanding of the circular dependence between on-policy sampling and updating also dispels a common misconception about variance reduction, showing that bounded variance estimators are not necessary for achieving global convergence. The main technical innovation is the stochastic NŁ inequality, and the subsequent arguments that establish global convergence, both of which depend critically on the value baseline.

This work leaves open a number of interesting questions. *First*, the $O(1/t)$ convergence rate contains an initialization dependent constant in Lemma 2, resulting from plateaus as observed in Figure 2a, which does not appear in results that use the direct parameterization [8]. Thus the difficulty appears due to the non-linear softmax transform. Removing or improving this constant would impact practical performance, so investigating other techniques, such as regularization, optimism or momentum might be helpful. *Second*, the results in this paper use the true state values as the baselines. It would be interesting to consider the effect of estimating the value baseline or using alternative baselines in policy optimization. *Finally*, the $O(1/t)$ last iteration convergence rate implies an optimal $O(\log T)$ regret in stochastic bandit problems [16]. The explanation of the circular dependence between sampling and updating is specific to on-policy PG optimization, but it is also consistent with the exploration exploitation dilemma in RL. In other words, this work suggests a completely new approach to the exploration-exploitation trade-off, achieving provable bounds with ever requiring explicit uncertainty estimates, nor any concrete instantiation of the principle of optimism under uncertainty.

## Acknowledgments and Disclosure of Funding

The authors would like to thank anonymous reviewers for their valuable comments. Jincheng Mei thanks Alekh Agarwal for reviewing a draft of this work. Csaba Szepesvári and Dale Schuurmans gratefully acknowledge funding from the Canada CIFAR AI Chairs Program, Amii and NSERC.

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
