# Appendix

The appendix is organized as follows.

## A  Proofs for One-state MDPs

**Lemma 1** (Stochastic non-uniform Łojasiewicz (NŁ))**.** Suppose Assumption 1 holds. Let $r \in [0,1]^K$, $a^* := \arg\max_{a \in [K]} r(a)$ denote the optimal action, and $\Delta := r(a^*) - \max_{a \neq a^*} r(a)$ denote the reward gap. Using Update 2 with on-policy sampling $a_t \sim \pi_{\theta_t}(\cdot)$ and IS estimator $\hat{r}_t$,

**(1)** if $\hat{r}_t$ is from Definition 2, then with constant learning rate $\eta > 0$, we have, for all $t \geq 1$,

$$\pi_{\theta_{t+1}}^\top r - \pi_{\theta_t}^\top r \geq 0, \qquad \text{almost surely (a.s.),} \qquad \text{and} \tag{26}$$

$$\mathbb{E}_t[\pi_{\theta_{t+1}}^\top r] - \pi_{\theta_t}^\top r \geq \frac{\eta}{1+\eta} \cdot \pi_{\theta_t}(a^*) \cdot \left(r(a^*) - \pi_{\theta_t}^\top r\right)^2, \tag{27}$$

where $\mathbb{E}_t[\cdot]$ is on randomness from on-policy sampling $a_t \sim \pi_{\theta_t}(\cdot)$.

**(2)** if $\hat{r}_t$ is from Definition 1, then with learning rate,

$$\eta = \frac{\pi_{\theta_t}(a_t) \cdot \left|r(a_t) - \pi_{\theta_t}^\top r\right|}{8 \cdot R_{\max}^2}, \tag{28}$$

we have, for all $t \geq 1$,

$$\mathbb{E}_t[\pi_{\theta_{t+1}}^\top r] - \pi_{\theta_t}^\top r \geq \frac{1}{16 \cdot R_{\max}^2} \cdot \sum_{i=1}^K \pi_{\theta_t}(i)^2 \cdot \left|r(i) - \pi_{\theta_t}^\top r\right|^3 \tag{29}$$

$$\geq \frac{1}{16 \cdot R_{\max}^2} \cdot \frac{\Delta}{K-1} \cdot \pi_{\theta_t}(a^*)^2 \cdot \left(r(a^*) - \pi_{\theta_t}^\top r\right)^2, \tag{30}$$

where $\mathbb{E}_t[\cdot]$ is on randomness from on-policy sampling $a_t \sim \pi_{\theta_t}(\cdot)$ and reward sampling $x \sim R_{a_t}$.

*Proof.* **First part. (1)** If $\hat{r}_t$ is from Definition 2.

Since the results are concerned with the policies $\{\pi_{\theta_t}\}_{t \geq 1}$ underlying the parameter $\{\theta_t\}_{t \geq 1}$ and not the parameter vectors themselves, as noted after Update 2, without loss of generality, in the rest of the proof we assume that the update over parameter vectors is according to,

$$\theta_{t+1}(a) \leftarrow \theta_t(a) + \eta \cdot \frac{\mathbb{I}\{a_t = a\}}{\pi_{\theta_t}(a)} \cdot \left(r(a) - \pi_{\theta_t}^\top r\right). \tag{31}$$

For all $t \geq 1$, for any action $i \in [K]$, denote

$$\left[\pi_{\theta_{t+1}}^\top r \mid a_t = i\right] \tag{32}$$

as the the value of $\pi_{\theta_{t+1}}^\top r$ given the sampled action $a_t = i$.

According to Eq. (31) and Definition 2, we have,

$$\left[\pi_{\theta_{t+1}}^\top r \mid a_t = i\right] = \frac{\exp\left\{\theta_t(i) + \eta \cdot \frac{r(i) - \pi_{\theta_t}^\top r}{\pi_{\theta_t}(i)}\right\} \cdot r(i) + \sum_{j \neq i} \exp\{\theta_t(j)\} \cdot r(j)}{\exp\left\{\theta_t(i) + \eta \cdot \frac{r(i) - \pi_{\theta_t}^\top r}{\pi_{\theta_t}(i)}\right\} + \sum_{j \neq i} \exp\{\theta_t(j)\}} \tag{33}$$

$$= \frac{\pi_{\theta_t}(i) \cdot \exp\left\{\eta \cdot \frac{r(i) - \pi_{\theta_t}^\top r}{\pi_{\theta_t}(i)}\right\} \cdot r(i) + \sum_{j \neq i} \pi_{\theta_t}(j) \cdot r(j)}{\pi_{\theta_t}(i) \cdot \exp\left\{\eta \cdot \frac{r(i) - \pi_{\theta_t}^\top r}{\pi_{\theta_t}(i)}\right\} + \sum_{j \neq i} \pi_{\theta_t}(j)}, \tag{34}$$

where the last equation is by dividing $\sum_{a \in [K]} \exp\left\{\theta_t(a)\right\}$ from both the numerator and the denominator. Therefore, by algebra we have,

$$\left[\pi_{\theta_{t+1}}^\top r \mid a_t = i\right] - \pi_{\theta_t}^\top r = \frac{\left[\pi_{\theta_t}(i) \cdot \exp\left\{\eta \cdot \frac{r(i) - \pi_{\theta_t}^\top r}{\pi_{\theta_t}(i)}\right\} - \pi_{\theta_t}(i)\right] \cdot \left(r(i) - \pi_{\theta_t}^\top r\right)}{\pi_{\theta_t}(i) \cdot \exp\left\{\eta \cdot \frac{r(i) - \pi_{\theta_t}^\top r}{\pi_{\theta_t}(i)}\right\} + \sum_{j \neq i} \pi_{\theta_t}(j)} \tag{35}$$

$$= \frac{\left[\exp\left\{\eta \cdot \frac{r(i) - \pi_{\theta_t}^\top r}{\pi_{\theta_t}(i)}\right\} - 1\right] \cdot \left(r(i) - \pi_{\theta_t}^\top r\right)}{\exp\left\{\eta \cdot \frac{r(i) - \pi_{\theta_t}^\top r}{\pi_{\theta_t}(i)}\right\} + \frac{1 - \pi_{\theta_t}(i)}{\pi_{\theta_t}(i)}} \geq 0, \tag{36}$$

where the last inequality is from $(e^{c \cdot y} - 1) \cdot y \geq 0$ for all $y \in \mathbb{R}$ with $c := \frac{\eta}{\pi_{\theta_t}(i)} > 0$. This proves Eq. (26), because of $i \in [K]$ is arbitrary.

For all $t \geq 1$, given current policy $\pi_{\theta_t}$, the expected reward of next policy $\pi_{\theta_{t+1}}^\top r$ is a random variable, and the randomness is from on-policy sampling $a_t \sim \pi_{\theta_t}(\cdot)$. The expected progress is,

$$\mathbb{E}_t[\pi_{\theta_{t+1}}^\top r] - \pi_{\theta_t}^\top r = \sum_{i=1}^{K} \pi_{\theta_t}(i) \cdot \mathbb{E}_t[\pi_{\theta_{t+1}}^\top r \mid a_t = i] - \pi_{\theta_t}^\top r \qquad (a_t \sim \pi_{\theta_t}(\cdot)) \tag{37}$$

$$= \sum_{i=1}^{K} \pi_{\theta_t}(i) \cdot \left(\left[\pi_{\theta_{t+1}}^\top r \mid a_t = i\right] - \pi_{\theta_t}^\top r\right) \tag{38}$$

$$= \sum_{i=1}^{K} \pi_{\theta_t}(i) \cdot \frac{\left[\exp\left\{\eta \cdot \frac{r(i) - \pi_{\theta_t}^\top r}{\pi_{\theta_t}(i)}\right\} - 1\right] \cdot \left(r(i) - \pi_{\theta_t}^\top r\right)}{\exp\left\{\eta \cdot \frac{r(i) - \pi_{\theta_t}^\top r}{\pi_{\theta_t}(i)}\right\} + \frac{1 - \pi_{\theta_t}(i)}{\pi_{\theta_t}(i)}} \qquad \text{(by Eq. (35))} \tag{39}$$

where $\left[\pi_{\theta_{t+1}}^\top r \mid a_t = i\right]$ means the value of $\pi_{\theta_{t+1}}^\top r$ given the sampled action $a_i = i$.

Partition the action set $[K]$ into three parts using $\pi_{\theta_t}^\top r$ as follows,

$$\mathcal{A}_t^0 := \left\{a^0 \in [K] : r(a^0) = \pi_{\theta_t}^\top r\right\}, \tag{40}$$

$$\mathcal{A}_t^+ := \left\{a^+ \in [K] : r(a^+) > \pi_{\theta_t}^\top r\right\}, \tag{41}$$

$$\mathcal{A}_t^- := \left\{a^- \in [K] : r(a^-) < \pi_{\theta_t}^\top r\right\}. \tag{42}$$

From Eq. (37), we have,

$$\mathbb{E}_t[\pi_{\theta_{t+1}}^\top r] - \pi_{\theta_t}^\top r = \sum_{a^+ \in \mathcal{A}_t^+} \pi_{\theta_t}(a^+) \cdot \frac{\left[\exp\left\{\eta \cdot \frac{r(a^+) - \pi_{\theta_t}^\top r}{\pi_{\theta_t}(a^+)}\right\} - 1\right] \cdot \left(r(a^+) - \pi_{\theta_t}^\top r\right)}{\exp\left\{\eta \cdot \frac{r(a^+) - \pi_{\theta_t}^\top r}{\pi_{\theta_t}(a^+)}\right\} + \frac{1 - \pi_{\theta_t}(a^+)}{\pi_{\theta_t}(a^+)}} \tag{43}$$

$$+ \sum_{a^- \in \mathcal{A}_t^-} \pi_{\theta_t}(a^-) \cdot \frac{\left[\exp\left\{\eta \cdot \frac{r(a^-) - \pi_{\theta_t}^\top r}{\pi_{\theta_t}(a^-)}\right\} - 1\right] \cdot \left(r(a^-) - \pi_{\theta_t}^\top r\right)}{\exp\left\{\eta \cdot \frac{r(a^-) - \pi_{\theta_t}^\top r}{\pi_{\theta_t}(a^-)}\right\} + \frac{1 - \pi_{\theta_t}(a^-)}{\pi_{\theta_t}(a^-)}}. \tag{44}$$

For any $a^+ \in \mathcal{A}_t^+$, we have,

$$\frac{\left[\exp\left\{\eta \cdot \frac{r(a^+)-\pi_{\theta_t}^\top r}{\pi_{\theta_t}(a^+)}\right\} - 1\right] \cdot \left(r(a^+) - \pi_{\theta_t}^\top r\right)}{\exp\left\{\eta \cdot \frac{r(a^+)-\pi_{\theta_t}^\top r}{\pi_{\theta_t}(a^+)}\right\} + \frac{1-\pi_{\theta_t}(a^+)}{\pi_{\theta_t}(a^+)}} \geq \frac{\eta \cdot \frac{r(a^+)-\pi_{\theta_t}^\top r}{\pi_{\theta_t}(a^+)} \cdot \left(r(a^+) - \pi_{\theta_t}^\top r\right)}{\eta \cdot \frac{r(a^+)-\pi_{\theta_t}^\top r}{\pi_{\theta_t}(a^+)} + \frac{1}{\pi_{\theta_t}(a^+)}} \qquad (e^x - 1 \geq x > 0) \tag{45}$$

$$= \frac{\eta \cdot \left(r(a^+) - \pi_{\theta_t}^\top r\right)^2}{\eta \cdot \left(r(a^+) - \pi_{\theta_t}^\top r\right) + 1} \geq \frac{\eta}{1+\eta} \cdot \left(r(a^+) - \pi_{\theta_t}^\top r\right)^2. \qquad \left(r \in [0,1]^K\right) \tag{46}$$

For any $a^- \in \mathcal{A}_t^-$, we have,

$$\frac{\left[\exp\left\{\eta \cdot \frac{r(a^-)-\pi_{\theta_t}^\top r}{\pi_{\theta_t}(a^-)}\right\} - 1\right] \cdot \left(r(a^-) - \pi_{\theta_t}^\top r\right)}{\exp\left\{\eta \cdot \frac{r(a^-)-\pi_{\theta_t}^\top r}{\pi_{\theta_t}(a^-)}\right\} + \frac{1-\pi_{\theta_t}(a^-)}{\pi_{\theta_t}(a^-)}} = \frac{\left[\exp\left\{\eta \cdot \frac{\pi_{\theta_t}^\top r - r(a^-)}{\pi_{\theta_t}(a^-)}\right\} - 1\right] \cdot \left(\pi_{\theta_t}^\top r - r(a^-)\right)}{\left[\exp\left\{\eta \cdot \frac{\pi_{\theta_t}^\top r - r(a^-)}{\pi_{\theta_t}(a^-)}\right\} - 1\right] \cdot \frac{1-\pi_{\theta_t}(a^-)}{\pi_{\theta_t}(a^-)} + \frac{1}{\pi_{\theta_t}(a^-)}} \tag{47}$$

$$\geq \frac{\eta \cdot \frac{\pi_{\theta_t}^\top r - r(a^-)}{\pi_{\theta_t}(a^-)} \cdot \left(\pi_{\theta_t}^\top r - r(a^-)\right)}{\eta \cdot \frac{\pi_{\theta_t}^\top r - r(a^-)}{\pi_{\theta_t}(a^-)} \cdot \frac{1-\pi_{\theta_t}(a^-)}{\pi_{\theta_t}(a^-)} + \frac{1}{\pi_{\theta_t}(a^-)}} \qquad (e^x - 1 \geq x > 0) \tag{48}$$

$$= \frac{\eta \cdot \pi_{\theta_t}(a^-) \cdot \left(\pi_{\theta_t}^\top r - r(a^-)\right)^2}{\eta \cdot \left(\pi_{\theta_t}^\top r - r(a^-)\right) \cdot \left(1 - \pi_{\theta_t}(a^-)\right) + \pi_{\theta_t}(a^-)} \tag{49}$$

$$\geq \frac{\eta}{1+\eta} \cdot \pi_{\theta_t}(a^-) \cdot \left(\pi_{\theta_t}^\top r - r(a^-)\right)^2 \qquad \left(r \in [0,1]^K, \ \pi_{\theta_t}(a^-) \in (0,1)\right) \tag{50}$$

Combining Eqs. (43), (45) and (47), we have,

$$\mathbb{E}_t[\pi_{\theta_{t+1}}^\top r] - \pi_{\theta_t}^\top r \geq \sum_{a^+ \in \mathcal{A}_t^+} \pi_{\theta_t}(a^+) \cdot \frac{\eta}{1+\eta} \cdot \left(r(a^+) - \pi_{\theta_t}^\top r\right)^2 \tag{51}$$

$$+ \sum_{a^- \in \mathcal{A}_t^-} \pi_{\theta_t}(a^-) \cdot \frac{\eta}{1+\eta} \cdot \pi_{\theta_t}(a^-) \cdot \left(\pi_{\theta_t}^\top r - r(a^-)\right)^2 \tag{52}$$

$$\geq \frac{\eta}{1+\eta} \cdot \pi_{\theta_t}(a^*) \cdot \left(r(a^*) - \pi_{\theta_t}^\top r\right)^2. \qquad \left(a^* \in \mathcal{A}_t^+\right) \tag{53}$$

**Second part. (2)** If $\hat{r}_t$ is from Definition 1.

As noted after Update 2, we analyze Update 3, which is duplicated as follows,

$$\theta_{t+1}(a) \leftarrow \theta_t(a) + \eta \cdot \frac{\mathbb{I}\{a_t = a\}}{\pi_{\theta_t}(a)} \cdot \left(x_t(a) - \pi_{\theta_t}^\top r\right). \tag{54}$$

For all $t \geq 1$, given current policy $\pi_{\theta_t}$, the expected reward of next policy $\pi_{\theta_{t+1}}^\top r$ is a random variable, and the randomness is from on-policy sampling $a_t \sim \pi_{\theta_t}(\cdot)$ and reward sampling $x \sim R_{a_t}$. The expected progress after one update is,

$$\mathbb{E}_t[\pi_{\theta_{t+1}}^\top r] - \pi_{\theta_t}^\top r = \sum_{i=1}^K \pi_{\theta_t}(i) \cdot \mathbb{E}_t[\pi_{\theta_{t+1}}^\top r \mid a_t = i] - \pi_{\theta_t}^\top r \qquad (a_t \sim \pi_{\theta_t}(\cdot)) \tag{55}$$

$$= \sum_{i=1}^K \pi_{\theta_t}(i) \cdot \underbrace{\left(\mathbb{E}_t[\pi_{\theta_{t+1}}^\top r \mid a_t = i] - \pi_{\theta_t}^\top r\right)}_{\text{expected progress of } a_t = i} \tag{56}$$

$$= \sum_{i=1}^K \pi_{\theta_t}(i) \cdot \left(\int_{-R_{\max}}^{R_{\max}} [\pi_{\theta_{t+1}}^\top r \mid a_t = i, \ R_t = x] \cdot P_i(x)\mu(dx) - \pi_{\theta_t}^\top r\right) \tag{57}$$

$$= \sum_{i=1}^K \pi_{\theta_t}(i) \cdot \int_{-R_{\max}}^{R_{\max}} \underbrace{\left([\pi_{\theta_{t+1}}^\top r \mid a_t = i, \ R_t = x] - \pi_{\theta_t}^\top r\right)}_{\text{progress of } a_t = i, \ R_t = x} \cdot P_i(x)\mu(dx), \tag{58}$$

where $\left[\pi_{\theta_{t+1}}^\top r \mid a_t = i, \ R_t = x\right]$ means the value of $\pi_{\theta_{t+1}}^\top r$ given the sampled action $a_i = i$ and sampled reward $R_t = x$. According to Eq. (54) and Definition 1, we have,

$$\left[\pi_{\theta_{t+1}}^\top r \mid a_t = i, \ R_t = x\right] = \frac{\exp\left\{\theta_t(i) + \eta \cdot \frac{x - \pi_{\theta_t}^\top r}{\pi_{\theta_t}(i)}\right\} \cdot r(i) + \sum_{j \neq i} \exp\{\theta_t(j)\} \cdot r(j)}{\exp\left\{\theta_t(i) + \eta \cdot \frac{x - \pi_{\theta_t}^\top r}{\pi_{\theta_t}(i)}\right\} + \sum_{j \neq i} \exp\{\theta_t(j)\}} \tag{59}$$

$$= \frac{\pi_{\theta_t}(i) \cdot \exp\left\{\eta \cdot \frac{x - \pi_{\theta_t}^\top r}{\pi_{\theta_t}(i)}\right\} \cdot r(i) + \sum_{j \neq i} \pi_{\theta_t}(j) \cdot r(j)}{\pi_{\theta_t}(i) \cdot \exp\left\{\eta \cdot \frac{x - \pi_{\theta_t}^\top r}{\pi_{\theta_t}(i)}\right\} + \sum_{j \neq i} \pi_{\theta_t}(j)}, \tag{60}$$

where the last equation is by dividing $\sum_{a \in [K]} \exp\{\theta_t(a)\}$ from both the numerator and the denominator. Therefore, by algebra we have,

$$\left[\pi_{\theta_{t+1}}^\top r \mid a_t = i, \ R_t = x\right] - \pi_{\theta_t}^\top r = \frac{\left[\pi_{\theta_t}(i) \cdot \exp\left\{\eta \cdot \frac{x - \pi_{\theta_t}^\top r}{\pi_{\theta_t}(i)}\right\} - \pi_{\theta_t}(i)\right] \cdot \left(r(i) - \pi_{\theta_t}^\top r\right)}{\pi_{\theta_t}(i) \cdot \exp\left\{\eta \cdot \frac{x - \pi_{\theta_t}^\top r}{\pi_{\theta_t}(i)}\right\} + \sum_{j \neq i} \pi_{\theta_t}(j)} \tag{61}$$

$$= \frac{\left[\exp\left\{\eta \cdot \frac{x - \pi_{\theta_t}^\top r}{\pi_{\theta_t}(i)}\right\} - 1\right] \cdot \left(r(i) - \pi_{\theta_t}^\top r\right)}{\exp\left\{\eta \cdot \frac{x - \pi_{\theta_t}^\top r}{\pi_{\theta_t}(i)}\right\} + \frac{1 - \pi_{\theta_t}(i)}{\pi_{\theta_t}(i)}}. \tag{62}$$

Combining Eqs. (55) and (61), we have,

$$\mathbb{E}_t[\pi_{\theta_{t+1}}^\top r] - \pi_{\theta_t}^\top r = \sum_{i=1}^K \pi_{\theta_t}(i) \cdot \int_{-R_{\max}}^{R_{\max}} \frac{\left[\exp\left\{\eta \cdot \frac{x - \pi_{\theta_t}^\top r}{\pi_{\theta_t}(i)}\right\} - 1\right] \cdot \left(r(i) - \pi_{\theta_t}^\top r\right)}{\exp\left\{\eta \cdot \frac{x - \pi_{\theta_t}^\top r}{\pi_{\theta_t}(i)}\right\} + \frac{1 - \pi_{\theta_t}(i)}{\pi_{\theta_t}(i)}} \cdot P_i(x)\mu(dx) \tag{63}$$

$$= \sum_{i=1}^K \pi_{\theta_t}(i) \cdot \left(r(i) - \pi_{\theta_t}^\top r\right) \cdot \left[\int_{x \in \mathcal{X}_t^+} \frac{\exp\left\{\eta \cdot \frac{x - \pi_{\theta_t}^\top r}{\pi_{\theta_t}(i)}\right\} - 1}{\exp\left\{\eta \cdot \frac{x - \pi_{\theta_t}^\top r}{\pi_{\theta_t}(i)}\right\} + \frac{1 - \pi_{\theta_t}(i)}{\pi_{\theta_t}(i)}} \cdot P_i(x)\mu(dx) \right. \tag{64}$$

$$\left. + \int_{x \in \mathcal{X}_t^-} \frac{\exp\left\{\eta \cdot \frac{x - \pi_{\theta_t}^\top r}{\pi_{\theta_t}(i)}\right\} - 1}{\exp\left\{\eta \cdot \frac{x - \pi_{\theta_t}^\top r}{\pi_{\theta_t}(i)}\right\} + \frac{1 - \pi_{\theta_t}(i)}{\pi_{\theta_t}(i)}} \cdot P_i(x)\mu(dx)\right], \tag{65}$$

where $\mathcal{X}_t^+$ and $\mathcal{X}_t^-$ are defined by partitioning the sampled reward range $[-R_{\max}, R_{\max}]$ into two parts for the current iteration,

$$\mathcal{X}_t^+ := \left\{x \in [-R_{\max}, R_{\max}] : x - \pi_{\theta_t}^\top r \geq 0\right\} = [\pi_{\theta_t}^\top r, \ R_{\max}], \tag{66}$$

$$\mathcal{X}_t^- := \left\{x \in [-R_{\max}, R_{\max}] : x - \pi_{\theta_t}^\top r < 0\right\} = [-R_{\max}, \ \pi_{\theta_t}^\top r). \tag{67}$$

We next prove that, in Eq. (63), for any sampled action $a_t = i \in [K]$, we have,

$$\int_{-R_{\max}}^{R_{\max}} \frac{\left[\exp\left\{\eta \cdot \frac{x - \pi_{\theta_t}^\top r}{\pi_{\theta_t}(i)}\right\} - 1\right] \cdot \left(r(i) - \pi_{\theta_t}^\top r\right)}{\exp\left\{\eta \cdot \frac{x - \pi_{\theta_t}^\top r}{\pi_{\theta_t}(i)}\right\} + \frac{1 - \pi_{\theta_t}(i)}{\pi_{\theta_t}(i)}} \cdot P_i(x)\mu(dx) \geq \frac{\eta}{2} \cdot \left(r(i) - \pi_{\theta_t}^\top r\right)^2. \tag{68}$$

There are three cases of sampled action $a_t = i \in [K]$.

**Case (a).** $i \in [K]$ is a "good" action at the current iteration, i.e., $r(i) - \pi_{\theta_t}^\top r > 0$.

According to Eq. (486) in Lemma 15, given any fixed $p \in (0, 1]$, and any fixed $\epsilon \in [0, 1]$, we have,

$$f_p(y) := \frac{e^y - 1}{e^y + \frac{1-p}{p}} \geq (1 - \epsilon) \cdot p \cdot y, \quad \text{for all } y \in [0, \epsilon]. \tag{69}$$

Let $p = \pi_{\theta_t}(i) \in (0, 1]$ according to the softmax parameterization. Let

$$\epsilon = \frac{1}{2} \cdot \frac{r(i) - \pi_{\theta_t}^\top r}{\int_{-R_{\max}}^{R_{\max}} \left| x - \pi_{\theta_t}^\top r \right| \cdot P_i(x)\mu(dx)} > 0, \tag{70}$$

where the inequality is because of $r(i) - \pi_{\theta_t}^\top r > 0$. Also note that,

$$\epsilon = \frac{1}{2} \cdot \frac{\left| r(i) - \pi_{\theta_t}^\top r \right|}{\int_{-R_{\max}}^{R_{\max}} \left| x - \pi_{\theta_t}^\top r \right| \cdot P_i(x)\mu(dx)} \qquad \left( r(i) - \pi_{\theta_t}^\top r > 0 \right) \tag{71}$$

$$= \frac{1}{2} \cdot \frac{\left| \int_{-R_{\max}}^{R_{\max}} x \cdot P_i(x)\mu(dx) - \pi_{\theta_t}^\top r \right|}{\int_{-R_{\max}}^{R_{\max}} \left| x - \pi_{\theta_t}^\top r \right| \cdot P_i(x)\mu(dx)} \qquad \text{(by Assumption 1)} \tag{72}$$

$$= \frac{1}{2} \cdot \frac{\left| \int_{-R_{\max}}^{R_{\max}} \left( x - \pi_{\theta_t}^\top r \right) \cdot P_i(x)\mu(dx) \right|}{\int_{-R_{\max}}^{R_{\max}} \left| x - \pi_{\theta_t}^\top r \right| \cdot P_i(x)\mu(dx)} \tag{73}$$

$$\leq \frac{1}{2} \cdot \frac{\int_{-R_{\max}}^{R_{\max}} \left| x - \pi_{\theta_t}^\top r \right| \cdot P_i(x)\mu(dx)}{\int_{-R_{\max}}^{R_{\max}} \left| x - \pi_{\theta_t}^\top r \right| \cdot P_i(x)\mu(dx)} \qquad \text{(by triangle inequality)} \tag{74}$$

$$= 1/2 \leq 1, \tag{75}$$

which means $\epsilon \in (0, 1]$. Let

$$y = \eta \cdot \frac{x - \pi_{\theta_t}^\top r}{\pi_{\theta_t}(i)}. \tag{76}$$

We have,

$$|y| = \frac{\pi_{\theta_t}(i) \cdot \left| r(i) - \pi_{\theta_t}^\top r \right|}{8 \cdot R_{\max}^2} \cdot \frac{\left| x - \pi_{\theta_t}^\top r \right|}{\pi_{\theta_t}(i)} \qquad \text{(by Eq. (6))} \tag{77}$$

$$\leq \frac{\left| r(i) - \pi_{\theta_t}^\top r \right|}{4 \cdot R_{\max}} \qquad \left( \left| x - \pi_{\theta_t}^\top r \right| \leq 2 \cdot R_{\max} \right) \tag{78}$$

$$\leq \frac{1}{2} \cdot \frac{\left| r(i) - \pi_{\theta_t}^\top r \right|}{\int_{-R_{\max}}^{R_{\max}} \left| x - \pi_{\theta_t}^\top r \right| \cdot P_i(x)\mu(dx)} \qquad \left( \int_{-R_{\max}}^{R_{\max}} \left| x - \pi_{\theta_t}^\top r \right| \cdot P_i(x)\mu(dx) \leq 2 \cdot R_{\max} \right) \tag{79}$$

$$= \epsilon. \tag{80}$$

Therefore, we have,

$$\int_{x \in \mathcal{X}_t^+} \frac{\exp\left\{ \eta \cdot \frac{x - \pi_{\theta_t}^\top r}{\pi_{\theta_t}(i)} \right\} - 1}{\exp\left\{ \eta \cdot \frac{x - \pi_{\theta_t}^\top r}{\pi_{\theta_t}(i)} \right\} + \frac{1 - \pi_{\theta_t}(i)}{\pi_{\theta_t}(i)}} \cdot P_i(x)\mu(dx) \tag{81}$$

$$\geq \int_{x \in \mathcal{X}_t^+} (1 - \epsilon) \cdot \pi_{\theta_t}(i) \cdot \eta \cdot \frac{x - \pi_{\theta_t}^\top r}{\pi_{\theta_t}(i)} \cdot P_i(x)\mu(dx) \qquad \text{(by Eq. (69))} \tag{82}$$

$$= \eta \cdot \int_{x \in \mathcal{X}_t^+} (1 - \epsilon) \cdot \left( x - \pi_{\theta_t}^\top r \right) \cdot P_i(x)\mu(dx). \tag{83}$$

According to Eq. (487) in Lemma 15, given any fixed $p \in (0, 1]$, and any fixed $\epsilon \in [0, 1]$, we have,

$$\frac{e^y - 1}{e^y + \frac{1 - p}{p}} \geq (1 + \epsilon) \cdot p \cdot y, \text{ for all } y \in [-\epsilon, 0]. \tag{84}$$

Using the same values of $p = \pi_{\theta_t}(i)$, $\epsilon$ in Eq. (70), and $y$ in Eq. (76), we have,

$$\int_{x \in \mathcal{X}_t^-} \frac{\exp\left\{\eta \cdot \frac{x - \pi_{\theta_t}^\top r}{\pi_{\theta_t}(i)}\right\} - 1}{\exp\left\{\eta \cdot \frac{x - \pi_{\theta_t}^\top r}{\pi_{\theta_t}(i)}\right\} + \frac{1 - \pi_{\theta_t}(i)}{\pi_{\theta_t}(i)}} \cdot P_i(x)\mu(dx) \tag{85}$$

$$\geq \int_{x \in \mathcal{X}_t^-} (1 + \epsilon) \cdot \pi_{\theta_t}(i) \cdot \eta \cdot \frac{x - \pi_{\theta_t}^\top r}{\pi_{\theta_t}(i)} \cdot P_i(x)\mu(dx) \qquad \text{(by Eq. (84))} \tag{86}$$

$$= \eta \cdot \int_{x \in \mathcal{X}_t^-} (1 + \epsilon) \cdot \left(x - \pi_{\theta_t}^\top r\right) \cdot P_i(x)\mu(dx). \tag{87}$$

Combining Eqs. (63), (81) and (85), we have,

$$\int_{-R_{\max}}^{R_{\max}} \frac{\left[\exp\left\{\eta \cdot \frac{x - \pi_{\theta_t}^\top r}{\pi_{\theta_t}(i)}\right\} - 1\right] \cdot \left(r(i) - \pi_{\theta_t}^\top r\right)}{\exp\left\{\eta \cdot \frac{x - \pi_{\theta_t}^\top r}{\pi_{\theta_t}(i)}\right\} + \frac{1 - \pi_{\theta_t}(i)}{\pi_{\theta_t}(i)}} \cdot P_i(x)\mu(dx) \tag{88}$$

$$\geq \left(r(i) - \pi_{\theta_t}^\top r\right) \cdot \eta \cdot \left[\int_{x \in \mathcal{X}_t^+} (1 - \epsilon) \cdot \left(x - \pi_{\theta_t}^\top r\right) \cdot P_i(x)\mu(dx)\right. \tag{89}$$

$$\left. + \int_{x \in \mathcal{X}_t^-} (1 + \epsilon) \cdot \left(x - \pi_{\theta_t}^\top r\right) \cdot P_i(x)\mu(dx)\right] \qquad \left(\text{since } r(i) - \pi_{\theta_t}^\top r > 0\right) \tag{90}$$

$$= \left(r(i) - \pi_{\theta_t}^\top r\right) \cdot \eta \cdot \left[\int_{-R_{\max}}^{R_{\max}} \left(x - \pi_{\theta_t}^\top r\right) \cdot P_i(x)\mu(dx) \qquad \text{(by Eq. (66))}\right. \tag{91}$$

$$\left. - \epsilon \cdot \left(\int_{x \in \mathcal{X}_t^+} \left(x - \pi_{\theta_t}^\top r\right) \cdot P_i(x)\mu(dx) - \int_{x \in \mathcal{X}_t^-} \left(x - \pi_{\theta_t}^\top r\right) \cdot P_i(x)\mu(dx)\right)\right] \tag{92}$$

$$= \left(r(i) - \pi_{\theta_t}^\top r\right) \cdot \eta \cdot \left[\left(r(i) - \pi_{\theta_t}^\top r\right) \qquad \text{(by Assumption 1)}\right. \tag{93}$$

$$\left. - \epsilon \cdot \int_{-R_{\max}}^{R_{\max}} \left|x - \pi_{\theta_t}^\top r\right| \cdot P_i(x)\mu(dx)\right] \qquad \text{(by Eq. (66))} \tag{94}$$

$$= \left(r(i) - \pi_{\theta_t}^\top r\right) \cdot \eta \cdot \left[\left(r(i) - \pi_{\theta_t}^\top r\right) - \frac{1}{2} \cdot \left(r(i) - \pi_{\theta_t}^\top r\right)\right] \qquad \text{(by Eq. (70))} \tag{95}$$

$$= \frac{\eta}{2} \cdot \left(r(i) - \pi_{\theta_t}^\top r\right)^2. \tag{96}$$

**Case (b).** $i \in [K]$ is a "bad" action at the current iteration, i.e., $r(i) - \pi_{\theta_t}^\top r < 0$.

According to Eq. (486) in Lemma 15, given any fixed $p \in (0, 1]$, and any fixed $\epsilon \in [0, 1]$, we have,

$$\frac{e^y - 1}{e^y + \frac{1-p}{p}} \leq (1 + \epsilon) \cdot p \cdot y, \text{ for all } y \in [0, \epsilon]. \tag{97}$$

Let $p = \pi_{\theta_t}(i) \in (0, 1]$ according to the softmax parameterization. Let

$$\epsilon = \frac{1}{2} \cdot \frac{-\left(r(i) - \pi_{\theta_t}^\top r\right)}{\sum_{m=1}^M P_i(m) \cdot \left|R_i(m) - \pi_{\theta_t}^\top r\right|} > 0. \tag{98}$$

We have $\epsilon \leq 1$ according to Eq. (71). Using the same value of $y$ in Eq. (76), we have,

$$\int_{x \in \mathcal{X}_t^+} \frac{\exp\left\{\eta \cdot \frac{x - \pi_{\theta_t}^\top r}{\pi_{\theta_t}(i)}\right\} - 1}{\exp\left\{\eta \cdot \frac{x - \pi_{\theta_t}^\top r}{\pi_{\theta_t}(i)}\right\} + \frac{1 - \pi_{\theta_t}(i)}{\pi_{\theta_t}(i)}} \cdot P_i(x)\mu(dx) \tag{99}$$

$$\leq \int_{x \in \mathcal{X}_t^+} (1 + \epsilon) \cdot \pi_{\theta_t}(i) \cdot \eta \cdot \frac{x - \pi_{\theta_t}^\top r}{\pi_{\theta_t}(i)} \cdot P_i(x)\mu(dx) \qquad \text{(by Eq. (97))} \tag{100}$$

$$= \eta \cdot \int_{x \in \mathcal{X}_t^+} (1 + \epsilon) \cdot \left(x - \pi_{\theta_t}^\top r\right) \cdot P_i(x)\mu(dx). \tag{101}$$

According to Eq. (487) in Lemma 15, given any fixed $p \in (0, 1]$, and any fixed $\epsilon \in [0, 1]$, we have,

$$\frac{e^y - 1}{e^y + \frac{1-p}{p}} \leq (1 - \epsilon) \cdot p \cdot y, \text{ for all } y \in [-\epsilon, 0]. \tag{102}$$

Using the same values of $p = \pi_{\theta_t}(i)$, $\epsilon$ in Eq. (98), and $y$ in Eq. (76), we have,

$$\int_{x \in \mathcal{X}_t^-} \frac{\exp\left\{\eta \cdot \frac{x - \pi_{\theta_t}^\top r}{\pi_{\theta_t}(i)}\right\} - 1}{\exp\left\{\eta \cdot \frac{x - \pi_{\theta_t}^\top r}{\pi_{\theta_t}(i)}\right\} + \frac{1 - \pi_{\theta_t}(i)}{\pi_{\theta_t}(i)}} \cdot P_i(x)\mu(dx) \tag{103}$$

$$\leq \int_{x \in \mathcal{X}_t^-} (1 - \epsilon) \cdot \pi_{\theta_t}(i) \cdot \eta \cdot \frac{x - \pi_{\theta_t}^\top r}{\pi_{\theta_t}(i)} \cdot P_i(x)\mu(dx) \qquad \text{(by Eq. (102))} \tag{104}$$

$$= \eta \cdot \int_{x \in \mathcal{X}_t^-} (1 - \epsilon) \cdot \left(x - \pi_{\theta_t}^\top r\right) \cdot P_i(x)\mu(dx). \tag{105}$$

Combining Eqs. (63), (99) and (103), we have,

$$\int_{-R_{\max}}^{R_{\max}} \frac{\left[\exp\left\{\eta \cdot \frac{x - \pi_{\theta_t}^\top r}{\pi_{\theta_t}(i)}\right\} - 1\right] \cdot \left(r(i) - \pi_{\theta_t}^\top r\right)}{\exp\left\{\eta \cdot \frac{x - \pi_{\theta_t}^\top r}{\pi_{\theta_t}(i)}\right\} + \frac{1 - \pi_{\theta_t}(i)}{\pi_{\theta_t}(i)}} \cdot P_i(x)\mu(dx) \tag{106}$$

$$\geq \left(r(i) - \pi_{\theta_t}^\top r\right) \cdot \eta \cdot \left[\int_{x \in \mathcal{X}_t^+} (1 + \epsilon) \cdot \left(x - \pi_{\theta_t}^\top r\right) \cdot P_i(x)\mu(dx)\right. \tag{107}$$

$$\left. + \int_{x \in \mathcal{X}_t^-} (1 - \epsilon) \cdot \left(x - \pi_{\theta_t}^\top r\right) \cdot P_i(x)\mu(dx)\right] \qquad \text{(since } r(i) - \pi_{\theta_t}^\top r < 0\text{)} \tag{108}$$

$$= \left(r(i) - \pi_{\theta_t}^\top r\right) \cdot \eta \cdot \left[\int_{-R_{\max}}^{R_{\max}} \left(x - \pi_{\theta_t}^\top r\right) \cdot P_i(x)\mu(dx) \qquad \text{(by Eq. (66))}\right. \tag{109}$$

$$\left. + \epsilon \cdot \left(\int_{x \in \mathcal{X}_t^+} \left(x - \pi_{\theta_t}^\top r\right) \cdot P_i(x)\mu(dx) - \int_{x \in \mathcal{X}_t^-} \left(x - \pi_{\theta_t}^\top r\right) \cdot P_i(x)\mu(dx)\right)\right] \tag{110}$$

$$= \left(r(i) - \pi_{\theta_t}^\top r\right) \cdot \eta \cdot \left[\left(r(i) - \pi_{\theta_t}^\top r\right) \qquad \text{(by Assumption 1)}\right. \tag{111}$$

$$\left. + \epsilon \cdot \int_{-R_{\max}}^{R_{\max}} \left|x - \pi_{\theta_t}^\top r\right| \cdot P_i(x)\mu(dx)\right] \qquad \text{(by Eq. (66))} \tag{112}$$

$$= \left(r(i) - \pi_{\theta_t}^\top r\right) \cdot \eta \cdot \left[\left(r(i) - \pi_{\theta_t}^\top r\right) - \frac{1}{2} \cdot \left(r(i) - \pi_{\theta_t}^\top r\right)\right] \qquad \text{(by Eq. (98))} \tag{113}$$

$$= \frac{\eta}{2} \cdot \left(r(i) - \pi_{\theta_t}^\top r\right)^2. \tag{114}$$

**Case (c).** $i \in [K]$ is an "indifferent" action at the current iteration, i.e., $r(i) - \pi_{\theta_t}^\top r = 0$.

According to Eq. (63), we have,

$$\int_{-R_{\max}}^{R_{\max}} \frac{\left[\exp\left\{\eta \cdot \frac{x - \pi_{\theta_t}^\top r}{\pi_{\theta_t}(i)}\right\} - 1\right] \cdot \left(r(i) - \pi_{\theta_t}^\top r\right)}{\exp\left\{\eta \cdot \frac{x - \pi_{\theta_t}^\top r}{\pi_{\theta_t}(i)}\right\} + \frac{1 - \pi_{\theta_t}(i)}{\pi_{\theta_t}(i)}} \cdot P_i(x)\mu(dx) \tag{115}$$

$$= 0 \geq \frac{\eta}{2} \cdot \left(r(i) - \pi_{\theta_t}^\top r\right)^2 . \qquad \text{(since } r(i) - \pi_{\theta_t}^\top r = 0\text{)} \tag{116}$$

Combining the three cases, i.e., Eqs. (88), (106) and (115), we have, for all action $i \in [K]$,

$$\int_{-R_{\max}}^{R_{\max}} \frac{\left[\exp\left\{\eta \cdot \frac{x - \pi_{\theta_t}^\top r}{\pi_{\theta_t}(i)}\right\} - 1\right] \cdot \left(r(i) - \pi_{\theta_t}^\top r\right)}{\exp\left\{\eta \cdot \frac{x - \pi_{\theta_t}^\top r}{\pi_{\theta_t}(i)}\right\} + \frac{1 - \pi_{\theta_t}(i)}{\pi_{\theta_t}(i)}} \cdot P_i(x)\mu(dx) \geq \frac{\eta}{2} \cdot \left(r(i) - \pi_{\theta_t}^\top r\right)^2 \tag{117}$$

$$= \frac{1}{2} \cdot \frac{\pi_{\theta_t}(i) \cdot \left|r(i) - \pi_{\theta_t}^\top r\right|}{8 \cdot R_{\max}^2} \cdot \left(r(i) - \pi_{\theta_t}^\top r\right)^2 . \qquad \text{(by Eq. (6))} \tag{118}$$

Combining Eqs. (63) and (117), we have,

$$\mathbb{E}_t[\pi_{\theta_{t+1}}^\top r] - \pi_{\theta_t}^\top r = \sum_{i=1}^{K} \pi_{\theta_t}(i) \cdot \int_{-R_{\max}}^{R_{\max}} \frac{\left[\exp\left\{\eta \cdot \frac{x - \pi_{\theta_t}^\top r}{\pi_{\theta_t}(i)}\right\} - 1\right] \cdot \left(r(i) - \pi_{\theta_t}^\top r\right)}{\exp\left\{\eta \cdot \frac{x - \pi_{\theta_t}^\top r}{\pi_{\theta_t}(i)}\right\} + \frac{1 - \pi_{\theta_t}(i)}{\pi_{\theta_t}(i)}} \cdot P_i(x)\mu(dx)$$

$$\tag{119}$$

$$\geq \frac{1}{16 \cdot R_{\max}^2} \cdot \sum_{i=1}^{K} \pi_{\theta_t}(i)^2 \cdot \left|r(i) - \pi_{\theta_t}^\top r\right|^3 \tag{120}$$

$$\geq \frac{1}{16 \cdot R_{\max}^2} \cdot \frac{\Delta}{K - 1} \cdot \pi_{\theta_t}(a^*)^2 \cdot \left(r(a^*) - \pi_{\theta_t}^\top r\right)^2 , \qquad \text{(by Lemma 16)} \tag{121}$$

thus finishing the proofs. $\qquad\square$

**Corollary 1.** The sequence $\{\pi_{\theta_t}^\top r\}_{t \geq 1}$ converges with probability one.

*Proof.* Setting $Y_t = r(a^*) - \pi_{\theta_t}^\top r$ we have $Y_t \in [0, 1]$. Define $\mathcal{F}_t$ as the $\sigma$-algebra generated by $a_1, x_1(a_1), a_2, x_2(a_2), \dots, a_{t-1}, x_{t-1}(a_{t-1})$. Note that $Y_t$ is $\mathcal{F}_t$-measurable since $\theta_t$ is a deterministic function of $a_1, x_1(a_1), \dots, a_{t-1}, x_{t-1}(a_{t-1})$. By Lemma 1, $\mathbb{E}[Y_{t+1}|\mathcal{F}_t] \leq Y_t$. Hence, the conditions of Doob's supermartingale theorem (Theorem 4) are satisfied and the result follows. $\quad\square$

**Lemma 2** (Non-vanishing stochastic NŁ coefficient / "automatic exploration"). Using Update 2 with the same settings as in Lemma 1, with arbitrary policy parameter initialization $\theta_1 \in \mathbb{R}^K$, we have,

$$c := \inf_{t \geq 1} \pi_{\theta_t}(a^*) > 0, \qquad \text{almost surely (a.s.).} \tag{122}$$

*Proof.* Since the claim is concerned with the policies underlying the parameter vectors and not the parameter vectors themselves, as noted after Update 2, without loss of generality, in the rest of the proof we assume that the parameter vector is updated according to Update 3 as follows,

$$\theta_{t+1}(a) \leftarrow \theta_t(a) + \eta \cdot \frac{\mathbb{I}\{a_t = a\}}{\pi_{\theta_t}(a)} \cdot \left(x_t(a) - \pi_{\theta_t}^\top r\right) . \tag{123}$$

Given $i \in [K]$, define the following set $\mathcal{P}(i)$ of "generalized one-hot policy",

$$\mathcal{A}(i) := \{j \in [K] : r(j) = r(i)\} , \tag{124}$$

$$\mathcal{P}(i) := \left\{\pi \in \Delta(K) : \sum_{j \in \mathcal{A}(i)} \pi(j) = 1\right\}. \tag{125}$$

We make the following two claims.

**Claim 1.** *Almost surely, $\pi_{\theta_t}$ approaches one "generalized one-hot policy", i.e., there exists (a possibly random) $i \in [K]$, such that $\sum_{j \in \mathcal{A}(i)} \pi_{\theta_t}(j) \to 1$ almost surely as $t \to \infty$.*

**Claim 2.** *Almost surely, $\pi_{\theta_t}$ cannot approach any "sub-optimal generalized one-hot policies", i.e., $i$ in the previous claim must be an optimal action.*

From Claim 2, it follows that $\sum_{j \in \mathcal{A}(a^*)} \pi_{\theta_t}(j) \to 1$ almost surely, as $t \to \infty$ and thus the policy sequence obtained almost surely convergences to a globally optimal policy $\pi^*$.

**Proof of Claim 1**.

According to Corollary 1, we have that for some (possibly random) $c \in [0, 1]$, almost surely,

$$\lim_{t \to \infty} \pi_{\theta_t}^\top r = c. \tag{126}$$

Thanks to $\pi_{\theta_t}^\top r \in [0, 1]$ and Eq. (11), $X_t = \pi_{\theta_t}^\top r$ ($t \geq 1$) satisfies the conditions of Corollary 3. Hence, by this result, almost surely,

$$\lim_{t \to \infty} \mathbb{E}_t[\pi_{\theta_{t+1}}^\top r] - \pi_{\theta_{t+1}}^\top r = 0, \tag{127}$$

which, combined with Eq. (126) also gives that $\lim_{t \to \infty} \mathbb{E}_t[\pi_{\theta_{t+1}}^\top r] = c$ almost surely. Hence,

$$\lim_{t \to \infty} \mathbb{E}_t[\pi_{\theta_{t+1}}^\top r] - \pi_{\theta_t}^\top r = c - c = 0, \qquad \text{a.s.} \tag{128}$$

According to Eq. (120) in the proof of Lemma 1, we have,

$$\mathbb{E}_t[\pi_{\theta_{t+1}}^\top r] - \pi_{\theta_t}^\top r \geq \frac{1}{16 \cdot R_{\max}^2} \cdot \sum_{i=1}^{K} \pi_{\theta_t}(i)^2 \cdot \left| r(i) - \pi_{\theta_t}^\top r \right|^3 \qquad \text{a.s.} \tag{129}$$

Combining Eqs. (128) and (129), we have, with probability 1,

$$\lim_{t \to \infty} \sum_{i=1}^{K} \pi_{\theta_t}(i)^2 \cdot \left| r(i) - \pi_{\theta_t}^\top r \right|^3 = 0, \tag{130}$$

which implies that, for all $i \in [K]$, almost surely,

$$\lim_{t \to \infty} \pi_{\theta_t}(i)^2 \cdot \left| r(i) - \pi_{\theta_t}^\top r \right|^3 = 0. \tag{131}$$

We claim that $c$, the almost sure limit of $\pi_{\theta_t}^\top r$, is such that almost surely, for some (possibly random) $i \in [K]$, $c = r(i)$ almost surely. We prove this by contradiction. Let $\mathcal{E}_i = \{c = r(i)\}$. Hence, our goal is to show that $\mathbb{P}(\cup_i \mathcal{E}_i) = 1$. Clearly, this follows from $\mathbb{P}(\cap_i \mathcal{E}_i^c) = 0$, hence, we prove this. On $\mathcal{E}_i^c$, since $\lim_{t \to \infty} \pi_{\theta_t}^\top r \neq r(i)$, we also have

$$\lim_{t \to \infty} \left| r(i) - \pi_{\theta_t}^\top r \right|^3 > 0, \qquad \text{almost surely on } \mathcal{E}_i^c. \tag{132}$$

This, together with Eq. (131) gives that almost surely on $\mathcal{E}_i^c$,

$$\lim_{t \to \infty} \pi_{\theta_t}(i)^2 = 0. \tag{133}$$

Hence, on $\cap_i \mathcal{E}_i^c$, almost surely, for all $i \in [K]$, $\lim_{t \to \infty} \pi_{\theta_t}(i)^2 = 0$. This contradicts with that $\sum_i \pi_{\theta_t}(i) = 1$ holds for all $t \geq 1$, and hence we must have that $\mathbb{P}(\cap_i \mathcal{E}_i^c) = 0$, finishing the proof that $\mathbb{P}(\cup_i \mathcal{E}_i) = 1$.

Now, let $i \in [K]$ be the (possibly random) index of the action for which $c = r(i)$ almost surely. Recall that $\mathcal{A}(i)$ contains all actions $j$ with $r(j) = r(i)$ (cf. Eq. (124)). Clearly, it holds that for all $j \in \mathcal{A}(i)$,

$$\lim_{t \to \infty} \pi_{\theta_t}^\top r = r(j), \qquad \text{a.s.,} \tag{134}$$

and we have, for all $k \notin \mathcal{A}(i)$,

$$\lim_{t \to \infty} \left| r(k) - \pi_{\theta_t}^\top r \right|^3 > 0, \qquad \text{a.s.,} \tag{135}$$

which implies that,

$$\lim_{t\to\infty} \sum_{k\notin\mathcal{A}(i)} \pi_{\theta_t}(k)^2 = 0, \qquad \text{a.s.} \tag{136}$$

Therefore, we have,

$$\lim_{t\to\infty} \sum_{j\in\mathcal{A}(i)} \pi_{\theta_t}(j) = 1, \qquad \text{a.s.,} \tag{137}$$

which means $\pi_{\theta_t}$ a.s. approaches the "generalized one-hot policy" $\mathcal{P}(i)$ in Eq. (125) as $t\to\infty$, finishing the proof of the first claim.

**Proof of Claim 2**. Recall that this claim stated that $\lim_{t\to\infty} \sum_{j\in\mathcal{A}(a^*)} \pi_{\theta_t}(j) = 1$. The brief sketch of the proof is as follows: By Claim 1, there exists a (possibly random) $i \in [K]$ such that $\sum_{j\in\mathcal{A}(i)} \pi_{\theta_t}(j) \to 1$ almost surely, as $t\to\infty$. If $i = a^*$ almost surely, Claim 2 follows. Hence, it suffices to consider the event that $\{i \neq a^*\}$ and show that this event has zero probability mass. Hence, in the rest of the proof we assume that we are on the event when $i \neq a^*$.

Since $i \neq a^*$, there exists at least one "good" action $a^+ \in [K]$ such that $r(a^+) > r(i)$. The two cases are as follows.

**2a)** All "good" actions are sampled finitely many times as $t\to\infty$.

**2b)** At least one "good" action is sampled infinitely many times as $t\to\infty$.

In both cases, we show that $\sum_{j\in\mathcal{A}(i)} \exp\{\theta_t(j)\} < \infty$ as $t\to\infty$ (but for different reasons), which is a contradiction with the assumption of $\sum_{j\in\mathcal{A}(i)} \pi_{\theta_t}(j) \to 1$ as $t\to\infty$, given that a "good" action's parameter is almost surely lower bounded. Hence, $i \neq a^*$ almost surely does not happen, which means that almost surely $i = a^*$.

Let us now turn to the details of the proof. We start with some useful extra notation. For each action $a \in [K]$, for $t \geq 2$, we have the following decomposition,

$$\theta_t(a) = \underbrace{\theta_t(a) - \mathbb{E}_{t-1}[\theta_t(a)]}_{W_t(a)} + \underbrace{\mathbb{E}_{t-1}[\theta_t(a)] - \theta_{t-1}(a)}_{P_{t-1}(a)} + \theta_{t-1}(a), \tag{138}$$

while we also have,

$$\theta_1(a) = \underbrace{\theta_1(a) - \mathbb{E}[\theta_1(a)]}_{W_1(a)} + \mathbb{E}[\theta_1(a)], \tag{139}$$

where $\mathbb{E}[\theta_1(a)]$ accounts for possible randomness in initialization of $\theta_1$.

Define the following notations,

$$Z_t(a) := W_1(a) + \cdots + W_t(a), \qquad \text{("cumulative noise")} \tag{140}$$
$$W_t(a) := \theta_t(a) - \mathbb{E}_{t-1}[\theta_t(a)], \qquad \text{("noise")} \tag{141}$$
$$P_t(a) := \mathbb{E}_t[\theta_{t+1}(a)] - \theta_t(a). \qquad \text{("progress")} \tag{142}$$

Recursing Eq. (138) gives,

$$\theta_t(a) = \mathbb{E}[\theta_1(a)] + Z_t(a) + \underbrace{P_1(a) + \cdots + P_{t-1}(a)}_{\text{"cumulative progress"}}. \tag{143}$$

We have that $\mathbb{E}_t[W_{t+1}(a)] = 0$, for $t = 0, 1, \ldots$. Let

$$I_t(a) = \begin{cases} 1, & \text{if } a_t = a, \\ 0, & \text{otherwise}. \end{cases} \tag{144}$$

The update rule (cf. Eq. (123)) is,

$$\theta_{t+1}(a) = \theta_t(a) + \eta \cdot \frac{I_t(a)}{\pi_{\theta_t}(a)} \cdot \left(x_t(a) - \pi_{\theta_t}^\top r\right), \tag{145}$$

where $a_t \sim \pi_{\theta_t}(\cdot)$, and $x_t(a) \sim P_a$. Let $\mathcal{F}_t$ be the $\sigma$-algebra generated by $a_1, x_1(a_1), \cdots, a_{t-1}$, $x_{t-1}(a_{t-1}), a_t$:

$$\mathcal{F}_t = \sigma(\{a_1, x_1(a_1), \cdots, a_{t-1}, x_{t-1}(a_{t-1}), a_t\}). \tag{146}$$

Note that $\theta_t, I_t$ are $\mathcal{F}_t$-measurable and $\hat{x}_t$ is $\mathcal{F}_{t+1}$-measurable for all $t \geq 1$. Let $\mathbb{E}_t$ denote the conditional expectation with respect to $\mathcal{F}_t$: $\mathbb{E}_t[X] = \mathbb{E}[X|\mathcal{F}_t]$.

Using the above notations, we have,

$$W_{t+1}(a) = \theta_{t+1}(a) - \mathbb{E}_t[\theta_{t+1}(a)] \tag{147}$$

$$= \cancel{\theta_t(a)} + \eta \cdot \frac{I_t(a)}{\pi_{\theta_t}(a)} \cdot \left(x_t(a) - \cancel{\pi_{\theta_t}^\top r}\right) - \mathbb{E}_t\left[\cancel{\theta_t(a)} + \eta \cdot \frac{I_t(a)}{\pi_{\theta_t}(a)} \cdot \left(x_t(a) - \cancel{\pi_{\theta_t}^\top r}\right)\right] \tag{148}$$

$$= \eta \cdot \frac{I_t(a)}{\pi_{\theta_t}(a)} \cdot (x_t(a) - r(a)), \tag{149}$$

which implies that,

$$Z_t(a) = W_1(a) + \cdots + W_t(a) \tag{150}$$

$$= \sum_{s=1}^{t-1} \eta \cdot \frac{I_s(a)}{\pi_{\theta_s}(a)} \cdot (x_s(a) - r(a)). \tag{151}$$

We also have,

$$P_t(a) = \mathbb{E}_t[\theta_{t+1}(a)] - \theta_t(a) \tag{152}$$

$$= \mathbb{E}_t\left[\cancel{\theta_t(a)} + \eta \cdot \frac{I_t(a)}{\pi_{\theta_t}(a)} \cdot \left(x_t(a) - \pi_{\theta_t}^\top r\right)\right] - \cancel{\theta_t(a)} \tag{153}$$

$$= \eta \cdot \frac{I_t(a)}{\pi_{\theta_t}(a)} \cdot \left(r(a) - \pi_{\theta_t}^\top r\right). \tag{154}$$

Using the learning rate of Eq. (6),

$$\eta = \frac{\pi_{\theta_t}(a_t) \cdot \left|r(a_t) - \pi_{\theta_t}^\top r\right|}{8 \cdot R_{\max}^2}, \tag{155}$$

we have,

$$W_{t+1}(a) = \frac{\pi_{\theta_t}(a_t) \cdot \left|r(a_t) - \pi_{\theta_t}^\top r\right|}{8 \cdot R_{\max}^2} \cdot \frac{I_t(a)}{\pi_{\theta_t}(a)} \cdot (x_t(a) - r(a)) \qquad \text{(by Eq. (147))} \tag{156}$$

$$= \frac{I_t(a)}{8 \cdot R_{\max}^2} \cdot \left|r(a) - \pi_{\theta_t}^\top r\right| \cdot (x_t(a) - r(a)) \tag{157}$$

$$\in \left[-\frac{1}{8 \cdot R_{\max}}, \frac{1}{8 \cdot R_{\max}}\right]. \tag{158}$$

Similarly, we have,

$$P_t(a) = \frac{I_t(a)}{8 \cdot R_{\max}^2} \cdot \left|r(a) - \pi_{\theta_t}^\top r\right| \cdot \left(r(a) - \pi_{\theta_t}^\top r\right), \tag{159}$$

and

$$Z_t(a) = \sum_{s=1}^{t-1} \frac{I_s(a)}{8 \cdot R_{\max}^2} \cdot \left|r(a) - \pi_{\theta_s}^\top r\right| \cdot (x_s(a) - r(a)). \tag{160}$$

Define the following notations,

$$N_t(a) := \sum_{s=1}^{t} I_s(a), \tag{161}$$

$$N_\infty(a) := \sum_{s=1}^{\infty} I_s(a), \tag{162}$$

$$N_{p:q}(a) := \sum_{s=p}^{q} I_s(a). \tag{163}$$

Recall that $i$ is the index of the (random) action $I \in [K]$ with

$$\lim_{t \to \infty} \sum_{j \in \mathcal{A}(I)} \pi_{\theta_t}(j) = 1, \qquad \text{a.s.} \tag{164}$$

As noted earlier we consider the event $\{I \neq a^*\}$, where $a^*$ is the index of an optimal action and we will show that this event has zero probability. Since $\{I \neq a^*\} = \cup_{i \in [K]} \{I = i, i \neq a^*\}$, it suffices to show that for any fixed $i \in [K]$ index with $r(i) < r(a^*)$, $\{I = i, i \neq a^*\}$ has zero probability. Hence, in what follows we fix such a suboptimal action's index $i \in [K]$ and consider the event $\{I = i, i \neq a^*\}$.

Partition the action set $[K]$ into three parts using $r(i)$ as follows,

$$\mathcal{A}(i) := \{j \in [K] : r(j) = r(i)\}, \qquad \text{(from Eq. (124))} \tag{165}$$
$$\mathcal{A}^+(i) := \{a^+ \in [K] : r(a^+) > r(i)\}, \tag{166}$$
$$\mathcal{A}^-(i) := \{a^- \in [K] : r(a^-) < r(i)\}. \tag{167}$$

Because $i$ was the index of a sub-optimal action, we have $\mathcal{A}^+(i) \neq \emptyset$. According to Eq. (164), on $\{I = i\} \supset \{I = i, i \neq a^*\}$, we have $\pi_{\theta_t}^\top r \to r(i)$ as $t \to \infty$ because

$$\left| r(i) - \pi_{\theta_t}^\top r \right| = \left| \sum_{k \notin \mathcal{A}(i)} \pi_{\theta_t}(k) \cdot (r(i) - r(k)) \right| \tag{168}$$

$$\leq \sum_{k \notin \mathcal{A}(i)} \pi_{\theta_t}(k) \cdot |r(i) - r(k)| \tag{169}$$

$$\leq 1 - \sum_{j \in \mathcal{A}(i)} \pi_{\theta_t}(j). \qquad \left( r \in [0,1]^K \right) \tag{170}$$

Therefore, there exists $\tau \geq 1$ such that almost surely on $\{I = i, i \neq a^*\}$ $\tau < \infty$ while we also have

$$r(a^+) - c' \geq \pi_{\theta_t}^\top r \geq r(a^-) + c', \qquad \text{for all } t \geq \tau, \tag{171}$$

for all $a^+ \in \mathcal{A}^+(i)$, $a^- \in \mathcal{A}^-(i)$, where $c' > 0$.

Now, take any $a^- \in \mathcal{A}^-(i)$. According to Lemma 9, we have, almost surely on $\{I = i, i \neq a^*\}$,

$$c_1 := \sup_{t \geq 1} \theta_t(a^-) < \infty. \tag{172}$$

**First case. 2a).** Consider the event,

$$\mathcal{E}_0 := \bigcap_{a^+ \in \mathcal{A}^+(i)} \underbrace{\left\{ N_\infty(a^+) < \infty \right\}}_{\mathcal{E}_0(a^+)}, \tag{173}$$

i.e., any "good" action $a^+ \in \mathcal{A}^+(i)$ has finitely many updates as $t \to \infty$. Pick $a^+ \in \mathcal{A}^+(i)$, such that $\mathbb{P}(N_\infty(a^+) < \infty) > 0$. According to the extended Borel-Cantelli lemma (Lemma 14), we have, almost surely,

$$\left\{ \sum_{t \geq 1} \pi_{\theta_t}(a^+) = \infty \right\} = \left\{ N_\infty(a^+) = \infty \right\}. \tag{174}$$

Hence, taking complements, we have,

$$\left\{ \sum_{t \geq 1} \pi_{\theta_t}(a^+) < \infty \right\} = \left\{ N_\infty(a^+) < \infty \right\} \tag{175}$$

also holds almost surely.

On event $\mathcal{E}_0(a^+)$, we also have,

$$c_2 := \inf_{t \geq 1} \theta_t(a^+) > -\infty, \tag{176}$$

$$c_3 := \sup_{t \geq 1} \theta_t(a^+) < \infty, \tag{177}$$

which is because on this event the parameter corresponding to $a^+$ receives finitely many updates and each update is bounded, i.e., for any $a \in [K]$,

$$\left|\theta_{t+1}(a) - \theta_t(a)\right| = \eta \cdot \frac{I_t(a)}{\pi_{\theta_t}(a)} \cdot \left|x_t(a) - \pi_{\theta_t}^\top r\right| \qquad \text{(by Eq. (145))} \tag{178}$$

$$= \frac{\pi_{\theta_t}(a_t) \cdot \left|r(a_t) - \pi_{\theta_t}^\top r\right|}{8 \cdot R_{\max}^2} \cdot \frac{I_t(a)}{\pi_{\theta_t}(a)} \cdot \left|x_t(a) - \pi_{\theta_t}^\top r\right| \qquad \text{(by Eq. (155))} \tag{179}$$

$$= \frac{I_t(a)}{8 \cdot R_{\max}^2} \cdot \left|r(a) - \pi_{\theta_t}^\top r\right| \cdot \left|x_t(a) - \pi_{\theta_t}^\top r\right| \leq \frac{1}{8 \cdot R_{\max}}. \tag{180}$$

Define

$$q_t = \sum_{a^+ \in \mathcal{A}^+(i)} \pi_{\theta_t}(a^+). \tag{181}$$

On event $\mathcal{E}' := \mathcal{E}_0 \cap \{I = i, i \neq a^*\}$, and by the softmax parameterization, we have,

$$q_t = \frac{\sum_{a^+ \in \mathcal{A}^+(i)} e^{\theta_t(a^+)}}{\sum_{j \in \mathcal{A}(i)} e^{\theta_t(j)} + \sum_{a^+ \in \mathcal{A}^+(i)} e^{\theta_t(a^+)} + \sum_{a^- \in \mathcal{A}^-(i)} e^{\theta_t(a^-)}} \tag{182}$$

$$\geq \frac{\sum_{a^+ \in \mathcal{A}^+(i)} e^{c_2}}{\sum_{j \in \mathcal{A}(i)} e^{\theta_t(j)} + \sum_{a^+ \in \mathcal{A}^+(i)} e^{c_2} + \sum_{a^- \in \mathcal{A}^-(i)} e^{\theta_t(a^-)}} \qquad \text{(by Eq. (176))} \tag{183}$$

$$\geq \frac{\sum_{a^+ \in \mathcal{A}^+(i)} e^{c_2}}{\sum_{j \in \mathcal{A}(i)} e^{\theta_t(j)} + \sum_{a^+ \in \mathcal{A}^+(i)} e^{c_2} + \sum_{a^- \in \mathcal{A}^-(i)} e^{c_1}} \qquad \text{(by Eq. (172))} \tag{184}$$

$$= \frac{e^{c_2} \cdot |\mathcal{A}^+(i)|}{\sum_{j \in \mathcal{A}(i)} e^{\theta_t(j)} + e^{c_2} \cdot |\mathcal{A}^+(i)| + e^{c_1} \cdot |\mathcal{A}^-(i)|}. \tag{185}$$

Next, we have,

$$1 - \sum_{j \in \mathcal{A}(i)} \pi_{\theta_t}(j) = \frac{\sum_{a^+ \in \mathcal{A}^+(i)} e^{\theta_t(a^+)} + \sum_{a^- \in \mathcal{A}^-(i)} e^{\theta_t(a^-)}}{\sum_{j \in \mathcal{A}(i)} e^{\theta_t(j)} + \sum_{a^+ \in \mathcal{A}^+(i)} e^{\theta_t(a^+)} + \sum_{a^- \in \mathcal{A}^-(i)} e^{\theta_t(a^-)}} \tag{186}$$

$$\leq \frac{\sum_{a^+ \in \mathcal{A}^+(i)} e^{c_3} + \sum_{a^- \in \mathcal{A}^-(i)} e^{c_1}}{\sum_{j \in \mathcal{A}(i)} e^{\theta_t(j)} + \sum_{a^+ \in \mathcal{A}^+(i)} e^{c_3} + \sum_{a^- \in \mathcal{A}^-(i)} e^{c_1}} \qquad \text{(by Eqs. (172) and (177))} \tag{187}$$

$$= \frac{e^{c_3} \cdot |\mathcal{A}^+(i)| + e^{c_1} \cdot |\mathcal{A}^-(i)|}{\sum_{j \in \mathcal{A}(i)} e^{\theta_t(j)} + e^{c_2} \cdot |\mathcal{A}^+(i)| + e^{c_1} \cdot |\mathcal{A}^-(i)| + (e^{c_3} - e^{c_2}) \cdot |\mathcal{A}^+(i)|} \tag{188}$$

$$\leq \frac{e^{c_3} \cdot |\mathcal{A}^+(i)| + e^{c_1} \cdot |\mathcal{A}^-(i)|}{\frac{e^{c_2}}{q_t} \cdot |\mathcal{A}^+(i)| + (e^{c_3} - e^{c_2}) \cdot |\mathcal{A}^+(i)|} \qquad \text{(by Eq. (182))} \tag{189}$$

$$= \frac{e^{c_3} \cdot |\mathcal{A}^+(i)| + e^{c_1} \cdot |\mathcal{A}^-(i)|}{e^{c_2} \cdot |\mathcal{A}^+(i)| + (e^{c_3} - e^{c_2}) \cdot |\mathcal{A}^+(i)| \cdot q_t} \cdot q_t \tag{190}$$

$$\leq \frac{e^{c_3} \cdot |\mathcal{A}^+(i)| + e^{c_1} \cdot |\mathcal{A}^-(i)|}{e^{c_2} \cdot |\mathcal{A}^+(i)|} \cdot q_t. \qquad \text{(because } q_t > 0) \tag{191}$$

Denote $C' := \frac{e^{c_3} \cdot |\mathcal{A}^+(i)| + e^{c_1} \cdot |\mathcal{A}^-(i)|}{e^{c_2} \cdot |\mathcal{A}^+(i)|}$. We have,

$$\left|r(i) - \pi_{\theta_t}^\top r\right| \leq 1 - \sum_{j \in \mathcal{A}(i)} \pi_{\theta_t}(j) \qquad \left(r \in [0,1]^K\right) \qquad \text{(by Eq. (168))} \tag{192}$$

$$\leq C' \cdot q_t. \qquad \text{(by Eq. (191))} \tag{193}$$

Take any $j \in \mathcal{A}(i)$, according to Eq. (143), we have,

$$\theta_t(j) = \mathbb{E}[\theta_1(j)] + Z_t(j) + \sum_{s=1}^{t-1} P_s(j). \tag{194}$$

According to Eq. (159), we have,

$$P_s(j) = \frac{I_s(j)}{8 \cdot R_{\max}^2} \cdot \left| r(j) - \pi_{\theta_s}^\top r \right| \cdot \left( r(j) - \pi_{\theta_s}^\top r \right). \tag{195}$$

Therefore, for all $s \geq 1$,

$$|P_s(j)| \leq \frac{1}{8 \cdot R_{\max}^2} \cdot \left( r(i) - \pi_{\theta_s}^\top r \right)^2 \qquad (j \in \mathcal{A}(i),\ r(j) = r(i)) \tag{196}$$

$$\leq \frac{C'}{8 \cdot R_{\max}^2} \cdot q_s^2 \qquad \text{(by Eq. (192))} \tag{197}$$

$$\leq \frac{C'}{8 \cdot R_{\max}^2} \cdot q_s. \qquad (q_s \in (0,1)) \tag{198}$$

For any $j \in \mathcal{A}(i)$, we have,

$$S_t^2(j) := \sum_{s=1}^{t} \left( r(j) - \pi_{\theta_s}^\top r \right)^2 \cdot I_s(j) \tag{199}$$

$$\leq \sum_{s=1}^{t} \left( r(j) - \pi_{\theta_s}^\top r \right)^2 \tag{200}$$

$$\leq \sum_{s=1}^{t} q_s^2 \qquad \text{(by Eq. (192))} \tag{201}$$

$$\leq \sum_{s=1}^{t} q_s \qquad (q_s \in [0,1]) \tag{202}$$

$$=: Q_t. \tag{203}$$

Fix $\delta \in [0,1]$. According to Lemma 11, $\exists\, \mathcal{E}_\delta$ with $\mathbb{P}(\mathcal{E}_\delta) \geq 1 - \delta$, and on $\mathcal{E}_\delta$, for all $t \geq 1$,

$$|Z_t(j)| \leq \frac{1}{8R_{\max}} \cdot \sqrt{(1 + S_t^2(j)) \cdot \left( 1 + 2\log\left( \frac{(1 + S_t^2(j))^{\frac{1}{2}}}{\delta} \right) \right)}. \tag{204}$$

Then, on $\mathcal{E}' \cap \mathcal{E}_\delta$, Eq. (203) holds and also,

$$\sum_{s=1}^{t-1} P_s(j) \leq \frac{C'}{8 \cdot R_{\max}^2} \cdot Q_t. \qquad \text{(by Eq. (198))} \tag{205}$$

According to Eqs. (194), (204) and (205), we have, on $\mathcal{E}' \cap \mathcal{E}_\delta$,

$$\theta_t(j) \leq \mathbb{E}[\theta_1(j)] + \frac{1}{8R_{\max}} \cdot \sqrt{(1 + Q_t) \cdot \left( 1 + 2\log\left( \frac{(1 + Q_t)^{\frac{1}{2}}}{\delta} \right) \right)} + \frac{C'}{8R_{\max}^2} \cdot Q_t \tag{206}$$

$$\leq \mathbb{E}[\theta_1(j)] + \frac{1}{8R_{\max}} \cdot \sqrt{(1 + Q) \cdot \left( 1 + 2\log\left( \frac{(1 + Q)^{\frac{1}{2}}}{\delta} \right) \right)} + \frac{C'}{8R_{\max}^2} \cdot Q, \tag{207}$$

where $Q = \lim_{t \to \infty} Q_t$ and the inequality follows because $(Q_t)$ is increasing. Note that on $\mathcal{E}'$, $Q$ is finite almost surely, according to Eqs. (175), (181) and (203).

Now take any $\omega \in \mathcal{E}'$. Because $\mathbb{P}(\mathcal{E}' \setminus (\mathcal{E}' \cap \mathcal{E}_\delta)) \leq \mathbb{P}(\Omega \setminus \mathcal{E}_\delta) \leq \delta \to 0$ as $\delta \to 0$, we have that $\mathbb{P}$-almost surely for all $\omega \in \mathcal{E}'$ there exists $\delta > 0$ such that $\omega \in \mathcal{E}' \cap \mathcal{E}_\delta$ while Eq. (207) also holds for this $\delta$. Take such a $\delta$. By Eq. (207),

$$\limsup_{t \to \infty} \theta_t(j)(\omega) < \infty. \tag{208}$$

Hence, almost surely on $\mathcal{E}'$,

$$c_4 := \limsup_{t \to \infty} \theta_t(j) < \infty. \tag{209}$$

Therefore, we have, almost surely on $\mathcal{E}'$,

$$\sum_{j\in\mathcal{A}(i)} \pi_{\theta_t}(j) = \frac{\sum_{j\in\mathcal{A}(i)} e^{\theta_t(j)}}{\sum_{j\in\mathcal{A}(i)} e^{\theta_t(j)} + \sum_{a^+\in\mathcal{A}^+(i)} e^{\theta_t(a^+)} + \sum_{a^-\in\mathcal{A}^-(i)} e^{\theta_t(a^-)}} \tag{210}$$

$$\leq \frac{\sum_{j\in\mathcal{A}(i)} e^{\theta_t(j)}}{\sum_{j\in\mathcal{A}(i)} e^{\theta_t(j)} + \sum_{a^+\in\mathcal{A}^+(i)} e^{\theta_t(a^+)}} \qquad \left(e^{\theta_t(a^-)} > 0\right) \tag{211}$$

$$\leq \frac{\sum_{j\in\mathcal{A}(i)} e^{\theta_t(j)}}{\sum_{j\in\mathcal{A}(i)} e^{\theta_t(j)} + e^{c_2}\cdot|\mathcal{A}^+(i)|} \qquad \text{(by Eq. (176))} \tag{212}$$

$$\leq \frac{e^{c_4}\cdot|\mathcal{A}(i)|}{e^{c_4}\cdot|\mathcal{A}(i)| + e^{c_2}\cdot|\mathcal{A}^+(i)|} \qquad \text{(by Eq. (209))} \tag{213}$$

$$\not\to 1, \tag{214}$$

which is a contradiction with the assumption of Eq. (164), showing that $\mathbb{P}(\mathcal{E}') = 0$.

**Second case. 2b).** Consider the complement $\mathcal{E}_0^c$ of $\mathcal{E}_0$, where $\mathcal{E}_0$ is by Eq. (173). $\mathcal{E}_0^c$ indicates the event for at least one "good" action $a^+\in\mathcal{A}^+(i)$ has infinitely many updates as $t\to\infty$.

We now show that also $\mathbb{P}(\mathcal{E}'') = 0$ where $\mathcal{E}'' = \mathcal{E}_0^c\cap\{I=i, i\neq a^*\} = (\cup_{a^+\in\mathcal{A}(i)}\{N_\infty(a^+) = \infty\})\cap\{I=i, i\neq a^*\}$. It suffices to show that for any $a^+\in\mathcal{A}^+(i)$, $\mathbb{P}(\{N_\infty(a^+)=\infty\})\cap\{I=i, i\neq a^*\}) = 0$.

Thus, fix an arbitrary $a^+\in\mathcal{A}^+(i)$ and let

$$\mathcal{E}' := \mathcal{E}_\infty(a^+)\cap\{I=i, i\neq a^*\},$$

where for $a\in[K]$, $\mathcal{E}_\infty(a) = \{N_\infty(a) = \infty\}$. With this notation, the goal is to show that $\mathbb{P}(\mathcal{E}') = 0$.[2] Since $\mathcal{E}'\subset\mathcal{E}_\infty(a^+)$, the statement follows if $\mathbb{P}(\mathcal{E}_\infty(a^+)) = 0$. Hence, assume that $\mathbb{P}(\mathcal{E}_\infty(a^+)) > 0$.

Fix $\delta\in[0,1]$. According to Corollary 2, there exists an event $\mathcal{E}_\delta$ such that $\mathbb{P}(\mathcal{E}_\delta)\geq 1-\delta$, and on $\mathcal{E}_\delta$, for all $t\geq 1$,

$$\left|Z_t(a^+)\right| \leq \frac{1}{8R_{\max}}\cdot\sqrt{(1+N_t(a^+))\cdot\left(1+2\log\left(\frac{(1+N_t(a^+))^{\frac{1}{2}}}{\delta}\right)\right)}. \tag{215}$$

Using a similar calculation as in the proof of Lemma 9, we have, on $\mathcal{E}_\delta\cap\mathcal{E}_\infty(a^+)$ that

$$\theta_t(a^+) \geq \mathbb{E}[\theta_1(a^+)] - \frac{1}{8R_{\max}}\cdot\sqrt{(1+N_t(a^+))\cdot\left(1+2\log\left(\frac{(1+N_t(a^+))^{\frac{1}{2}}}{\delta}\right)\right)} \tag{216}$$

$$+ \frac{c}{8\cdot R_{\max}^2}\cdot\underbrace{N_{t-1}(a^+)}_{\to\infty} - \frac{c}{8\cdot R_{\max}^2}\cdot(\tau-1) + P_1(a^+) + \cdots + P_{\tau-1}(a^+). \tag{217}$$

On $\mathcal{E}_\infty(a^+)\cap\mathcal{E}_\delta$, $N_{t-1}(a^+)\to\infty$ as $t\to\infty$, we have $\theta_t(a^+)\to\infty$ as $t\to\infty$.

Since $\mathbb{P}(\mathcal{E}_\infty(a^+)\setminus(\mathcal{E}_\infty(a^+)\cap\mathcal{E}_\delta))\to 0$ as $\delta\to 0$, we have, almost surely on $\mathcal{E}_\infty(a^+)$,

$$\lim_{t\to\infty}\theta_t(a^+) = \infty, \tag{218}$$

which implies that there exists $\tau\geq 1$ such that on $\mathcal{E}'(=\mathcal{E}_\infty(a^+)\cap\{I=i, i\neq a^*\})$ we have almost surely that $\tau < +\infty$ while we also have that for all $t\geq\tau$,

$$\sum_{a^-\in\mathcal{A}^-(i)} \frac{r(i) - r(a^-)}{\exp\{\theta_t(a^+) - c_1\}} < \frac{r(a^+) - r(i)}{2}. \tag{219}$$

---

[2]Here, $\mathcal{E}'$ is redefined to minimize clutter; the previous definition is not used in this part of the proof.

Hence, on $\mathcal{E}'$, for $t \geq \tau$, almost surely,

$$\pi_{\theta_t}^\top r = \sum_{j \in \mathcal{A}(i)} \pi_{\theta_t}(j) \cdot r(i) + \sum_{a^- \in \mathcal{A}^-(i)} \pi_{\theta_t}(a^-) \cdot r(a^-) + \sum_{\tilde{a}^+ \in \mathcal{A}^+(i)} \pi_{\theta_t}(\tilde{a}^+) \cdot r(\tilde{a}^+) \tag{220}$$

$$= r(i) - \sum_{a^- \in \mathcal{A}^-(i)} \pi_{\theta_t}(a^-) \cdot \left(r(i) - r(a^-)\right) + \sum_{\tilde{a}^+ \in \mathcal{A}^+(i)} \pi_{\theta_t}(\tilde{a}^+) \cdot \left(r(\tilde{a}^+) - r(i)\right) \tag{221}$$

$$\geq r(i) - \sum_{a^- \in \mathcal{A}^-(i)} \pi_{\theta_t}(a^-) \cdot \left(r(i) - r(a^-)\right) + \pi_{\theta_t}(a^+) \cdot \left(r(a^+) - r(i)\right) \qquad \left(r(\tilde{a}^+) - r(i) > 0, \text{ Eq. (166)}\right) \tag{222}$$

$$= r(i) + \pi_{\theta_t}(a^+) \cdot \left[\left(r(a^+) - r(i)\right) - \sum_{a^- \in \mathcal{A}^-(i)} \frac{\pi_{\theta_t}(a^-)}{\pi_{\theta_t}(a^+)} \cdot \left(r(i) - r(a^-)\right)\right] \tag{223}$$

$$= r(i) + \pi_{\theta_t}(a^+) \cdot \left[\left(r(a^+) - r(i)\right) - \sum_{a^- \in \mathcal{A}^-(i)} \frac{r(i) - r(a^-)}{\exp\{\theta_t(a^+) - \theta_t(a^-)\}}\right] \tag{224}$$

$$\geq r(i) + \pi_{\theta_t}(a^+) \cdot \left[\left(r(a^+) - r(i)\right) - \sum_{a^- \in \mathcal{A}^-(i)} \frac{r(i) - r(a^-)}{\exp\{\theta_t(a^+) - c_1\}}\right] \qquad \text{(by Eq. (172))} \tag{225}$$

$$> r(i) + \frac{r(a^+) - r(i)}{2} \cdot \pi_{\theta_t}(a^+). \qquad \text{(by Eq. (219))} \tag{226}$$

Therefore, on $\mathcal{E}'$, for all $s \geq \tau$, for any $j \in \mathcal{A}(i)$, almost surely,

$$P_s(j) = \frac{I_s(j)}{8 \cdot R_{\max}^2} \cdot \left|r(j) - \pi_{\theta_s}^\top r\right| \cdot \left(r(j) - \pi_{\theta_s}^\top r\right) \qquad \text{(by Eq. (159))} \tag{227}$$

$$= -\frac{I_s(j)}{8 \cdot R_{\max}^2} \cdot \left(r(j) - \pi_{\theta_s}^\top r\right)^2. \qquad \text{(by Eq. (220), } r(i) - \pi_{\theta_s}^\top r < 0) \tag{228}$$

From now on assume that $\mathcal{E}'$ holds. Therefore, we have, for all $t \geq \tau$,

$$\sum_{s=1}^{t-1} P_s(j) = \sum_{s=1}^{\tau-1} P_s(j) + \sum_{s=\tau}^{t} P_s(j) - P_t(j) \tag{229}$$

$$= \sum_{s=1}^{\tau-1} P_s(j) - \frac{1}{8 \cdot R_{\max}^2} \cdot \sum_{s=\tau}^{t} \left(r(j) - \pi_{\theta_s}^\top r\right)^2 \cdot I_s(j) - P_t(j) \qquad \text{(by Eq. (227))} \tag{230}$$

$$= \sum_{s=1}^{\tau-1} P_s(j) - \frac{1}{8 \cdot R_{\max}^2} \cdot \left[S_t^2(j) - \sum_{s=1}^{\tau-1} \left(r(j) - \pi_{\theta_s}^\top r\right)^2 \cdot I_s(j)\right] - P_t(j) \tag{231}$$

$$= -\frac{1}{8 \cdot R_{\max}^2} \cdot S_t^2(j) + \sum_{s=1}^{\tau-1} \left[P_s(j) + \frac{\left(r(j) - \pi_{\theta_s}^\top r\right)^2 \cdot I_s(j)}{8 \cdot R_{\max}^2} \cdot\right] - P_t(j) \tag{232}$$

$$\leq -\frac{1}{8 \cdot R_{\max}^2} \cdot S_t^2(j) + \frac{\tau - 1}{4 \cdot R_{\max}^2} + \frac{1}{8 \cdot R_{\max}^2}, \qquad \left(|P_t(j)| \leq \frac{1}{8 \cdot R_{\max}^2}, \text{ Eq. (227)}\right) \tag{233}$$

where $S_t^2(j) = \sum_{s=1}^{t} \left(r(j) - \pi_{\theta_s}^\top r\right)^2 \cdot I_s(j)$. According to Lemma 11, for any $\delta \in [0, 1]$, there exist an event $\mathcal{E}_\delta$ such that $\mathbb{P}(\mathcal{E}_\delta) \geq 1 - \delta$ and on $\mathcal{E}_\delta \cap \mathcal{E}'$, we have,

$$\theta_t(j) \leq \mathbb{E}[\theta_1(j)] + Z_t(j) + \sum_{s=1}^{t-1} P_s(j) \qquad \text{(by Eq. (143))} \tag{234}$$

$$\leq \mathbb{E}[\theta_1(j)] + \frac{1}{8R_{\max}} \cdot \sqrt{(1 + S_t^2(j)) \cdot \left(1 + 2\log\left(\frac{(1 + S_t^2(j))^{\frac{1}{2}}}{\delta}\right)\right)} \tag{235}$$

$$- \frac{1}{8 \cdot R_{\max}^2} \cdot \left(1 + S_t^2(j)\right) + \frac{\tau}{4 \cdot R_{\max}^2}. \tag{236}$$

Note that,

$$M(\delta) := \sup_{s \geq 0} \frac{1}{8R_{\max}} \cdot \sqrt{(1+s) \cdot \left(1 + 2\log\left(\frac{(1+s)^{\frac{1}{2}}}{\delta}\right)\right)} - \frac{1}{8 \cdot R_{\max}^2} \cdot (1+s) \qquad (237)$$

$$< \infty. \qquad (238)$$

Therefore, on $\mathcal{E}' \cap \mathcal{E}_\delta$ for $t \geq \tau$ we have,

$$\theta_t(j) \leq \mathbb{E}[\theta_1(j)] + M(\delta) + \frac{\tau}{4 \cdot R_{\max}^2} . \qquad (239)$$

Since $\mathbb{P}(\mathcal{E}_\delta^c) \to 0$ as $\delta \to 0$, with an argument parallel to that used in the proof of the first part (cf. the argument around Eq. (208)), we get that there exists a random constant $c_5(j)$ such that almost surely on $\mathcal{E}'$, $c_5(j) < \infty$ and $\sup_{t \geq \tau} \theta_t(j) \leq c_5(j)$. Define $c_5 := \max_{j \in \mathcal{A}(i)} c_5(j)$. Then, almost surely on $\mathcal{E}'$, $c_5 < \infty$ and

$$\sup_{t \geq \tau} \max_{j \in \mathcal{A}(i)} \theta_t(j) \leq c_5 . \qquad (240)$$

By Eq. (218), there exists $\tau' \geq 1$, such that almost surely on $\mathcal{E}'$, $\tau' < \infty$ while we also have

$$\inf_{t \geq \tau'} \theta_t(a^+) \geq 0, \qquad (241)$$

for all $t \geq \tau'$. Hence, on $\mathcal{E}'$, almost surely for all $t \geq \max(\tau, \tau')$,

$$\sum_{j \in \mathcal{A}(i)} \pi_{\theta_t}(j) = \frac{\sum_{j \in \mathcal{A}(i)} e^{\theta_t(j)}}{\sum_{j \in \mathcal{A}(i)} e^{\theta_t(j)} + \sum_{\tilde{a}^+ \in \mathcal{A}^+(i)} e^{\theta_t(\tilde{a}^+)} + \sum_{a^- \in \mathcal{A}^-(i)} e^{\theta_t(a^-)}} \qquad (242)$$

$$\leq \frac{\sum_{j \in \mathcal{A}(i)} e^{\theta_t(j)}}{\sum_{j \in \mathcal{A}(i)} e^{\theta_t(j)} + e^{\theta_t(a^+)}} \qquad \left(e^{\theta_t(k)} > 0 \text{ for any } k \in [K]\right) \qquad (243)$$

$$\leq \frac{\sum_{j \in \mathcal{A}(i)} e^{\theta_t(j)}}{\sum_{j \in \mathcal{A}(i)} e^{\theta_t(j)} + 1} \qquad \text{(by Eq. (241) )} \qquad (244)$$

$$\leq \frac{e^{c_5} \cdot |\mathcal{A}(i)|}{e^{c_5} \cdot |\mathcal{A}(i)| + 1} \qquad \text{(by Eq. (239))} \qquad (245)$$

$$\not\to 1 . \qquad (246)$$

Hence, $\mathbb{P}(\mathcal{E}') = 0$, finishing the proof. $\qquad \square$

Let us now turn to the proof of the results that were used in the above proof.

**Lemma 9.** *Let $I$ be as in Eq. (164), let $i$ be a sub-optimal action, and let $\tau$ be as in Eq. (171), Then, on $\{I = i, i \neq a^*\}$, for any action $a^- \in \mathcal{A}^-(i)$ (using Update 2) we have, almost surely,*

$$c_1 := \sup_{t \geq 1} \theta_t(a^-) < \infty. \qquad (247)$$

*Proof.* According to Eq. (159), we have, for all $t \geq \tau$,

$$P_t(a^-) = \frac{I_t(a^-)}{8 \cdot R_{\max}^2} \cdot \left|r(a^-) - \pi_{\theta_t}^\top r\right| \cdot \left(r(a^-) - \pi_{\theta_t}^\top r\right) \qquad (248)$$

$$\leq -c \cdot \frac{I_t(a^-)}{8 \cdot R_{\max}^2}, \qquad \text{(by Eq. (171))} \qquad (249)$$

which implies that,

$$\theta_t(a^-) = \mathbb{E}[\theta_1(a^-)] + Z_t(a^-) + P_1(a^-) + \cdots + P_{\tau-1}(a^-) \qquad \text{(by Eq. (143))} \qquad (250)$$

$$+ P_\tau(a^-) + \cdots + P_{t-1}(a^-) \qquad (251)$$

$$\leq \mathbb{E}[\theta_1(a^-)] + Z_t(a^-) + P_1(a^-) + \cdots + P_{\tau-1}(a^-) \qquad (252)$$

$$- \frac{c}{8 \cdot R_{\max}^2} \cdot \left(I_\tau(a^-) + \cdots + I_{t-1}(a^-)\right) \qquad \text{(by Eq. (248))} \qquad (253)$$

$$= \mathbb{E}[\theta_1(a^-)] + Z_t(a^-) + P_1(a^-) + \cdots + P_{\tau-1}(a^-) \qquad (254)$$

$$- \frac{c}{8 \cdot R_{\max}^2} \cdot N_{\tau:t-1}(a^-) \qquad \text{(Eq. (163))} \qquad (255)$$

Denote $\mathcal{E}_\infty(a) := \{N_\infty(a) = \infty\}$, where $N_\infty(a)$ is defined in Eq. (162).

Fix $\delta \in [0, 1]$. Take $\mathcal{E}_\delta$ from Corollary 2. Consider on event $\mathcal{E}_\infty(a^-) \cap \mathcal{E}_\delta$, we have,

$$\theta_t(a^-) \le \mathbb{E}[\theta_1(a^-)] + \frac{1}{8R_{\max}} \cdot \sqrt{(1 + N_t(a)) \cdot \left(1 + 2\log\left(\frac{(1 + N_t(a))^{\frac{1}{2}}}{\delta}\right)\right)} \tag{256}$$

$$- \frac{c}{8 \cdot R_{\max}^2} \cdot N_{\tau:t-1}(a^-) + P_1(a^-) + \cdots + P_{\tau-1}(a^-). \tag{257}$$

Note that,

$$N_{\tau:t-1}(a^-) = N_{t-1}(a^-) - N_{1:\tau-1}(a^-) \qquad \text{(Eqs. (161) and (163))} \tag{258}$$

$$\ge N_{t-1}(a^-) - (\tau - 1). \tag{259}$$

We have,

$$\theta_t(a^-) \le \mathbb{E}[\theta_1(a^-)] + \frac{1}{8R_{\max}} \cdot \sqrt{(1 + N_t(a)) \cdot \left(1 + 2\log\left(\frac{(1 + N_t(a))^{\frac{1}{2}}}{\delta}\right)\right)} \tag{260}$$

$$- \frac{c}{8 \cdot R_{\max}^2} \cdot \underbrace{N_{t-1}(a^-)}_{\to \infty} + \frac{c}{8 \cdot R_{\max}^2} \cdot (\tau - 1) + P_1(a^-) + \cdots + P_{\tau-1}(a^-). \tag{261}$$

On $\mathcal{E}_\infty(a^-) \cap \mathcal{E}_\delta$, $N_{t-1}(a^-) \to \infty$ as $t \to \infty$, we have $\theta_t(a^-) \to -\infty$ as $t \to \infty$.

Since $\mathbb{P}(\mathcal{E}_\infty(a^-) \setminus (\mathcal{E}_\infty(a^-) \cap \mathcal{E}_\delta)) \to 0$ as $\delta \to 0$, we have, almost surely on $\mathcal{E}_\infty(a^-)$,

$$\lim_{t \to \infty} \theta_t(a^-) = -\infty, \tag{262}$$

which implies that on $\mathcal{E}_\infty(a^-)$, we have $\sup_{t \ge 1} \theta_t(a^-) < \infty$.

On the other hand, on $(\mathcal{E}_\infty(a^-))^c$, we have $\sup_{t \ge 1} \theta_t(a^-) < \infty$ by construction (finitely many updates of $a^-$ as $t \to \infty$, and each update is bounded according to Eq. (178)).

Therefore, we have $\sup_{t \ge 1} \theta_t(a^-) < \infty$ almost surely. $\qquad \square$

**Lemma 10** (Lemma 6 in [1]). *Let $X_t = \sum_{s=1}^t I_s \cdot \eta_s$, and $N_t = \sum_{s=1}^t I_s$. Assume $\eta_t$ is conditionally $\sigma$-sub-Gaussian, and $I_t$ is $\mathcal{F}_t$-measurable. Then, for all $\delta \in [0, 1]$, with probability $1 - \delta$, for all $t \ge 1$,*

$$|X_t| \le \sigma \cdot \sqrt{(1 + N_t) \cdot \left(1 + 2\log\left(\frac{(1 + N_t)^{\frac{1}{2}}}{\delta}\right)\right)}. \tag{263}$$

**Corollary 2.** *For all $a \in [K]$, $\forall \delta$, $\exists\, \mathcal{E}_\delta$ with $\mathbb{P}(\mathcal{E}_\delta) \ge 1 - \delta$, such that on $\mathcal{E}_\delta$, for all $t \ge 1$,*

$$|Z_t(a)| \le \frac{1}{8R_{\max}} \cdot \sqrt{(1 + N_t(a)) \cdot \left(1 + 2\log\left(\frac{(1 + N_t(a))^{\frac{1}{2}}}{\delta}\right)\right)}. \tag{264}$$

**Lemma 11.** *For all $a \in [K]$, $\forall \delta \in [0, 1]$, $\exists\, \mathcal{E}_\delta$ with $\mathbb{P}(\mathcal{E}_\delta) \ge 1 - \delta$, such that on $\mathcal{E}_\delta$, for all $t \ge 1$,*

$$|Z_t(a)| \le \frac{1}{8R_{\max}} \cdot \sqrt{(1 + S_t^2(a)) \cdot \left(1 + 2\log\left(\frac{(1 + S_t^2(a))^{\frac{1}{2}}}{\delta}\right)\right)}, \tag{265}$$

*where $S_t^2(a) := \sum_{s=1}^t \left(r(a) - \pi_{\theta_s}^\top r\right)^2 \cdot I_s(a)$.*

*Proof.* Follow the steps of the proof of Lemma 6 in [1]. $\qquad \square$

**Theorem 1** (Almost sure global convergence rate). Using Update 2 with on-policy sampling $a_t \sim \pi_{\theta_t}(\cdot)$, the IS estimator in Definition 1, $\eta$ in Eq. (6), and any initialization $\theta_1 \in \mathbb{R}^K$, we have,

$$\mathbb{E}[(\pi^* - \pi_{\theta_t})^\top r] \leq \frac{16 \cdot R_{\max}^2}{\Delta \cdot \mathbb{E}[c^2]} \cdot \frac{K-1}{t}, \qquad \text{and} \tag{266}$$

$$\limsup_{t \geq 1} \left\{ \frac{\Delta \cdot c^2}{16 \cdot R_{\max}^2} \cdot \frac{t}{K-1} \cdot (\pi^* - \pi_{\theta_t})^\top r \right\} < \infty, \qquad \text{a.s.,} \tag{267}$$

where $\mathbb{E}_t[\cdot]$ denotes $\mathbb{E}_t[\cdot | \mathcal{F}_t]$, and $\mathcal{F}_t$ is the $\sigma$-algebra generated by $a_1, x_1(a_1), \ldots, a_{t-1}, x_{t-1}(a_{t-1})$, $\pi^* := \arg\max_{\pi \in \Delta(K)} \pi^\top r$ is the optimal policy, $R_{\max}$ is the sampled reward range from Assumption 1, $\Delta := r(a^*) - \max_{a \neq a^*} r(a)$ is the reward gap of $r$, and $c > 0$ is from Lemma 2.

*Proof.* **First part.** According to Lemma 1, we have,

$$\mathbb{E}_t[\pi_{\theta_{t+1}}^\top r] - \pi_{\theta_t}^\top r \geq \frac{1}{16 \cdot R_{\max}^2} \cdot \frac{\Delta}{K-1} \cdot \pi_{\theta_t}(a^*)^2 \cdot \left( r(a^*) - \pi_{\theta_t}^\top r \right)^2 \tag{268}$$

$$\geq \frac{1}{16 \cdot R_{\max}^2} \cdot \frac{\Delta}{K-1} \cdot \inf_{t \geq 1} \pi_{\theta_t}(a^*)^2 \cdot \left( r(a^*) - \pi_{\theta_t}^\top r \right)^2 \tag{269}$$

$$= \frac{1}{16 \cdot R_{\max}^2} \cdot \frac{\Delta}{K-1} \cdot c^2 \cdot \left( r(a^*) - \pi_{\theta_t}^\top r \right)^2, \tag{270}$$

where $c := \inf_{t \geq 1} \pi_{\theta_t}(a^*) > 0$ is according to Lemma 2. Let $\delta(\theta_t) := (\pi^* - \pi_{\theta_t})^\top r$ denote the sub-optimality gap. We have,

$$\delta(\theta_t) - \mathbb{E}_t[\delta(\theta_{t+1})] = (\pi^* - \pi_{\theta_t})^\top r - \mathbb{E}_t\left[ \left( \pi^* - \pi_{\theta_{t+1}} \right)^\top r \right] \tag{271}$$

$$= (\pi^* - \pi_{\theta_t})^\top r - \left( \pi^* - \mathbb{E}_t[\pi_{\theta_{t+1}}] \right)^\top r \tag{272}$$

$$= \mathbb{E}_t[\pi_{\theta_{t+1}}^\top r] - \pi_{\theta_t}^\top r \tag{273}$$

$$\geq \frac{1}{16 \cdot R_{\max}^2} \cdot \frac{\Delta}{K-1} \cdot c^2 \cdot \left( r(a^*) - \pi_{\theta_t}^\top r \right)^2 \tag{274}$$

$$= \frac{1}{16 \cdot R_{\max}^2} \cdot \frac{\Delta}{K-1} \cdot c^2 \cdot \delta(\theta_t)^2. \tag{275}$$

Taking expectation, we have,

$$\mathbb{E}\left[ \delta(\theta_t) \right] - \mathbb{E}\left[ \delta(\theta_{t+1}) \right] \geq \frac{\Delta \cdot \mathbb{E}[c^2]}{16 \cdot R_{\max}^2} \cdot \frac{1}{K-1} \cdot \mathbb{E}\left[ \delta(\theta_t)^2 \right] \tag{276}$$

$$\geq \frac{\Delta \cdot \mathbb{E}[c^2]}{16 \cdot R_{\max}^2} \cdot \frac{1}{K-1} \cdot \left( \mathbb{E}\left[ \delta(\theta_t) \right] \right)^2. \qquad \text{(by Jensen's inequality)} \tag{277}$$

Therefore, we have, for all $t \geq 1$,

$$\frac{1}{\mathbb{E}\left[ \delta(\theta_t) \right]} = \frac{1}{\mathbb{E}\left[ \delta(\theta_1) \right]} + \sum_{s=1}^{t-1} \left[ \frac{1}{\mathbb{E}\left[ \delta(\theta_{s+1}) \right]} - \frac{1}{\mathbb{E}\left[ \delta(\theta_s) \right]} \right] \tag{278}$$

$$= \frac{1}{\mathbb{E}\left[ \delta(\theta_1) \right]} + \sum_{s=1}^{t-1} \frac{1}{\mathbb{E}\left[ \delta(\theta_{s+1}) \right] \cdot \mathbb{E}\left[ \delta(\theta_s) \right]} \cdot \left( \mathbb{E}\left[ \delta(\theta_s) \right] - \mathbb{E}\left[ \delta(\theta_{s+1}) \right] \right) \tag{279}$$

$$\geq \frac{1}{\mathbb{E}\left[ \delta(\theta_1) \right]} + \sum_{s=1}^{t-1} \frac{1}{\mathbb{E}\left[ \delta(\theta_{s+1}) \right] \cdot \mathbb{E}\left[ \delta(\theta_s) \right]} \cdot \frac{\Delta \cdot \mathbb{E}[c^2]}{16 \cdot R_{\max}^2} \cdot \frac{1}{K-1} \cdot \left( \mathbb{E}\left[ \delta(\theta_s) \right] \right)^2 \tag{280}$$

$$\geq \frac{1}{\mathbb{E}\left[ \delta(\theta_1) \right]} + \sum_{s=1}^{t-1} \frac{\Delta \cdot \mathbb{E}[c^2]}{16 \cdot R_{\max}^2} \cdot \frac{1}{K-1} \qquad \left( \mathbb{E}\left[ \delta(\theta_s) \right] \geq \mathbb{E}\left[ \delta(\theta_{s+1}) \right] > 0 \right) \tag{281}$$

$$= \frac{1}{\mathbb{E}\left[ \delta(\theta_1) \right]} + \frac{\Delta \cdot \mathbb{E}[c^2]}{16 \cdot R_{\max}^2} \cdot \frac{1}{K-1} \cdot (t-1) \tag{282}$$

$$\geq \frac{\Delta \cdot \mathbb{E}[c^2]}{16 \cdot R_{\max}^2} \cdot \frac{t}{K-1}, \qquad \left( \mathbb{E}\left[ \delta(\theta_1) \right] \leq 1 < \frac{16 \cdot R_{\max}^2}{\Delta \cdot \mathbb{E}[c^2]} \cdot (K-1) \right) \tag{283}$$

which implies that, for all $t \geq 1$,

$$\mathbb{E}[(\pi^* - \pi_{\theta_t})^\top r] = \mathbb{E}\left[\delta(\theta_t)\right] \leq \frac{16 \cdot R_{\max}^2}{\Delta \cdot \mathbb{E}[c^2]} \cdot \frac{K-1}{t}. \tag{284}$$

**Second part.** The result follows from the following Lemma 12 by choosing $X_t = (\pi^* - \pi_{\theta_t})^\top r$ and $f(t) = \frac{\Delta \cdot \mathbb{E}[c^2]}{16 \cdot R_{\max}^2} \cdot \frac{t}{K-1}$. $\square$

**Lemma 12.** *Let $(X_t)_{t \geq 1}$ be a sequence of random variables such that $X_t \in [0,1]$, $X_t \to 0$ almost surely and for $t \geq 1$, $\mathbb{E}[X_t] \leq \frac{1}{f(t)}$ with $f(t) \to \infty$ as $t \to \infty$. Then $\limsup_{t \to \infty} f(t)X_t < \infty$ almost surely.*

*Proof of Lemma 12.* Let $\mathcal{E}$ be the event when $\limsup_{t \to \infty} \{f(t) \cdot X_t\} = \infty$. It suffices to show that $\mathbb{P}(\mathcal{E}) = 0$. Consider the event $\mathcal{E}$. On this event, there exists a strictly increasing sequence $\{t_k\}_{k \geq 1}$, such that $f(t_k) \cdot X_{t_k} \to \infty$ as $k \to \infty$. Since $X_t \geq 0$, we have,

$$\mathbb{E}[X_{t_k}] \geq \mathbb{E}[X_{t_k} \cdot \mathbb{I}_\mathcal{E}]. \tag{285}$$

Then we have,

$$1 \geq \lim_{k \to \infty} \mathbb{E}[f(t_k) \cdot X_{t_k}] \tag{286}$$

$$\geq \lim_{k \to \infty} \mathbb{E}[f(t_k) \cdot X_{t_k} \cdot \mathbb{I}_\mathcal{E}] \tag{287}$$

$$= \liminf_{k \to \infty} \mathbb{E}[f(t_k) \cdot X_{t_k} \cdot \mathbb{I}_\mathcal{E}] \tag{288}$$

$$\geq \mathbb{E}[(\liminf_{k \to \infty} f(t_k) \cdot X_{t_k}) \cdot \mathbb{I}_\mathcal{E}]. \qquad \text{(Fatou's lemma)} \tag{289}$$

If $\mathbb{P}(\mathcal{E}) > 0$, the right-hand side above is $\infty$, which would imply that $\infty \leq 1$. Hence, we must have $\mathbb{P}(\mathcal{E}) = 0$. $\square$

# B Proofs for General MDPs

**Lemma 3** (Stochastic NŁ). *Using Algorithm 1 with constant $\eta > 0$, we have, for all $t \geq 1$,*

$$V^{\pi_{\theta_{t+1}}}(s_0) - V^{\pi_{\theta_t}}(s_0) \geq 0, \qquad \text{a.s.,} \qquad \forall s_0 \in \mathcal{S}, \qquad \text{and} \tag{290}$$

$$\mathbb{E}_t[V^{\pi_{\theta_{t+1}}}(\mu)] - V^{\pi_{\theta_t}}(\mu) \geq \frac{\eta \cdot (1-\gamma)^4}{1+\eta} \cdot \min_s \mu(s) \cdot \left\| \frac{d_\mu^{\pi^*}}{\mu} \right\|_\infty^{-1} \cdot \frac{\min_s \pi_{\theta_t}(a^*(s)|s)^2}{S} \cdot \left(V^{\pi^*}(\mu) - V^{\pi_{\theta_t}}(\mu)\right)^2, \tag{291}$$

*where $\mathbb{E}_t[\cdot]$ is on randomness from state sampling $s_t \sim d_\mu^{\pi_{\theta_t}}(\cdot)$ and on-policy sampling $a_t \sim \pi_{\theta_t}(\cdot|s_t)$, and $a^*(s)$ is the action selected by the optimal policy $\pi^*$ under state $s$.*

*Proof.* For all $t \geq 1$, for any state action pair $(s, i) \in \mathcal{S} \times \mathcal{A}$, denote

$$\left[V^{\pi_{\theta_{t+1}}}(s_0) \mid s_t = s, a_t = i\right] \tag{292}$$

as the the value of $V^{\pi_{\theta_{t+1}}}(s_0)$ given the sampled state action pair $(s_t, a_t) = (s, i)$.

Given $s_t = s$, for all $s' \neq s$, we have, for all $a \in \mathcal{A}$,

$$\pi_{\theta_{t+1}}(a|s') = \frac{\exp\{\theta_{t+1}(s', a)\}}{\sum_{a' \in \mathcal{A}} \exp\{\theta_{t+1}(s', a')\}} \tag{293}$$

$$= \frac{\exp\{\theta_t(s', a)\}}{\sum_{a' \in \mathcal{A}} \exp\{\theta_t(s', a')\}} \qquad (s' \neq s_t, \text{ Algorithm 1}) \tag{294}$$

$$= \pi_{\theta_t}(a|s'). \tag{295}$$

According to the performance difference Lemma 17, we have,

$$\left[ V^{\pi_{\theta_{t+1}}}(s_0) \mid s_t = s, a_t = i \right] - V^{\pi_{\theta_t}}(s_0) \tag{296}$$

$$= \frac{1}{1-\gamma} \cdot \sum_{s' \in \mathcal{S}} d_{s_0}^{\pi_{\theta_{t+1}}}(s') \cdot \sum_a \left( \pi_{\theta_{t+1}}(a|s') - \pi_{\theta_t}(a|s') \right) \cdot Q^{\pi_{\theta_t}}(s', a) \tag{297}$$

$$= \frac{1}{1-\gamma} \cdot d_{s_0}^{\pi_{\theta_{t+1}}}(s) \cdot \sum_a \left( \pi_{\theta_{t+1}}(a|s) - \pi_{\theta_t}(a|s) \right) \cdot Q^{\pi_{\theta_t}}(s, a). \qquad \text{(by Eq. (293))} \tag{298}$$

Note that, in the above equation $d_{s_0}^{\pi_{\theta_{t+1}}}(s) = \left[ d_{s_0}^{\pi_{\theta_{t+1}}}(s) \mid s_t = s, a_t = i \right]$, which means that for each sampled state action pair $(s_t, a_t) = (s, i)$, we have a different $\pi_{\theta_{t+1}}$ and thus $d_{s_0}^{\pi_{\theta_{t+1}}}$. According to the update in Algorithm 1, we have,

$$\left[ \sum_a \pi_{\theta_{t+1}}(a|s) \cdot Q^{\pi_{\theta_t}}(s, a) \mid s_t = s, a_t = i \right] \tag{299}$$

$$= \frac{\exp\left\{ \theta_t(s, i) + \eta \cdot \frac{Q^{\pi_{\theta_t}}(s,i) - V^{\pi_{\theta_t}}(s)}{\pi_{\theta_t}(i|s)} \right\} \cdot Q^{\pi_{\theta_t}}(s, i) + \sum_{j \neq i} \exp\{\theta_t(s, j)\} \cdot Q^{\pi_{\theta_t}}(s, j)}{\exp\left\{ \theta_t(s, i) + \eta \cdot \frac{Q^{\pi_{\theta_t}}(s,i) - V^{\pi_{\theta_t}}(s)}{\pi_{\theta_t}(i|s)} \right\} + \sum_{j \neq i} \exp\{\theta_t(s, j)\}}, \tag{300}$$

which is similar to Eq. (33). Therefore, by algebra we have,

$$\left[ \sum_a \left( \pi_{\theta_{t+1}}(a|s) - \pi_{\theta_t}(a|s) \right) \cdot Q^{\pi_{\theta_t}}(s, a) \mid s_t = s, a_t = i \right] \tag{301}$$

$$= \frac{\left[ \exp\left\{ \eta \cdot \frac{Q^{\pi_{\theta_t}}(s,i) - V^{\pi_{\theta_t}}(s)}{\pi_{\theta_t}(i|s)} \right\} - 1 \right] \cdot \left( Q^{\pi_{\theta_t}}(s, i) - V^{\pi_{\theta_t}}(s) \right)}{\exp\left\{ \eta \cdot \frac{Q^{\pi_{\theta_t}}(s,i) - V^{\pi_{\theta_t}}(s)}{\pi_{\theta_t}(i|s)} \right\} + \frac{1 - \pi_{\theta_t}(i|s)}{\pi_{\theta_t}(i|s)}} \geq 0, \tag{302}$$

where the last inequality is from $(e^{c \cdot y} - 1) \cdot y \geq 0$ for all $y \in \mathbb{R}$ with $c := \frac{\eta}{\pi_{\theta_t}(i|s)} > 0$.

Combining Eqs. (296) and (301), we have,

$$\left[ V^{\pi_{\theta_{t+1}}}(s_0) \mid s_t = s, a_t = i \right] - V^{\pi_{\theta_t}}(s_0) \tag{303}$$

$$= \frac{d_{s_0}^{\pi_{\theta_{t+1}}}(s)}{1-\gamma} \cdot \frac{\left[ \exp\left\{ \eta \cdot \frac{Q^{\pi_{\theta_t}}(s,i) - V^{\pi_{\theta_t}}(s)}{\pi_{\theta_t}(i|s)} \right\} - 1 \right] \cdot \left( Q^{\pi_{\theta_t}}(s, i) - V^{\pi_{\theta_t}}(s) \right)}{\exp\left\{ \eta \cdot \frac{Q^{\pi_{\theta_t}}(s,i) - V^{\pi_{\theta_t}}(s)}{\pi_{\theta_t}(i|s)} \right\} + \frac{1 - \pi_{\theta_t}(i|s)}{\pi_{\theta_t}(i|s)}} \geq 0, \tag{304}$$

which proves Eq. (290) because of $(s, i) \in \mathcal{S} \times \mathcal{A}$ is arbitrary.

For all $t \geq 1$, given current policy $\pi_{\theta_t}$, the value function of next policy $V^{\pi_{\theta_{t+1}}}(\mu)$ is a random variable, and the randomness is from state sampling $s_t \sim d_\mu^{\pi_{\theta_t}}(\cdot)$ and on-policy sampling $a_t \sim \pi_{\theta_t}(\cdot|s_t)$. According to Eq. (303), the expected progress after one update is,

$$\mathbb{E}_t[V^{\pi_{\theta_{t+1}}}(\mu)] - V^{\pi_{\theta_t}}(\mu) = \sum_s d_\mu^{\pi_{\theta_t}}(s) \sum_i \pi_{\theta_t}(i|s) \cdot \left( \left[ V^{\pi_{\theta_{t+1}}}(\mu) \mid s_t = s, a_t = i \right] - V^{\pi_{\theta_t}}(\mu) \right) \tag{305}$$

$$= \sum_s d_\mu^{\pi_{\theta_t}}(s) \sum_i \pi_{\theta_t}(i|s) \cdot \frac{d_\mu^{\pi_{\theta_{t+1}}}(s)}{1-\gamma} \cdot \frac{\left[ \exp\left\{ \eta \cdot \frac{Q^{\pi_{\theta_t}}(s,i) - V^{\pi_{\theta_t}}(s)}{\pi_{\theta_t}(i|s)} \right\} - 1 \right] \cdot \left( Q^{\pi_{\theta_t}}(s, i) - V^{\pi_{\theta_t}}(s) \right)}{\exp\left\{ \eta \cdot \frac{Q^{\pi_{\theta_t}}(s,i) - V^{\pi_{\theta_t}}(s)}{\pi_{\theta_t}(i|s)} \right\} + \frac{1 - \pi_{\theta_t}(i|s)}{\pi_{\theta_t}(i|s)}} \tag{306}$$

$$\geq \sum_s \mu(s) \cdot d_\mu^{\pi_{\theta_t}}(s) \sum_i \pi_{\theta_t}(i|s) \cdot \frac{\left[ \exp\left\{ \eta \cdot \frac{Q^{\pi_{\theta_t}}(s,i) - V^{\pi_{\theta_t}}(s)}{\pi_{\theta_t}(i|s)} \right\} - 1 \right] \cdot \left( Q^{\pi_{\theta_t}}(s, i) - V^{\pi_{\theta_t}}(s) \right)}{\exp\left\{ \eta \cdot \frac{Q^{\pi_{\theta_t}}(s,i) - V^{\pi_{\theta_t}}(s)}{\pi_{\theta_t}(i|s)} \right\} + \frac{1 - \pi_{\theta_t}(i|s)}{\pi_{\theta_t}(i|s)}}, \tag{307}$$

where the inequality is because of Eq. (301) and for any $\theta$ and $\mu$,

$$d_\mu^{\pi_\theta}(s) = \mathop{\mathbb{E}}_{s_0 \sim \mu}\left[d_\mu^{\pi_\theta}(s)\right] \tag{308}$$

$$= \mathop{\mathbb{E}}_{s_0 \sim \mu}\left[(1-\gamma) \cdot \sum_{t=0}^{\infty} \gamma^t \cdot \mathbb{P}(s_t = s \mid s_0, \pi_\theta, \mathcal{P})\right] \tag{309}$$

$$\geq (1-\gamma) \cdot \mathop{\mathbb{E}}_{s_0 \sim \mu}\left[\mathbb{P}(s_0 = s|s_0)\right] \tag{310}$$

$$= (1-\gamma) \cdot \mu(s). \tag{311}$$

Partition the action set $\mathcal{A}$ under state $s \in \mathcal{S}$ into three parts using $V^{\pi_{\theta_t}}(s)$ as follows,

$$\mathcal{A}_t^0(s) := \left\{a^0 \in \mathcal{A} : Q^{\pi_{\theta_t}}(s, a^0) = V^{\pi_{\theta_t}}(s)\right\}, \tag{312}$$

$$\mathcal{A}_t^+(s) := \left\{a^+ \in \mathcal{A} : Q^{\pi_{\theta_t}}(s, a^+) > V^{\pi_{\theta_t}}(s)\right\}, \tag{313}$$

$$\mathcal{A}_t^-(s) := \left\{a^- \in \mathcal{A} : Q^{\pi_{\theta_t}}(s, a^-) < V^{\pi_{\theta_t}}(s)\right\}. \tag{314}$$

From Eq. (305), we have,

$$\mathbb{E}_t[V^{\pi_{\theta_{t+1}}}(\mu)] - V^{\pi_{\theta_t}}(\mu) \tag{315}$$

$$\geq \sum_s \mu(s) \cdot d_\mu^{\pi_{\theta_t}}(s) \sum_{a^+ \in \mathcal{A}_t^+(s)} \pi_{\theta_t}(a^+|s) \cdot \frac{\left[\exp\left\{\eta \cdot \frac{Q^{\pi_{\theta_t}}(s,a^+) - V^{\pi_{\theta_t}}(s)}{\pi_{\theta_t}(a^+|s)}\right\} - 1\right] \cdot (Q^{\pi_{\theta_t}}(s,a^+) - V^{\pi_{\theta_t}}(s))}{\exp\left\{\eta \cdot \frac{Q^{\pi_{\theta_t}}(s,a^+) - V^{\pi_{\theta_t}}(s)}{\pi_{\theta_t}(a^+|s)}\right\} + \frac{1 - \pi_{\theta_t}(a^+|s)}{\pi_{\theta_t}(a^+|s)}} \tag{316}$$

$$+ \sum_s \mu(s) \cdot d_\mu^{\pi_{\theta_t}}(s) \sum_{a^- \in \mathcal{A}_t^+(s)} \pi_{\theta_t}(a^-|s) \cdot \frac{\left[\exp\left\{\eta \cdot \frac{Q^{\pi_{\theta_t}}(s,a^-) - V^{\pi_{\theta_t}}(s)}{\pi_{\theta_t}(a^-|s)}\right\} - 1\right] \cdot (Q^{\pi_{\theta_t}}(s,a^-) - V^{\pi_{\theta_t}}(s))}{\exp\left\{\eta \cdot \frac{Q^{\pi_{\theta_t}}(s,a^-) - V^{\pi_{\theta_t}}(s)}{\pi_{\theta_t}(a^-|s)}\right\} + \frac{1 - \pi_{\theta_t}(a^-|s)}{\pi_{\theta_t}(a^-|s)}}. \tag{317}$$

For any $a^+ \in \mathcal{A}_t^+(t)$, using similar calculations in Eq. (45), we have,

$$\frac{\left[\exp\left\{\eta \cdot \frac{Q^{\pi_{\theta_t}}(s,a^+) - V^{\pi_{\theta_t}}(s)}{\pi_{\theta_t}(a^+|s)}\right\} - 1\right] \cdot (Q^{\pi_{\theta_t}}(s,a^+) - V^{\pi_{\theta_t}}(s))}{\exp\left\{\eta \cdot \frac{Q^{\pi_{\theta_t}}(s,a^+) - V^{\pi_{\theta_t}}(s)}{\pi_{\theta_t}(a^+|s)}\right\} + \frac{1 - \pi_{\theta_t}(a^+|s)}{\pi_{\theta_t}(a^+|s)}} \tag{318}$$

$$\geq \frac{\eta \cdot (Q^{\pi_{\theta_t}}(s,a^+) - V^{\pi_{\theta_t}}(s))^2}{\eta \cdot (Q^{\pi_{\theta_t}}(s,a^+) - V^{\pi_{\theta_t}}(s)) + 1} \tag{319}$$

$$\geq \frac{\eta}{1 + \frac{\eta}{1-\gamma}} \cdot \left(Q^{\pi_{\theta_t}}(s,a^+) - V^{\pi_{\theta_t}}(s)\right)^2 \qquad (Q^{\pi_\theta}(s,a) \in [0, 1/(1-\gamma)]) \tag{320}$$

$$\geq \frac{\eta}{1 + \frac{\eta}{1-\gamma}} \cdot \pi_{\theta_t}(a^+|s) \cdot \left(Q^{\pi_{\theta_t}}(s,a^+) - V^{\pi_{\theta_t}}(s)\right)^2. \qquad \left(\pi_{\theta_t}(a^+|s) \in (0,1)\right) \tag{321}$$

For any $a^- \in \mathcal{A}_t^-(s)$, using similar calculations in Eq. (47), we have,

$$\frac{\left[\exp\left\{\eta \cdot \frac{Q^{\pi_{\theta_t}}(s,a^-) - V^{\pi_{\theta_t}}(s)}{\pi_{\theta_t}(a^-|s)}\right\} - 1\right] \cdot (Q^{\pi_{\theta_t}}(s,a^-) - V^{\pi_{\theta_t}}(s))}{\exp\left\{\eta \cdot \frac{Q^{\pi_{\theta_t}}(s,a^-) - V^{\pi_{\theta_t}}(s)}{\pi_{\theta_t}(a^-|s)}\right\} + \frac{1 - \pi_{\theta_t}(a^-|s)}{\pi_{\theta_t}(a^-|s)}} \tag{322}$$

$$\geq \frac{\eta \cdot \pi_{\theta_t}(a^-|s) \cdot (V^{\pi_{\theta_t}}(s) - Q^{\pi_{\theta_t}}(s,a^-))^2}{\eta \cdot (V^{\pi_{\theta_t}}(s) - Q^{\pi_{\theta_t}}(s,a^-)) \cdot (1 - \pi_{\theta_t}(a^-|s)) + \pi_{\theta_t}(a^-|s)} \tag{323}$$

$$\geq \frac{\eta \cdot \pi_{\theta_t}(a^-|s) \cdot (V^{\pi_{\theta_t}}(s) - Q^{\pi_{\theta_t}}(s,a^-))^2}{\eta \cdot (V^{\pi_{\theta_t}}(s) - Q^{\pi_{\theta_t}}(s,a^-)) + 1} \qquad \left(\pi_{\theta_t}(a^-|s) \in (0,1)\right) \tag{324}$$

$$\geq \frac{\eta}{1 + \frac{\eta}{1-\gamma}} \cdot \pi_{\theta_t}(a^-|s) \cdot \left(V^{\pi_{\theta_t}}(s) - Q^{\pi_{\theta_t}}(s,a^-)\right)^2. \qquad (Q^{\pi_\theta}(s,a) \in [0, 1/(1-\gamma)]) \tag{325}$$

Combining Eqs. (315), (318) and (322), we have,

$$\mathbb{E}_t[V^{\pi_{\theta_{t+1}}}(\mu)] - V^{\pi_{\theta_t}}(\mu) \tag{326}$$

$$\geq \sum_s \mu(s) \cdot d_\mu^{\pi_{\theta_t}}(s) \sum_{a^+ \in \mathcal{A}_t^+(s)} \pi_{\theta_t}(a^+|s) \cdot \frac{\eta}{1 + \frac{\eta}{1-\gamma}} \cdot \pi_{\theta_t}(a^+|s) \cdot \left(Q^{\pi_{\theta_t}}(s, a^+) - V^{\pi_{\theta_t}}(s)\right)^2 \tag{327}$$

$$+ \sum_s \mu(s) \cdot d_\mu^{\pi_{\theta_t}}(s) \sum_{a^- \in \mathcal{A}_t^+(s)} \pi_{\theta_t}(a^-|s) \cdot \frac{\eta}{1 + \frac{\eta}{1-\gamma}} \cdot \pi_{\theta_t}(a^-|s) \cdot \left(V^{\pi_{\theta_t}}(s) - Q^{\pi_{\theta_t}}(s, a^-)\right)^2 \tag{328}$$

$$= \frac{\eta}{1 + \frac{\eta}{1-\gamma}} \cdot \sum_s \mu(s) \cdot d_\mu^{\pi_{\theta_t}}(s) \cdot \sum_a \pi_{\theta_t}(a|s)^2 \cdot \left(Q^{\pi_{\theta_t}}(s, a) - V^{\pi_{\theta_t}}(s)\right)^2 \tag{329}$$

$$\geq \frac{\eta \cdot (1-\gamma)}{1 + \eta} \cdot \sum_s \mu(s) \cdot d_\mu^{\pi_{\theta_t}}(s) \cdot \sum_a \pi_{\theta_t}(a|s)^2 \cdot \left(Q^{\pi_{\theta_t}}(s, a) - V^{\pi_{\theta_t}}(s)\right)^2 \tag{330}$$

Therefore, we have,

$$\mathbb{E}_t[V^{\pi_{\theta_{t+1}}}(\mu)] - V^{\pi_{\theta_t}}(\mu) \tag{331}$$

$$\geq \frac{\eta \cdot (1-\gamma)}{1 + \eta} \cdot \sum_s \mu(s) \cdot d_\mu^{\pi_{\theta_t}}(s) \cdot \pi_{\theta_t}(a^*(s)|s)^2 \cdot \left(Q^{\pi_{\theta_t}}(s, a^*(s)) - V^{\pi_{\theta_t}}(s)\right)^2 \qquad \text{(fewer terms)} \tag{332}$$

$$= \frac{\eta \cdot (1-\gamma)}{1 + \eta} \cdot \sum_s \mu(s) \cdot \frac{d_\mu^{\pi_{\theta_t}}(s)}{d_\mu^{\pi^*}(s)} \cdot d_\mu^{\pi^*}(s) \cdot \pi_{\theta_t}(a^*(s)|s)^2 \cdot \left(Q^{\pi_{\theta_t}}(s, a^*(s)) - V^{\pi_{\theta_t}}(s)\right)^2 \tag{333}$$

$$\geq \frac{\eta \cdot (1-\gamma)}{1 + \eta} \cdot \min_s \mu(s) \cdot \left\| \frac{d_\mu^{\pi^*}}{d_\mu^{\pi_{\theta_t}}} \right\|_\infty^{-1} \cdot \min_s \pi_{\theta_t}(a^*(s)|s)^2 \cdot \sum_s d_\mu^{\pi^*}(s) \cdot \left(Q^{\pi_{\theta_t}}(s, a^*(s)) - V^{\pi_{\theta_t}}(s)\right)^2 \tag{334}$$

$$\geq \frac{\eta \cdot (1-\gamma)^2}{1 + \eta} \cdot \min_s \mu(s) \cdot \left\| \frac{d_\mu^{\pi^*}}{\mu} \right\|_\infty^{-1} \cdot \min_s \pi_{\theta_t}(a^*(s)|s)^2 \cdot \sum_s d_\mu^{\pi^*}(s) \cdot \left(Q^{\pi_{\theta_t}}(s, a^*(s)) - V^{\pi_{\theta_t}}(s)\right)^2, \tag{335}$$

where $\min_s \mu(s) > 0$ is by Assumption 2, and the last inequality is according to Eq. (308),

$$\left\| \frac{d_\mu^{\pi^*}}{d_\mu^{\pi_{\theta_t}}} \right\|_\infty := \max_{s \in \mathcal{S}} \frac{d_\mu^{\pi^*}(s)}{d_\mu^{\pi_{\theta_t}}(s)} \leq \max_{s \in \mathcal{S}} \frac{d_\mu^{\pi^*}(s)}{(1-\gamma) \cdot \mu(s)} = \frac{1}{1-\gamma} \cdot \left\| \frac{d_\mu^{\pi^*}}{\mu} \right\|_\infty. \tag{336}$$

From Eq. (336), since $d_\mu^{\pi^*}(s)^2 \in (0, 1)$, we have,

$$\mathbb{E}_t[V^{\pi_{\theta_{t+1}}}(\mu)] - V^{\pi_{\theta_t}}(\mu) \tag{337}$$

$$\geq \frac{\eta \cdot (1-\gamma)^2}{1 + \eta} \cdot \min_s \mu(s) \cdot \left\| \frac{d_\mu^{\pi^*}}{\mu} \right\|_\infty^{-1} \cdot \min_s \pi_{\theta_t}(a^*(s)|s)^2 \cdot \sum_s d_\mu^{\pi^*}(s)^2 \cdot \left(Q^{\pi_{\theta_t}}(s, a^*(s)) - V^{\pi_{\theta_t}}(s)\right)^2 \tag{338}$$

$$\geq \frac{\eta \cdot (1-\gamma)^2}{1 + \eta} \cdot \min_s \mu(s) \cdot \left\| \frac{d_\mu^{\pi^*}}{\mu} \right\|_\infty^{-1} \cdot \frac{\min_s \pi_{\theta_t}(a^*(s)|s)^2}{S} \cdot \left[ \sum_s d_\mu^{\pi^*}(s) \cdot |Q^{\pi_{\theta_t}}(s, a^*(s)) - V^{\pi_{\theta_t}}(s)| \right]^2, \tag{339}$$

where the last inequality is by Cauchy–Schwarz. Note that,

$$\sum_s d_\mu^{\pi^*}(s) \cdot |Q^{\pi_{\theta_t}}(s, a^*(s)) - V^{\pi_{\theta_t}}(s)| \geq \sum_s d_\mu^{\pi^*}(s) \cdot \left(Q^{\pi_{\theta_t}}(s, a^*(s)) - V^{\pi_{\theta_t}}(s)\right) \tag{340}$$

$$= \sum_s d_\mu^{\pi^*}(s) \cdot \sum_a \left(\pi^*(a|s) - \pi_{\theta_t}(a|s)\right) \cdot Q^{\pi_{\theta_t}}(s, a) \tag{341}$$

$$= (1-\gamma) \cdot \left(V^{\pi^*}(\mu) - V^{\pi_{\theta_t}}(\mu)\right). \qquad \text{(by Lemma 17)} \tag{342}$$

Combining Eqs. (337) and (340), we have,

$$\mathbb{E}_t[V^{\pi_{\theta_{t+1}}}(\mu)] - V^{\pi_{\theta_t}}(\mu) \tag{343}$$

$$\geq \frac{\eta \cdot (1-\gamma)^4}{1+\eta} \cdot \min_s \mu(s) \cdot \left\| \frac{d_\mu^{\pi^*}}{\mu} \right\|_\infty^{-1} \cdot \frac{\min_s \pi_{\theta_t}(a^*(s)|s)^2}{S} \cdot \left(V^{\pi^*}(\mu) - V^{\pi_{\theta_t}}(\mu)\right)^2, \tag{344}$$

thus finishing the proofs. $\qquad\square$

**Lemma 4** (Non-vanishing stochastic NŁ coefficient / "automatic exploration"). *Using Algorithm 1 with the same assumptions as Lemma 3, with arbitrary initialization $\theta_1 \in \mathbb{R}^{\mathcal{S} \times \mathcal{A}}$, we have,*

$$c := \inf_{t \geq 1, s \in \mathcal{S}} \pi_{\theta_t}(a^*(s)|s) > 0, \qquad \text{a.s.} \tag{345}$$

*Proof.* Given any sampled state action pair $(s_t, a_t) = (s, i)$, we have,

$$\left[ V^{\pi_{\theta_{t+1}}}(\mu) \mid s_t = s, a_t = i \right] - V^{\pi_{\theta_t}}(\mu) \tag{346}$$

$$= \frac{1}{1-\gamma} \cdot \left[ \sum_{s'} d_\mu^{\pi_{\theta_{t+1}}}(s') \cdot \sum_a \left( \pi_{\theta_{t+1}}(a|s') - \pi_{\theta_t}(a|s') \right) \cdot Q^{\pi_{\theta_t}}(s', a) \, \Big| \, s_t = s, a_t = i \right] \tag{347}$$

$$= \frac{1}{1-\gamma} \cdot \left[ d_\mu^{\pi_{\theta_{t+1}}}(s) \cdot \sum_a \left( \pi_{\theta_{t+1}}(a|s) - \pi_{\theta_t}(a|s) \right) \cdot Q^{\pi_{\theta_t}}(s, a) \, \Big| \, a_t = i \right] \tag{348}$$

$$= \frac{1}{1-\gamma} \cdot d_\mu^{\pi_{\theta_{t+1}}}(s) \cdot \frac{\left[ \exp\left\{ \eta \cdot \frac{Q^{\pi_{\theta_t}}(s,i) - V^{\pi_{\theta_t}}(s)}{\pi_{\theta_t}(i|s)} \right\} - 1 \right] \cdot (Q^{\pi_{\theta_t}}(s,i) - V^{\pi_{\theta_t}}(s))}{\exp\left\{ \eta \cdot \frac{Q^{\pi_{\theta_t}}(s,i) - V^{\pi_{\theta_t}}(s)}{\pi_{\theta_t}(i|s)} \right\} + \frac{1 - \pi_{\theta_t}(i|s)}{\pi_{\theta_t}(i|s)}} \tag{349}$$

$$\geq 0, \qquad \text{(by Eq. (301))} \tag{350}$$

where the second equation is due to $\pi_{\theta_{t+1}}(a|s') = \pi_{\theta_t}(a|s')$ for all $s' \neq s$ by Algorithm 1.

From Eq. (346), we have $V^{\pi_{\theta_{t+1}}}(\mu) \geq V^{\pi_{\theta_t}}(\mu)$ holds almost surely. According to the definition of $Q^\pi(s, a)$, we have,

$$Q^{\pi_{\theta_{t+1}}}(s, a) - Q^{\pi_{\theta_t}}(s, a) = \gamma \cdot \sum_{s'} \mathcal{P}(s'|s, a) \cdot (V^{\pi_{\theta_{t+1}}}(s') - V^{\pi_{\theta_t}}(s')) \geq 0, \tag{351}$$

where the last inequality is by Eq. (346). Also note that $Q^\pi(s, a) \in [0, 1/(1-\gamma)]$ since $r(s, a) \in [0, 1]$ for all $(s, a) \in \mathcal{S} \times \mathcal{A}$. According to monotone convergence theorem, we have, for all $(s, a) \in \mathcal{S} \times \mathcal{A}$, the following exists,

$$Q^\infty(s, a) := \lim_{t \to \infty} Q^{\pi_{\theta_t}}(s, a). \tag{352}$$

Also, define $V^\infty(s) := \lim_{t \to \infty} V^{\pi_{\theta_t}}(s)$ for all $s \in \mathcal{S}$.

For all state $s \in \mathcal{S}$, given $i \in \mathcal{A}$, define the following set $\mathcal{P}(s, i)$ of "generalized one-hot policy" under state $s$,

$$\mathcal{A}(s, i) := \{ j \in \mathcal{A} : Q^\infty(s, j) = Q^\infty(s, i) \}, \tag{353}$$

$$\mathcal{P}(s, i) := \left\{ \pi(\cdot|s) \in \Delta(\mathcal{A}) : \sum_{j \in \mathcal{A}(s,i)} \pi(j|s) = 1 \right\}. \tag{354}$$

Similar to Claims 1 and 2 in the proofs for Lemma 2, we make the following two claims.

**Claim 3.** *Almost surely, $\pi_{\theta_t}(\cdot|s)$ approaches one "generalized one-hot policy" under all state $s \in \mathcal{S}$, i.e., there exists (a possibly random) $i \in \mathcal{A}$, such that $\sum_{j \in \mathcal{A}(s,i)} \pi_{\theta_t}(j|s) \to 1$ as $t \to \infty$ almost surely as $t \to \infty$.*

**Claim 4.** *Almost surely, $\pi_{\theta_t}(\cdot|s)$ cannot approach any "sub-optimal generalized one-hot policies" under all state $s \in \mathcal{S}$, i.e., $i$ in the previous claim must be an optimal action.*

From Claim 4, it follows that $\sum_{j \in \mathcal{A}(a^*(s))} \pi_{\theta_t}(j|s) \to 1$ almost surely under all state $s \in \mathcal{S}$, as $t \to \infty$ and thus the policy sequence obtained almost surely convergences to a globally optimal policy $\pi^*$.

**Proof of Claim 3.**

Using similar arguments in Eq. (128), we have,

$$\lim_{t \to \infty} \mathbb{E}_t[V^{\pi_{\theta_{t+1}}}(\mu)] - V^{\pi_{\theta_t}}(\mu) = 0, \qquad \text{a.s.} \tag{355}$$

According to Eqs. (292) and (326), we have,

$$\mathbb{E}_t[V^{\pi_{\theta_{t+1}}}(\mu)] - V^{\pi_{\theta_t}}(\mu) \geq \sum_s d_\mu^{\pi_{\theta_t}}(s) \cdot \mu(s) \cdot \frac{\eta \cdot (1-\gamma)}{1+\eta} \cdot \sum_a \pi_{\theta_t}(a|s)^2 \cdot (Q^{\pi_{\theta_t}}(s,a) - V^{\pi_{\theta_t}}(s))^2. \tag{356}$$

Since $d_\mu^{\pi_{\theta_t}}(s) \geq (1-\gamma) \cdot \mu(s) > 0$ by Eq. (308) and Assumption 2, we have, almost surely,

$$\lim_{t \to \infty} \sum_s \sum_a \pi_{\theta_t}(a|s)^2 \cdot (Q^{\pi_{\theta_t}}(s,a) - V^{\pi_{\theta_t}}(s))^2 = 0, \tag{357}$$

which implies that for all $s \in \mathcal{S}$, almost surely,

$$\lim_{t \to \infty} \sum_a \pi_{\theta_t}(a|s)^2 \cdot (Q^{\pi_{\theta_t}}(s,a) - V^{\pi_{\theta_t}}(s))^2 = 0. \tag{358}$$

Using similar arguments in Eq. (130), we have, for each state $s \in \mathcal{S}$, there exists $i \in \mathcal{A}$, such that,

$$\lim_{t \to \infty} \sum_{j \in \mathcal{A}(s,i)} \pi_{\theta_t}(j|s) = 1, \qquad \text{a.s.,} \tag{359}$$

which means $\pi_{\theta_t}(\cdot|s)$ a.s. approaches the "generalized one-hot policy" $\mathcal{P}(s,i)$ in Eq. (354) as $t \to \infty$, finishing the proof of Claim 3.

**Proof of Claim 4.** The brief sketch of the proof is as follows: By Claim 3, for each state $s \in \mathcal{S}$, there exists a (possibly random) $i \in \mathcal{A}$ such that $\sum_{j \in \mathcal{A}(s,i)} \pi_{\theta_t}(j|s) \to 1$ almost surely, as $t \to \infty$. If $i = a^*(s)$ almost surely, Claim 4 follows. Hence, it suffices to consider the event that $\{i \neq a^*(s)\}$ for at least one state $s \in \mathcal{S}$, and show that this event has zero probability mass. Hence, in the rest of the proof we assume that we are on the event when $i \neq a^*(s)$ for one state $s \in \mathcal{S}$.

Since $i \neq a^*(s)$, there exists at least one "good" action $a^+ \in \mathcal{A}$ such that $Q^\infty(s, a^+) > Q^\infty(s, i)$. The two cases are as follows.

**2a)** All "good" actions are sampled finitely many times as $t \to \infty$.

**2b)** At least one "good" action is sampled infinitely many times as $t \to \infty$.

In both cases, we show that $\sum_{j \in \mathcal{A}(s,i)} \exp\{\theta_t(j|s)\} < \infty$ as $t \to \infty$ (but for different reasons), which is a contradiction with the assumption of $\sum_{j \in \mathcal{A}(s,i)} \pi_{\theta_t}(j|s) \to 1$ as $t \to \infty$, given that a "good" action's parameter is almost surely lower bounded. Hence, $i \neq a^*(s)$ almost surely does not happen, which means that almost surely $i = a^*(s)$. Let

$$I_t(s,a) = \begin{cases} 1, & \text{if } (s_t, a_t) = (s,a); \\ 0, & \text{otherwise}. \end{cases} \tag{360}$$

Define the following notations,

$$N_t(s,a) := \sum_{u=1}^t I_s(s,a), \tag{361}$$

$$N_\infty(s,a) := \sum_{u=1}^\infty I_u(s,a). \tag{362}$$

Assume $\{i \neq a^*(s)\}$ for at least one state $s \in \mathcal{S}$, and $\sum_{j \in \mathcal{A}(s,i)} \pi_{\theta_t}(j|s) \to 1$ almost surely. Partition the action set $\mathcal{A}$ under $s \in \mathcal{S}$ into three parts using $V^\infty(s)$ as follows,

$$\mathcal{A}(s,i) := \left\{ j \in \mathcal{A} : Q^\infty(s,j) = Q^\infty(s,i) \right\}, \tag{363}$$

$$\mathcal{A}^+(s,i) := \left\{ a^+ \in \mathcal{A} : Q^\infty(s,a^+) > Q^\infty(s,i) \right\}, \tag{364}$$

$$\mathcal{A}^-(s,i) := \left\{ a^- \in \mathcal{A} : Q^\infty(s,a^-) < Q^\infty(s,i) \right\}. \tag{365}$$

Since $i \neq a^*(s)$, we have, $\mathcal{A}^+(s,i) \neq \emptyset$. Note that,

$$\left| V^{\pi_{\theta_t}}(s) - Q^\infty(s,i) \right| = \Bigg| \sum_{k \notin \mathcal{A}(s,i)} \pi_{\theta_t}(k|s) \cdot (Q^{\pi_{\theta_t}}(s,k) - Q^\infty(s,i)) \tag{366}$$

$$+ \sum_{\substack{j \neq i, \\ j \in \mathcal{A}(s,i)}} \pi_{\theta_t}(j|s) \cdot (Q^{\pi_{\theta_t}}(s,j) - Q^\infty(s,i)) \Bigg| \tag{367}$$

$$\leq \sum_{k \notin \mathcal{A}(s,i)} \pi_{\theta_t}(k|s) \cdot |Q^{\pi_{\theta_t}}(s,k) - Q^\infty(s,i)| \qquad \text{(triangle inequality)} \tag{368}$$

$$+ \sum_{\substack{j \neq i, \\ j \in \mathcal{A}(s,i)}} \pi_{\theta_t}(j|s) \cdot |Q^{\pi_{\theta_t}}(s,j) - Q^\infty(s,i)| \tag{369}$$

$$\leq \frac{1}{1-\gamma} \cdot \underbrace{\left( 1 - \sum_{j \in \mathcal{A}(s,i)} \pi_{\theta_t}(j|s) \right)}_{\to 0} + \sum_{\substack{j \neq i, \\ j \in \mathcal{A}(s,i)}} \underbrace{|Q^{\pi_{\theta_t}}(s,j) - Q^\infty(s,i)|}_{\to 0}, \tag{370}$$

which implies that $V^{\pi_{\theta_t}}(s) \to Q^\infty(s,i)$ as $t \to \infty$. Therefore, there exists $1 \leq \tau$, almost surely on $\{i \neq a^*(s)\}$ $\tau < \infty$ while we also have, for all $t \geq \tau$,

$$Q^{\pi_{\theta_t}}(s,a^+) - c \geq V^{\pi_{\theta_t}}(s) \geq Q^{\pi_{\theta_t}}(s,a^-) + c, \tag{371}$$

$$\tag{372}$$

for all $a^+ \in \mathcal{A}^+(s,i)$, $a^- \in \mathcal{A}^-(s,i)$, where $c > 0$. For all $t \geq \tau$, for any $a^+ \in \mathcal{A}^+(s,a)$, we have, almost surely,

$$\theta_{t+1}(s,a^+) = \theta_t(s,a^+) + \eta \cdot I_t(s,a^+) \cdot \frac{Q^{\pi_{\theta_t}}(s,a^+) - V^{\pi_{\theta_t}}(s)}{\pi_{\theta_t}(a^+|s)} \qquad \text{(by Algorithm 1)} \tag{373}$$

$$\geq \theta_t(s,a^+) + \eta \cdot I_t(s,a^+) \cdot \frac{c}{\pi_{\theta_t}(a^+|s)} \qquad \text{(by Eq. (371))} \tag{374}$$

$$\geq \theta_t(s,a^+) + \eta \cdot I_t(s,a^+) \cdot c \qquad \left( \pi_{\theta_t}(a^+|s) \in (0,1) \right) \tag{375}$$

$$\geq \theta_t(s,a^+), \tag{376}$$

which implies that, almost surely,

$$c_1 := \inf_{t \geq 1} \theta_t(s,a^+) > -\infty. \tag{377}$$

On the other hand, for all $t \geq \tau$, for any $a^- \in \mathcal{A}^-(s,a)$, we have, almost surely,

$$\theta_{t+1}(s,a^-) = \theta_t(s,a^-) + \eta \cdot I_t(s,a^-) \cdot \frac{Q^{\pi_{\theta_t}}(s,a^-) - V^{\pi_{\theta_t}}(s)}{\pi_{\theta_t}(a^-|s)} \qquad \text{(by Algorithm 1)} \tag{378}$$

$$\leq \theta_t(s,a^-) - \eta \cdot I_t(s,a^-) \cdot \frac{c}{\pi_{\theta_t}(a^-|s)} \qquad \text{(by Eq. (371))} \tag{379}$$

$$\leq \theta_t(s,a^-) - \eta \cdot I_t(s,a^-) \cdot c \qquad \left( \pi_{\theta_t}(a^-|s) \in (0,1) \right) \tag{380}$$

$$\leq \theta_t(s,a^-), \tag{381}$$

which implies that, almost surely,

$$c_2 := \sup_{t \geq 1} \theta_t(s,a^-) < \infty. \tag{382}$$

**First case. 2a).** Consider the event,

$$\mathcal{E}_0 := \bigcap_{a^+ \in \mathcal{A}^+(s,i)} \underbrace{\left\{ N_\infty(s, a^+) < \infty \right\}}_{\mathcal{E}_0(s,a^+)}, \tag{383}$$

i.e., any "good" action $a^+ \in \mathcal{A}^+(s,i)$ has finitely many updates as $t \to \infty$. Using the extended Borel-Cantelli lemma (Lemma 14), we have, almost surely,

$$\left\{ \sum_{t \geq 1} \pi_{\theta_t}(a^+|s) < \infty \right\} = \left\{ N_\infty(s, a^+) < \infty \right\}. \tag{384}$$

Next, we have, almost surely,

$$1 - \sum_{j \in \mathcal{A}(s,i)} \pi_{\theta_t}(j|s) = \frac{\sum_{a^+ \in \mathcal{A}^+(s,i)} e^{\theta_t(s,a^+)} + \sum_{a^- \in \mathcal{A}^-(s,i)} e^{\theta_t(s,a^-)}}{\sum_{a \in \mathcal{A}} e^{\theta_t(s,a)}} \tag{385}$$

$$\leq \frac{\sum_{a^+ \in \mathcal{A}^+(s,i)} e^{\theta_t(s,a^+)} + \sum_{a^- \in \mathcal{A}^-(s,i)} e^{c_2}}{\sum_{a \in \mathcal{A}} e^{\theta_t(s,a)}} \qquad \text{(by Eq. (382))} \tag{386}$$

$$= \frac{\sum_{a^+ \in \mathcal{A}^+(s,i)} e^{\theta_t(s,a^+)} + e^{c_2 - c_1} \cdot \frac{|\mathcal{A}^-(s,i)|}{|\mathcal{A}^+(s,i)|} \cdot |\mathcal{A}^+(s,i)| \cdot e^{c_1}}{\sum_{a \in \mathcal{A}} e^{\theta_t(s,a)}} \tag{387}$$

$$\leq \frac{\sum_{a^+ \in \mathcal{A}^+(s,i)} e^{\theta_t(s,a^+)} + e^{c_2 - c_1} \cdot \frac{|\mathcal{A}^-(s,i)|}{|\mathcal{A}^+(s,i)|} \cdot \sum_{a^+ \in \mathcal{A}^+(s,i)} e^{\theta_t(s,a^+)}}{\sum_{a \in \mathcal{A}} e^{\theta_t(s,a)}} \qquad \text{(by Eq. (377))} \tag{388}$$

$$= \frac{\sum_{a^+ \in \mathcal{A}^+(s,i)} e^{\theta_t(s,a^+)}}{\sum_{a \in \mathcal{A}} e^{\theta_t(s,a)}} \cdot \left( 1 + e^{c_2 - c_1} \cdot \frac{|\mathcal{A}^-(s,i)|}{|\mathcal{A}^+(s,i)|} \right) \tag{389}$$

$$= \left( 1 + e^{c_2 - c_1} \cdot \frac{|\mathcal{A}^-(s,i)|}{|\mathcal{A}^+(s,i)|} \right) \cdot \sum_{a^+ \in \mathcal{A}^+(s,i)} \pi_{\theta_t}(a^+|s). \tag{390}$$

Define

$$q_t := \sum_{a^+ \in \mathcal{A}^+(s,i)} \pi_{\theta_t}(a^+|s). \tag{391}$$

According to Eq. (384), we have, on $\mathcal{E}_0$, almost surely,

$$\sum_{t=1}^{\infty} q_t < \infty. \tag{392}$$

On the other hand, according to the assumption of $\sum_{j \in \mathcal{A}(s,i)} \pi_{\theta_t}(j|s) \to 1$, there exists at least one $j \in \mathcal{A}(s,i)$, such that almost surely, for all $t \geq \tau$, $\pi_{\theta_t}(j|s) > c'$ for some $c' > 0$. We have,

$$\theta_{t+1}(s,j) = \theta_t(s,j) + \eta \cdot I_t(s,j) \cdot \frac{Q^{\pi_{\theta_t}}(s,j) - V^{\pi_{\theta_t}}(s)}{\pi_{\theta_t}(j|s)} \qquad \text{(by Algorithm 1)} \tag{393}$$

$$\leq \theta_t(s,j) + \eta \cdot I_t(s,j) \cdot \frac{1 - \sum_{j \in \mathcal{A}(s,i)} \pi_{\theta_t}(j|s)}{\pi_{\theta_t}(j|s)} \cdot \frac{1}{1 - \gamma} \tag{394}$$

$$\leq \theta_t(s,j) + \eta \cdot I_t(s,j) \cdot \frac{1 - \sum_{j \in \mathcal{A}(s,i)} \pi_{\theta_t}(j|s)}{c'} \cdot \frac{1}{1 - \gamma}, \qquad \left( \pi_{\theta_t}(j|s) > c' \right) \tag{395}$$

which implies that, for $C := \max_{t \in [1,\tau]} \theta_t(s,j)$, we have

$$\sup_{t \geq 1} \theta_t(s,j) \leq C + \frac{\eta \cdot \left( 1 + e^{c_2 - c_1} \cdot \frac{|\mathcal{A}^-(s,i)|}{|\mathcal{A}^+(s,i)|} \right)}{(1 - \gamma) \cdot c'} \cdot \sum_{t=\tau}^{\infty} \sum_{a^+ \in \mathcal{A}^+(s,i)} \pi_{\theta_t}(a^+|s) < \infty. \tag{396}$$

Following calculations in Eq. (210), almost surely on $\mathcal{E}' := \mathcal{E}_0 \cap \{i \neq a^*(s)\}$, we have, $\sum_{j \in \mathcal{A}(s,i)} \pi_{\theta_t}(j|s) \not\to 1$, which is a contradiction with the assumption, showing that $\mathbb{P}(\mathcal{E}') = 0$.

**Second case. 2b).** Consider the complement $\mathcal{E}_0^c$ of $\mathcal{E}_0$, where $\mathcal{E}_0$ is by Eq. (383). We now show that also $\mathbb{P}(\mathcal{E}'') = 0$ where $\mathcal{E}'' = \mathcal{E}_0^c \cap \{i \neq a^*(s)\}$.

Pick $a^+ \in \mathcal{A}^+(s,i)$, such that $\mathbb{P}(N_\infty(s,a^+) = \infty) > 0$. On event $\mathcal{E}_\infty(s,a^+) := \{N_\infty(s,a^+) = \infty\}$, accoding to Eq. (373), we have, almost surely,

$$c_3 := \lim_{t\to\infty} \theta_t(s,a^+) = \infty. \tag{397}$$

Therefore, we have, for all $t \geq \tau$,

$$V^{\pi_{\theta_t}}(s) = Q^{\pi_{\theta_t}}(s,i) + \sum_{\substack{j \neq i, \\ j \in \mathcal{A}(s,i)}} \pi_{\theta_t}(j|s) \cdot \underbrace{(Q^{\pi_{\theta_t}}(s,j) - Q^{\pi_{\theta_t}}(s,i))}_{\to 0} \tag{398}$$

$$+ \sum_{a^- \in \mathcal{A}^-(s,i)} \pi_{\theta_t}(a^-|s) \cdot \underbrace{\left(Q^{\pi_{\theta_t}}(s,a^-) - Q^{\pi_{\theta_t}}(s,i)\right)}_{<0} \tag{399}$$

$$+ \sum_{\tilde{a}^+ \in \mathcal{A}^+(s,i)} \pi_{\theta_t}(\tilde{a}^+|s) \cdot \underbrace{\left(Q^{\pi_{\theta_t}}(s,\tilde{a}^+) - Q^{\pi_{\theta_t}}(s,i)\right)}_{>0} \tag{400}$$

$$\geq Q^{\pi_{\theta_t}}(s,i) + \sum_{\substack{j \neq i, \\ j \in \mathcal{A}(s,i)}} \pi_{\theta_t}(j|s) \cdot (Q^{\pi_{\theta_t}}(s,j) - Q^{\pi_{\theta_t}}(s,i)) \tag{401}$$

$$+ \pi_{\theta_t}(a^+|s) \cdot \left[ \left(Q^{\pi_{\theta_t}}(s,a^+) - Q^{\pi_{\theta_t}}(s,i)\right) - \sum_{a^- \in \mathcal{A}^-(s,i)} \frac{Q^{\pi_{\theta_t}}(s,i) - Q^{\pi_{\theta_t}}(s,a^-)}{\exp\{\theta_t(s,a^+) - \theta_t(s,a^-)\}} \right]. \tag{402}$$

According to Eqs. (382) and (397), $\theta_t(s,a^+) - \theta_t(s,a^-) \to \infty$, which implies that, on event $\mathcal{E}_\infty(s,a^+)$, almost surely, for all $t \geq \tau$,

$$V^{\pi_{\theta_t}}(s) > Q^{\pi_{\theta_t}}(s,i) + \sum_{\substack{j \neq i, \\ j \in \mathcal{A}(s,i)}} \pi_{\theta_t}(j|s) \cdot (Q^{\pi_{\theta_t}}(s,j) - Q^{\pi_{\theta_t}}(s,i)), \tag{403}$$

which implies that,

$$\sum_{k \in \mathcal{A}(s,i)} \pi_{\theta_t}(k|s) \cdot V^{\pi_{\theta_t}}(s) > \sum_{k \in \mathcal{A}(s,i)} \pi_{\theta_t}(k|s) \cdot Q^{\pi_{\theta_t}}(s,k) \tag{404}$$

$$+ \sum_{k \in \mathcal{A}(s,i)} \pi_{\theta_t}(k|s) \cdot \sum_{\substack{j \neq k, \\ j \in \mathcal{A}(s,i)}} \pi_{\theta_t}(j|s) \cdot (Q^{\pi_{\theta_t}}(s,j) - Q^{\pi_{\theta_t}}(s,k)) \tag{405}$$

$$= \sum_{k \in \mathcal{A}(s,i)} \pi_{\theta_t}(k|s) \cdot Q^{\pi_{\theta_t}}(s,k). \tag{406}$$

For all $t \geq \tau$, we have,

$$\theta_{t+1}(s,i) = \theta_t(s,i) + \eta \cdot I_t(s,i) \cdot \frac{Q^{\pi_{\theta_t}}(s,i) - V^{\pi_{\theta_t}}(s)}{\pi_{\theta_t}(i|s)} \qquad \text{(by Algorithm 1)} \tag{407}$$

$$\leq \theta_t(s,i), \tag{408}$$

which implies that,

$$\sup_{t \geq 1} \theta_t(s,i) < \infty. \tag{409}$$

Following calculations in Eq. (242), almost surely on $\mathcal{E}'' = \mathcal{E}_0^c \cap \{i \neq a^*(s)\}$, we have, $\sum_{j \in \mathcal{A}(s,i)} \pi_{\theta_t}(j|s) \not\to 1$, which is a contradiction with the assumption, showing that $\mathbb{P}(\mathcal{E}'') = 0$. $\quad\square$

**Theorem 2** (Almost sure global convergence rate) **.** Using Algorithm 1 with any initialization $\theta_1 \in \mathbb{R}^K$, under the same assumptions as Lemmas 3, we have, for all $t \geq 1$,

$$\mathbb{E}[V^*(\mu) - V^{\pi_{\theta_t}}(\mu)] \leq \frac{1 + \eta}{\eta \cdot (1 - \gamma)^4 \cdot \min_s \mu(s)} \cdot \left\| \frac{d_\mu^{\pi^*}}{\mu} \right\|_\infty \cdot \frac{S}{\mathbb{E}[c^2]} \cdot \frac{1}{t}, \quad \text{and} \tag{410}$$

$$\limsup_{t \geq 1} \left\{ \frac{\eta \cdot (1 - \gamma)^4 \cdot \min_s \mu(s)}{1 + \eta} \cdot \left\| \frac{d_\mu^{\pi^*}}{\mu} \right\|_\infty^{-1} \cdot \frac{c^2 \cdot t}{S} \cdot (V^*(\mu) - V^{\pi_{\theta_t}}(\mu)) \right\} < \infty, \quad \text{a.s.,} \tag{411}$$

where we use $\mathbb{E}_t[\cdot]$ to denote $\mathbb{E}_t[\cdot | \mathcal{F}_t]$ for brevity, and $\mathcal{F}_t$ is the $\sigma$-algebra generated by $(s_1, a_1), (s_2, a_2), \ldots, (s_{t-1}, a_{t-1})$, $\pi^*$ is the global optimal policy, $S$ is the state number, $\min_s \mu(s) > 0$ by Assumption 2, and $c := \inf_{t \geq 1, s \in \mathcal{S}} \pi_{\theta_t}(a^*(s)|s) > 0$ is from Lemma 4.

*Proof.* **First part.** According to Lemma 3, we have,

$$\mathbb{E}_t[V^{\pi_{\theta_{t+1}}}(\mu)] - V^{\pi_{\theta_t}}(\mu) \tag{412}$$

$$\geq \frac{\eta \cdot (1 - \gamma)^4 \cdot \min_s \mu(s)}{1 + \eta} \cdot \left\| \frac{d_\mu^{\pi^*}}{\mu} \right\|_\infty^{-1} \cdot \frac{\min_s \pi_{\theta_t}(a^*(s)|s)^2}{S} \cdot \left( V^{\pi^*}(\mu) - V^{\pi_{\theta_t}}(\mu) \right)^2 \tag{413}$$

$$\geq \frac{\eta \cdot (1 - \gamma)^4 \cdot \min_s \mu(s)}{1 + \eta} \cdot \left\| \frac{d_\mu^{\pi^*}}{\mu} \right\|_\infty^{-1} \cdot \frac{\inf_{t \geq 1, s \in \mathcal{S}} \pi_{\theta_t}(a^*(s)|s)^2}{S} \cdot \left( V^{\pi^*}(\mu) - V^{\pi_{\theta_t}}(\mu) \right)^2 \tag{414}$$

$$= \frac{\eta \cdot (1 - \gamma)^4 \cdot \min_s \mu(s)}{1 + \eta} \cdot \left\| \frac{d_\mu^{\pi^*}}{\mu} \right\|_\infty^{-1} \cdot \frac{c^2}{S} \cdot \left( V^{\pi^*}(\mu) - V^{\pi_{\theta_t}}(\mu) \right)^2, \tag{415}$$

where $c := \inf_{t \geq 1, s \in \mathcal{S}} \pi_{\theta_t}(a^*(s)|s) > 0$ according to Lemma 4. Let $\delta(\theta_t) := V^*(\mu) - V^{\pi_{\theta_t}}(\mu)$ denote the sub-optimality gap. Using similar calculations in Theorem 1, we have, for all $t \geq 1$,

$$\mathbb{E}[V^*(\mu) - V^{\pi_{\theta_t}}(\mu)] = \mathbb{E}[\delta(\theta_t)] \leq \frac{1 + \eta}{\eta \cdot (1 - \gamma)^4 \cdot \min_s \mu(s)} \cdot \left\| \frac{d_\mu^{\pi^*}}{\mu} \right\|_\infty \cdot \frac{S}{\mathbb{E}[c^2]} \cdot \frac{1}{t}. \tag{416}$$

**Second part.** The result follows from Lemma 12 by choosing $X_t = V^*(\mu) - V^{\pi_{\theta_t}}(\mu)$ and $f(t) = \frac{\eta \cdot (1 - \gamma)^4 \cdot \min_s \mu(s)}{1 + \eta} \cdot \left\| \frac{d_\mu^{\pi^*}}{\mu} \right\|_\infty^{-1} \cdot \frac{\mathbb{E}[c^2]}{S} \cdot t.$ $\square$

## C  Proofs for Understanding Baselines

**Proposition 2** (Unbiasedness of NPG)**.** For NPG with and without a state value baseline, corresponding to Updates 1 and 2 respectively, we have $\mathbb{E}_{a_t \sim \pi_{\theta_t}(\cdot)}[\hat{r}_t] = \mathbb{E}_{a_t \sim \pi_{\theta_t}(\cdot)}[\hat{r}_t - \hat{b}_t] = r.$

*Proof.* **First part.** $\mathbb{E}_{a_t \sim \pi_{\theta_t}(\cdot)}[\hat{r}_t] = r.$

According to Definition 2, we have, for all $i \in [K]$,

$$\underset{a_t \sim \pi_{\theta_t}(\cdot)}{\mathbb{E}}[\hat{r}_t(i)] = \sum_{a \in [K]} \mathbb{P}(a_t = a) \cdot \hat{r}_t(i) \tag{417}$$

$$= \sum_{a \in [K]} \pi_{\theta_t}(a) \cdot \frac{\mathbb{I}\{a = i\}}{\pi_{\theta_t}(i)} \cdot r(i) = r(i). \qquad (a_t \sim \pi_{\theta_t}(\cdot)) \tag{418}$$

**Second part.** $\mathbb{E}_{a_t \sim \pi_{\theta_t}(\cdot)}[\hat{r}_t - \hat{b}_t] = r.$ According to Definition 2, we have, for all $i \in [K]$,

$$\underset{a_t \sim \pi_{\theta_t}(\cdot)}{\mathbb{E}}[\hat{r}_t(i) - \hat{b}_t(i)] = \sum_{a \in [K]} \pi_{\theta_t}(a) \cdot \left[ \frac{\mathbb{I}\{a = i\}}{\pi_{\theta_t}(i)} \cdot \left( r(i) - \pi_{\theta_t}^\top r \right) + \pi_{\theta_t}^\top r \right] \qquad \text{(by Update 2)} \tag{419}$$

$$= r(i) - \pi_{\theta_t}^\top r + \pi_{\theta_t}^\top r \tag{420}$$

$$= r(i). \qquad \square$$

**Proposition 3** (Unboundedness of NPG)**.** For NPG without a baseline, Update 1, we have $\mathbb{E}_{a_t \sim \pi_{\theta_t}(\cdot)} \|\hat{r}_t\|_2^2 = \sum_{a \in [K]} \frac{r(a)^2}{\pi_{\theta_t}(a)}$. For NPG with a state value baseline, Update 2, we have $\mathbb{E}_{a_t \sim \pi_{\theta_t}(\cdot)} \|\hat{r}_t - \hat{b}_t\|_2^2 = \sum_{a \in [K]} \frac{(r(a) - \pi_{\theta_t}^\top r)^2}{\pi_{\theta_t}(a)} - K \cdot (\pi_{\theta_t}^\top r)^2 + 2 \cdot (\pi_{\theta_t}^\top r) \cdot (r^\top \mathbf{1})$.

*Proof.* **First part.** $\mathbb{E}_{a_t \sim \pi_{\theta_t}(\cdot)} \|\hat{r}_t\|_2^2 = \sum_{a \in [K]} \frac{r(a)^2}{\pi_{\theta_t}(a)}$.

According to Definition 2, we have,

$$\|\hat{r}_t\|_2^2 = \sum_i \hat{r}_t(i)^2 = \sum_i \frac{(\mathbb{I}\{a_t = i\})^2}{\pi_{\theta_t}(i)^2} \cdot r(i)^2 = \sum_i \frac{\mathbb{I}\{a_t = i\}}{\pi_{\theta_t}(i)^2} \cdot r(i)^2. \tag{421}$$

Taking expectation, we have,

$$\mathbb{E}_{a_t \sim \pi_{\theta_t}(\cdot)} \|\hat{r}_t\|_2^2 = \sum_{a \in [K]} \pi_{\theta_t}(a) \cdot \sum_i \frac{\mathbb{I}\{a = i\}}{\pi_{\theta_t}(i)^2} \cdot r(i)^2 \tag{422}$$

$$= \sum_{a \in [K]} \pi_{\theta_t}(a) \cdot \frac{1}{\pi_{\theta_t}(a)^2} \cdot r(a)^2 \tag{423}$$

$$= \sum_{a \in [K]} \frac{r(a)^2}{\pi_{\theta_t}(a)}. \tag{424}$$

**Second part.** $\mathbb{E}_{a_t \sim \pi_{\theta_t}(\cdot)} \|\hat{r}_t - \hat{b}_t\|_2^2 = \sum_{a \in [K]} \frac{(r(a) - \pi_{\theta_t}^\top r)^2}{\pi_{\theta_t}(a)} - K \cdot (\pi_{\theta_t}^\top r)^2 + 2 \cdot (\pi_{\theta_t}^\top r) \cdot (r^\top \mathbf{1})$.

According to Definition 2, we have,

$$\left\|\hat{r}_t - \hat{b}_t\right\|_2^2 = \sum_i \left(\hat{r}_t(i) - \hat{b}_t(i)\right)^2 \tag{425}$$

$$= \sum_i \left[\frac{\mathbb{I}\{a_t = i\}}{\pi_{\theta_t}(i)} \cdot \left(r(i) - \pi_{\theta_t}^\top r\right) + \pi_{\theta_t}^\top r\right]^2 \tag{426}$$

$$= \sum_i \frac{(\mathbb{I}\{a_t = i\})^2}{\pi_{\theta_t}(i)^2} \cdot \left(r(i) - \pi_{\theta_t}^\top r\right)^2 + \sum_i \left(\pi_{\theta_t}^\top r\right)^2 + 2 \cdot \sum_i \frac{\mathbb{I}\{a_t = i\}}{\pi_{\theta_t}(i)} \cdot \left(r(i) - \pi_{\theta_t}^\top r\right) \cdot \left(\pi_{\theta_t}^\top r\right) \tag{427}$$

$$= \sum_i \frac{\mathbb{I}\{a_t = i\}}{\pi_{\theta_t}(i)^2} \cdot \left(r(i) - \pi_{\theta_t}^\top r\right)^2 + K \cdot \left(\pi_{\theta_t}^\top r\right)^2 + 2 \cdot \sum_i \frac{\mathbb{I}\{a_t = i\}}{\pi_{\theta_t}(i)} \cdot \left(r(i) - \pi_{\theta_t}^\top r\right) \cdot \left(\pi_{\theta_t}^\top r\right). \tag{428}$$

Taking expectation, we have,

$$\mathbb{E}_{a_t \sim \pi_{\theta_t}(\cdot)} \left\|\hat{r}_t - \hat{b}_t\right\|_2^2 = \sum_{a \in [K]} \pi_{\theta_t}(a) \cdot \sum_i \frac{\mathbb{I}\{a = i\}}{\pi_{\theta_t}(i)^2} \cdot \left(r(i) - \pi_{\theta_t}^\top r\right)^2 \tag{429}$$

$$+ \sum_{a \in [K]} \pi_{\theta_t}(a) \cdot K \cdot \left(\pi_{\theta_t}^\top r\right)^2 + 2 \cdot \left(\pi_{\theta_t}^\top r\right) \cdot \sum_{a \in [K]} \pi_{\theta_t}(a) \cdot \sum_i \frac{\mathbb{I}\{a_t = i\}}{\pi_{\theta_t}(i)} \cdot \left(r(i) - \pi_{\theta_t}^\top r\right) \tag{430}$$

$$= \sum_{a \in [K]} \pi_{\theta_t}(a) \cdot \frac{1}{\pi_{\theta_t}(a)^2} \cdot \left(r(a) - \pi_{\theta_t}^\top r\right)^2 \tag{431}$$

$$+ K \cdot \left(\pi_{\theta_t}^\top r\right)^2 + 2 \cdot \left(\pi_{\theta_t}^\top r\right) \cdot \sum_{a \in [K]} \pi_{\theta_t}(a) \cdot \frac{1}{\pi_{\theta_t}(a)} \cdot \left(r(a) - \pi_{\theta_t}^\top r\right) \tag{432}$$

$$= \sum_{a \in [K]} \frac{(r(a) - \pi_{\theta_t}^\top r)^2}{\pi_{\theta_t}(a)} - K \cdot (\pi_{\theta_t}^\top r)^2 + 2 \cdot (\pi_{\theta_t}^\top r) \cdot (r^\top \mathbf{1}). \qquad \square$$

**Lemma 5** (Bad sampling). Let $\pi_{\theta_t}(a) \in (0, 1)$ be the probability of sampling action $a$ using online sampling $a_t \sim \pi_{\theta_t}(\cdot)$, for all $t \geq 1$. If $1 - \pi_{\theta_t}(a) \in O(1/t^{1+\epsilon})$, where $\epsilon > 0$, then $\prod_{t=1}^{\infty} \pi_{\theta_t}(a) > 0$.

*Proof.* According to Lemma 18, we have, for a sequence $u_t \in (0, 1)$ for all $t \geq 1$, if $\sum_{t=1}^{\infty} u_t < \infty$, then $\prod_{t=1}^{\infty} (1 - u_t) > 0$.

Let $u_t = 1 - \pi_{\theta_t}(a) \in (0, 1)$ according to the softmax parameterization. If $1 - \pi_{\theta_t}(a) \in O(1/t^{1+\epsilon})$, such as $1 - \pi_{\theta_t}(a) \in \Theta(1/t^{\alpha})$ where $a \in (1, \infty)$, then we have, for all $C > 0$,

$$\sum_{t=1}^{\infty} u_t = \sum_{t=1}^{\infty} (1 - \pi_{\theta_t}(a)) \tag{433}$$

$$= \sum_{t=1}^{\infty} \frac{C}{t^{\alpha}} \tag{434}$$

$$\leq C \cdot \left( 1 + \int_{t=1}^{\infty} \frac{1}{t^{\alpha}} dt \right) \tag{435}$$

$$= \frac{C \cdot \alpha}{\alpha - 1}, \tag{436}$$

or if $1 - \pi_{\theta_t}(a) \in \Theta(e^{-c \cdot t})$ where $c > 0$, then we have, for all $C > 0$ and $C' > 0$,

$$\sum_{t=1}^{\infty} u_t = \sum_{t=1}^{\infty} (1 - \pi_{\theta_t}(a)) \tag{437}$$

$$= \sum_{t=1}^{\infty} \frac{C}{\exp\{C' \cdot t\}} \tag{438}$$

$$\leq \int_{t=0}^{\infty} \frac{C}{\exp\{C' \cdot t\}} \tag{439}$$

$$= \frac{C}{C'}. \tag{440}$$

Therefore, using Lemma 18, we have,

$$\prod_{t=1}^{\infty} (1 - u_t) = \prod_{t=1}^{\infty} \pi_{\theta_t}(a) > 0, \tag{441}$$

finishing the proofs. $\qquad \square$

**Lemma 6** (NPG aggressiveness). Fix sampling $a_t = a$ for all $t \geq 1$, using Update 1 with constant learning rate $\eta > 0$, where $\hat{r}_t$ is from Definition 2, we have $1 - \pi_{\theta_t}(a) \in O(e^{-c \cdot t})$ for all $t \geq 1$, where $c > 0$.

*Proof.* See [21, Theorem 3]. We include a proof for completeness.

Suppose $a_1 = a, a_2 = a, \cdots, a_{t-1} = a$. We have,

$$\theta_t(a) = \theta_1(a) + \eta \cdot \sum_{s=1}^{t-1} \hat{r}_s(a) \qquad \text{(by Update 1)} \tag{442}$$

$$= \theta_1(a) + \eta \cdot \sum_{s=1}^{t-1} \frac{\mathbb{I}\{a_s = a\}}{\pi_{\theta_s}(a)} \cdot r(a) \qquad \text{(by Definition 2)} \tag{443}$$

$$= \theta_1(a) + \eta \cdot \sum_{s=1}^{t-1} \frac{r(a)}{\pi_{\theta_s}(a)} \qquad (a_s = a \text{ for all } s \in \{1, 2, \ldots, t-1\}) \tag{444}$$

$$\geq \theta_1(a) + \eta \cdot \sum_{s=1}^{t-1} r(a) \qquad (\pi_{\theta_s}(a) \in (0, 1)) \tag{445}$$

$$= \theta_1(a) + \eta \cdot r(a) \cdot (t - 1). \tag{446}$$

On the other hand, we have, for any other action $a' \neq a$,

$$\theta_t(a') = \theta_1(a') + \eta \cdot \sum_{s=1}^{t-1} \frac{\mathbb{I}\{a_s = a'\}}{\pi_{\theta_s}(a')} \cdot r(a') \qquad \text{(by Update 1 and Definition 2)} \qquad (447)$$

$$= \theta_1(a'). \qquad (a_s \neq a' \text{ for all } s \in \{1, 2, \ldots, t-1\}) \qquad (448)$$

Therefore, we have,

$$\pi_{\theta_t}(a) = 1 - \sum_{a' \neq a} \pi_{\theta_t}(a') \qquad (449)$$

$$= 1 - \frac{\sum_{a' \neq a} \exp\{\theta_t(a')\}}{\exp\{\theta_t(a)\} + \sum_{a' \neq a} \exp\{\theta_t(a')\}} \qquad (450)$$

$$\geq 1 - \frac{\sum_{a' \neq a} \exp\{\theta_1(a')\}}{\exp\{\theta_1(a) + \eta \cdot r(a) \cdot (t-1)\} + \sum_{a' \neq a} \exp\{\theta_1(a')\}}, \qquad \text{(by Eqs. (442) and (447))} \qquad (451)$$

which implies that,

$$1 - \pi_{\theta_t}(a) \leq \frac{\sum_{a' \neq a} \exp\{\theta_1(a')\}}{\exp\{\theta_1(a) + \eta \cdot r(a) \cdot (t-1)\} + \sum_{a' \neq a} \exp\{\theta_1(a')\}} \qquad (452)$$

$$\in O(e^{-c \cdot t}), \qquad (453)$$

where $c := \eta \cdot r(a) > 0$. $\qquad \square$

**Lemma 7** (Good sampling). Let $\pi_{\theta_t}(a) \in (0,1)$ and $a_t \sim \pi_{\theta_t}(\cdot)$, for all $t \geq 1$. If $\sum_{t=1}^{\infty} (1 - \pi_{\theta_t}(a)) = \infty$ (e.g., $1 - \pi_{\theta_t}(a) \in \Omega(1/t)$), then $\prod_{t=1}^{\infty} \pi_{\theta_t}(a) = 0$.

*Proof.* According to Lemma 19, we have, for a sequence $u_t \in (0,1)$ for all $t \geq 1$, if $\sum_{t=1}^{\infty} u_t = \infty$, then $\prod_{t=1}^{\infty} (1 - u_t) = 0$.

Let $u_t = 1 - \pi_{\theta_t}(a) \in (0,1)$ according to the softmax parameterization, the result follows. $\qquad \square$

**Lemma 8** (Value baselines reduce NPG aggressiveness). Fix sampling $a_t = a$ for all $t \geq 1$. Then using Update 2 with a constant learning rate $\eta > 0$ and $\hat{r}_t$ from Definition 2 obtains $1 - \pi_{\theta_t}(a) \in \Omega(1/t)$ for all $t \geq 1$.

*Proof.* Since the claim is concerned with the policies underlying the parameter vectors and not the parameter vectors themselves, as noted after Update 2, we used the equivalent Update 3 with the change of $\hat{r}_t$ is from Definition 2 as follows,

$$\theta_{t+1}(a) \leftarrow \theta_t(a) + \eta \cdot \frac{\mathbb{I}\{a_t = a\}}{\pi_{\theta_t}(a)} \cdot \left(r(a) - \pi_{\theta_t}^\top r\right). \qquad (454)$$

Since $a_t = a$ for all $t \geq 1$ by assumption, we have,

$$\theta_{t+1}(a) \leftarrow \theta_t(a) + \eta \cdot \frac{r(a) - \pi_{\theta_t}^\top r}{\pi_{\theta_t}(a)}, \qquad (455)$$

while for all $a' \neq a$,

$$\theta_{t+1}(a') \leftarrow \theta_t(a'). \qquad (456)$$

If $\pi_{\theta_t}^\top r < r(a)$, then we have,

$$\theta_{t+1}(a) = \theta_t(a) + \eta \cdot \frac{r(a) - \pi_{\theta_t}^\top r}{\pi_{\theta_t}(a)} \qquad \text{(by Eq. (455))} \qquad (457)$$

$$\geq 0, \qquad \left(\pi_{\theta_t}^\top r < r(a)\right) \qquad (458)$$

which implies that,

$$\pi_{\theta_{t+1}}(a) = \frac{\exp\{\theta_{t+1}(a)\}}{\exp\{\theta_{t+1}(a)\} + \sum_{a' \neq a} \exp\{\theta_{t+1}(a')\}} \tag{459}$$

$$= \frac{\exp\{\theta_{t+1}(a)\}}{\exp\{\theta_{t+1}(a)\} + \sum_{a' \neq a} \exp\{\theta_t(a')\}} \quad \text{(by Eq. (456))} \tag{460}$$

$$\geq \frac{\exp\{\theta_t(a)\}}{\exp\{\theta_t(a)\} + \sum_{a' \neq a} \exp\{\theta_t(a')\}} \quad \text{(by Eq. (457))} \tag{461}$$

$$= \pi_{\theta_t}(a), \tag{462}$$

which means $1 - \pi_{\theta_t}(a)$ is decreasing. Otherwise, if $\pi_{\theta_t}^\top r \geq r(a)$, then using similar calculations, we have $\pi_{\theta_{t+1}}(a) \leq \pi_{\theta_t}(a)$, i.e., $1 - \pi_{\theta_t}(a)$ is increasing and will not approach $0$. Since we prove $1 - \pi_{\theta_t}(a) \in \Omega(1/t)$, we assume the non-trivial case where $\pi_{\theta_t}^\top r < r(a)$ for all $t \geq 1$.

According to Lemma 20, we have,

$$\left| \pi_{\theta_{t+1}}(a) - \pi_{\theta_t}(a) - \left\langle \frac{d\pi_{\theta_t}(a)}{d\theta_t}, \theta_{t+1} - \theta_t \right\rangle \right| \leq \frac{3}{4} \cdot \|\theta_{t+1} - \theta_t\|_2^2. \tag{463}$$

Therefore, we have,

$$(1 - \pi_{\theta_t}(a)) - (1 - \pi_{\theta_{t+1}}(a)) = \pi_{\theta_{t+1}}(a) - \pi_{\theta_t}(a) - \left\langle \frac{d\pi_{\theta_t}(a)}{d\theta_t}, \theta_{t+1} - \theta_t \right\rangle + \left\langle \frac{d\pi_{\theta_t}(a)}{d\theta_t}, \theta_{t+1} - \theta_t \right\rangle \tag{464}$$

$$\leq \frac{3}{4} \cdot \|\theta_{t+1} - \theta_t\|_2^2 + \left\langle \frac{d\pi_{\theta_t}(a)}{d\theta_t}, \theta_{t+1} - \theta_t \right\rangle \quad \text{(by Eq. (455))} \tag{465}$$

$$= \frac{3 \cdot \eta^2}{4} \cdot \frac{(r(a) - \pi_{\theta_t}^\top r)^2}{\pi_{\theta_t}(a)^2} + \eta \cdot \frac{d\pi_{\theta_t}(a)}{d\theta_t(a)} \cdot \frac{r(a) - \pi_{\theta_t}^\top r}{\pi_{\theta_t}(a)}, \quad \text{(using the update)} \tag{466}$$

$$= \frac{3 \cdot \eta^2}{4} \cdot \frac{(r(a) - \pi_{\theta_t}^\top r)^2}{\pi_{\theta_t}(a)^2} + \eta \cdot (1 - \pi_{\theta_t}(a)) \cdot (r(a) - \pi_{\theta_t}^\top r) \quad \left( \frac{d\pi_{\theta_t}(a)}{d\theta_t(a)} = \pi_{\theta_t}(a) \cdot (1 - \pi_{\theta_t}(a)) \right) \tag{467}$$

$$\leq \frac{3 \cdot \eta^2}{4} \cdot \frac{(r(a) - \pi_{\theta_t}^\top r)^2}{\pi_{\theta_1}(a)^2} + \eta \cdot (1 - \pi_{\theta_t}(a)) \cdot (r(a) - \pi_{\theta_t}^\top r) \quad \text{(by Eq. (459))} \tag{468}$$

$$\leq \frac{3 \cdot \eta^2}{4} \cdot \frac{(1 - \pi_{\theta_t}(a))^2}{\pi_{\theta_1}(a)^2} + \eta \cdot (1 - \pi_{\theta_t}(a))^2 \tag{469}$$

$$= C \cdot (1 - \pi_{\theta_t}(a))^2 \quad \left( C := \frac{3 \cdot \eta^2}{4 \cdot \pi_{\theta_1}(a)^2} + \eta \right) \tag{470}$$

where the last inequality is because of,

$$r(a) - \pi_{\theta_t}^\top r = \sum_{a' \neq a} \pi_{\theta_t}(a') \cdot (r(a) - r(a')) \tag{471}$$

$$\leq 1 - \pi_{\theta_t}(a). \quad \left( r \in (0, 1]^K \right) \tag{472}$$

Next, we have,

$$\frac{1}{1 - \pi_{\theta_t}(a)} = \frac{1}{1 - \pi_{\theta_1}(a)} + \sum_{s=1}^{t-1} \left[ \frac{1}{1 - \pi_{\theta_{s+1}}(a)} - \frac{1}{1 - \pi_{\theta_s}(a)} \right] \tag{473}$$

$$= \frac{1}{1 - \pi_{\theta_1}(a)} + \sum_{s=1}^{t-1} \frac{1}{\left(1 - \pi_{\theta_{s+1}}(a)\right) \cdot \left(1 - \pi_{\theta_s}(a)\right)} \cdot \left[ \left(1 - \pi_{\theta_s}(a)\right) - \left(1 - \pi_{\theta_{s+1}}(a)\right) \right] \tag{474}$$

$$\leq \frac{1}{1 - \pi_{\theta_1}(a)} + \sum_{s=1}^{t-1} \frac{1}{\left(1 - \pi_{\theta_{s+1}}(a)\right) \cdot \left(1 - \pi_{\theta_s}(a)\right)} \cdot C \cdot \left(1 - \pi_{\theta_s}(a)\right)^2 \qquad \text{(by Eq. (464))} \tag{475}$$

$$\leq \frac{1}{1 - \pi_{\theta_1}(a)} + \frac{C}{2} \cdot (t - 1), \tag{476}$$

which implies that, for all large enough $t \geq 1$,

$$1 - \pi_{\theta_t}(a) \geq \frac{1}{\frac{1}{1 - \pi_{\theta_1}(a)} + \frac{C}{2} \cdot (t - 1)} \in \Omega(1/t). \qquad \square$$

## D  Simulation Settings

### D.1  One-state MDPs

The detailed settings for simulations in Figure 2 are as follows. The total number of actions is $K = 20$, and after sorting rewards the true mean reward vector $r \in (0, 1)^K$ is,

$$\begin{aligned}
r = (&0.96990985, \ 0.95071431, \ 0.86617615, \ 0.83244264, \\
&0.73199394, \ 0.70807258, \ 0.60111501, \ 0.59865848, \\
&0.52475643, \ 0.43194502, \ 0.37454012, \ 0.30424224, \\
&0.29122914, \ 0.21233911, \ 0.18340451, \ 0.18182497, \\
&0.15601864, \ 0.15599452, \ 0.05808361, \ 0.02058449)^\top.
\end{aligned}$$

For each $a \in [K]$, the sampled reward distribution is Bernoulli$(0.5)$, such that with probability $0.5$, one of the following two sampled reward values is observed,

$$\begin{aligned}
R_1 &= (-2.03009015, \ 3.96990985), & R_2 &= (-2.04928569, \ 3.95071431), \\
R_3 &= (-2.13382385, \ 3.86617615), & R_4 &= (-2.16755736, \ 3.83244264), \\
R_5 &= (-2.26800606, \ 3.73199394), & R_6 &= (-2.29192742, \ 3.70807258), \\
R_7 &= (-2.39888499, \ 3.60111501), & R_8 &= (-2.40134152, \ 3.59865848), \\
R_9 &= (-2.47524357, \ 3.52475643), & R_{10} &= (-2.56805498, \ 3.43194502), \\
R_{11} &= (-2.62545988, \ 3.37454012), & R_{12} &= (-2.69575776, \ 3.30424224), \\
R_{13} &= (-2.70877086, \ 3.29122914), & R_{14} &= (-2.78766089, \ 3.21233911), \\
R_{15} &= (-2.81659549, \ 3.18340451), & R_{16} &= (-2.81817503, \ 3.18182497), \\
R_{17} &= (-2.84398136, \ 3.15601864), & R_{18} &= (-2.84400548, \ 3.15599452), \\
R_{19} &= (-2.94191639, \ 3.05808361), & R_{20} &= (-2.97941551, \ 3.02058449).
\end{aligned}$$

The initial parameter $\theta_1 \in \mathbb{R}^K$ is,

$$\theta(i) = \begin{cases} 5, & \text{if } i = 2, \\ 0, & \text{otherwise,} \end{cases} \tag{477}$$

such that the initial probability of best sub-optimal action is,

$$\pi_{\theta_1}(2) = \frac{e^5}{e^5 + 19 \cdot e^0} \approx 0.8865, \tag{478}$$

and all the other action's probability, including the optimal action, is

$$\pi_{\theta_1}(1) = \frac{e^0}{e^5 + 19 \cdot e^0} \approx 0.0060. \tag{479}$$

We run Update 2 with learning rate,

$$\eta = \frac{1}{2} \cdot \frac{\pi_{\theta_t}(a_t) \cdot \left| r(a_t) - \pi_{\theta_t}^\top r \right|}{9}, \tag{480}$$

and the results are shown in Figures 2a and 2b.

For the results in Figure 2c, Definition 2 is used, i.e., the true mean reward value $r(a_t)$ is observed for sampled action $a_t$, and we run the same update Update 2 using the same true mean reward vector $r \in (0,1)^K$ with learning rate $\eta = 0.1$ and uniform initial policy $\pi_{\theta_1}(a) = 1/K$ for all $a \in [K]$.

### D.2 Tree MDPs

We conduct experiments using a synthetic tree MDP with depth $d = 4$ and branch factor (number of actions) $k = 4$. The total number of states is

$$S = \sum_{i=0}^{d-1} k^i = \sum_{i=0}^{3} 4^i = 85. \tag{481}$$

The discount factor $\gamma = 0.9$. For each state $s \in \mathcal{S}$, the immediate reward vector is,

$$r(s, \cdot) := (1.0, 0.9, 0.8, 0.2)^\top. \tag{482}$$

The state distribution $\rho$ we used to measure the sub-optimality gap $V^*(\rho) - V^{\pi_{\theta_t}}(\rho)$ is $\rho(s_0) = 1$ for the root state $s_0$. The initial state distribution $\mu$ we used in the algorithm is set to satisfy Assumption 2 as follows,

$$\mu = 0.2 \cdot \rho + \frac{0.8}{S-1} \cdot (1-\rho), \tag{483}$$

i.e., $\mu(s_0) = 0.2$ and $\mu(s') = \frac{0.8}{84}$ for any other state $s' \neq s_0$. We use an adversarial initialization, such that optimal actions have smallest initial probabilities, i.e., for all $s \in \mathcal{S}$,

$$\pi_{\theta_1}(a^*(s)|s) = 0.07, \tag{484}$$

and $\pi_{\theta_1}(a'|s) = 0.31$ for any sub-optimal action $a' \neq a^*(s)$, where the optimal action $a^*(s)$ and policy $\pi^*$ are calculated using dynamic programming.

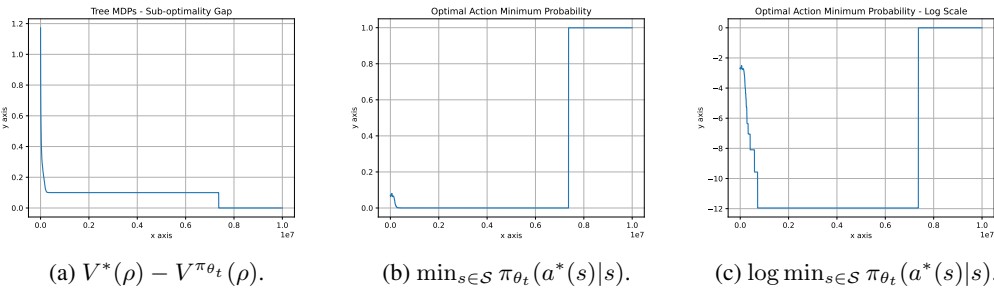

(a) $V^*(\rho) - V^{\pi_{\theta_t}}(\rho)$.     (b) $\min_{s \in \mathcal{S}} \pi_{\theta_t}(a^*(s)|s)$.     (c) $\log \min_{s \in \mathcal{S}} \pi_{\theta_t}(a^*(s)|s)$.

Figure 3: Results on a tree MDP, adversarial initialization.

As shown in Figure 3, the sub-optimality gap $V^*(\rho) - V^{\pi_{\theta_t}}(\rho)$ quickly approached about $0.1$ value, while the optimal action's minimum probability $\min_{s \in \mathcal{S}} \pi_{\theta_t}(a^*(s)|s)$ approaching very close to 0. The algorithm got stuck on the sub-optimality plateau and finally escaped and approached the global optimal policy $\pi^*$ after about $7 \times 10^6$ iterations.

Figure 4 demonstrates a more detailed process of the optimization. Note that the tree MDP has four layers of states, with state numbers $S_1 = 1$ (root state), $S_2 = k = 4$, $S_3 = k^2 = 16$, and $S_4 = k^3 = 64$, respectively. We calculated the optimal actions' probabilities for each layers of states. For example, Figure 4(b) shows $\pi_{\theta_t}(a^*(s)|s)$ for all state $s$ in Layer 2.

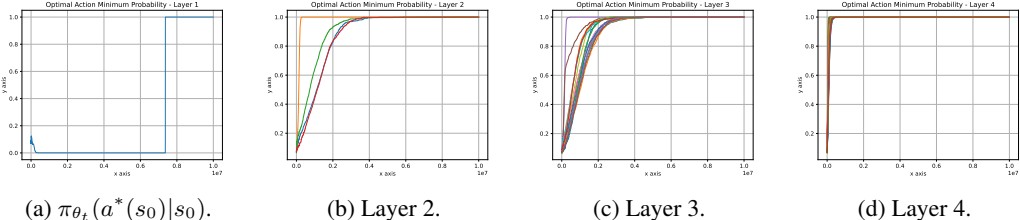

(a) $\pi_{\theta_t}(a^*(s_0)|s_0)$.      (b) Layer 2.      (c) Layer 3.      (d) Layer 4.

Figure 4: Optimal actions' probabilities for different layers of states.

As shown in Figure 4, $\pi_{\theta_t}(a^*(s)|s)$ for states in Layer 4 approaches to 1 most quickly comparing to other layers of states. However, it took $\pi_{\theta_t}(a^*(s)|s)$ for Layers 2 and 3 several millions of iterations to approach 1, and in the meanwhile $\pi_{\theta_t}(a^*(s_0)|s_0)$ decreased to near zero values. Therefore, $a^*(s_0)$ would have very small chance to be sampled and learned using on-policy sampling, which created the sub-optimality plateau for about $7 \times 10^6$ iterations.

## E    Miscellaneous Extra Supporting Results

Recall that $(X_t, \mathcal{F}_t)_{t \geq 1}$ is a *sub-martingale* (super-martingale, martingale) if $(X_t)_{t \geq 1}$ is adapted to the filtration $(\mathcal{F}_t)_{t \geq 1}$ and $\mathbb{E}[X_{t+1}|\mathcal{F}_t] \geq X_t$ ($\mathbb{E}[X_{t+1}|\mathcal{F}_t] \leq X_t$, $\mathbb{E}[X_{t+1}|\mathcal{F}_t] = X_t$, respectively) holds almost surely for any $t \geq 1$. For brevity, let $\mathbb{E}_t[\cdot]$ denote $\mathbb{E}[\cdot|\mathcal{F}_t]$ where the filtration should be clear from the context and we also extend this notation to $t = 0$ such that $\mathbb{E}_0 U = \mathbb{E}[U]$.

**Theorem 3** (Theorem 13.3.2 of [3]). *Let $(X_t, \mathcal{F}_t)_{t \geq 1}$ be a sub-martingale such that $\sup_{n \geq 1} \mathbb{E}[X_n^+] < \infty$. Then $(X_t)_{t \geq 1}$ converges to a finite limit $X_\infty$ a.s. and $\mathbb{E}[|X_\infty|] < \infty$.*

Theorem 3 implies the following Theorem 4.

**Theorem 4** (Doob's supermartingale convergence theorem [9]). *If $(Y_t)_{t \geq 1}$ is an $\{\mathcal{F}_t\}_{t \geq 1}$-adapted sequence such that $\mathbb{E}[Y_{t+1}|\mathcal{F}_t] \leq Y_t$ and $\sup_t \mathbb{E}[|Y_t|] < \infty$ then $\{Y_t\}_{t \geq 1}$ almost surely converges (a.s.) and, in particular, $Y_t \to Y$ a.s. as $t \to \infty$ where $Y = \limsup_{t \to \infty} Y_t$ is such that $\mathbb{E}[|Y|] < \infty$.*

**Lemma 13.** *Let $(X_t, \mathcal{F}_t)_{t \geq 1}$ be a sub-martingale such that $\sup_{n \geq 1} \mathbb{E}[X_n^+] < \infty$. Let $Z_n = \sum_{t=0}^{n-1} X_{t+1} - \mathbb{E}_t[X_{t+1}]$ and assume that for any $n$, $\mathbb{E}[|Z_n|] < \infty$. Then, $X_{t+1} - \mathbb{E}_t[X_{t+1}] \to 0$ almost surely as $t \to \infty$.*

*Proof.* By construction, and the assumption that $\mathbb{E}[|Z_n|] < \infty$, $(Z_n, \mathcal{F}_n)_{n \geq 1}$ is a martingale and as such, it is also a sub-martingale. Further, for any $n \geq 1$,

$$
\begin{aligned}
Z_n &= (X_n - \mathbb{E}_{n-1}[X_n]) + (X_{n-1} - \mathbb{E}_{n-2}[X_{n-1}]) + \cdots + (X_1 - \mathbb{E}_0[X_1]) \\
&= X_n + (X_{n-1} - \mathbb{E}_{n-1}[X_n]) + (X_{n-2} - \mathbb{E}_{n-2}[X_{n-1}]) + \cdots + (X_1 - \mathbb{E}_1[X_2]) - \mathbb{E}_0[X_1] \\
&\leq X_n - \mathbb{E}_0[X_1].
\end{aligned}
$$

Hence, $Z_n^+ \leq (X_n - \mathbb{E}_0[X_1])^+ \leq (X_n + |\mathbb{E}_0[X_1]|)^+ \leq X_n^+ + \mathbb{E}[|X_1|]$, and hence $\sup_{n \geq 1} \mathbb{E}[Z_n^+] \leq \sup_{n \geq 1} \mathbb{E}[X_n^+] + \mathbb{E}[|X_1|] < \infty$. Applying Theorem 3 to $(Z_n, \mathcal{F}_n)_{n \geq 1}$, we get that there exist a random variable $Z_\infty$ such that $\mathbb{E}[|Z_\infty|] < \infty$ and $Z_n \to Z_\infty$ almost surely as $n \to \infty$. On the set where $(Z_n)_{n \geq 1}$ converges to $Z_\infty$, $(Z_n)_{n \geq 1}$ is a Cauchy sequence, and it follows that $|X_{n+1} - \mathbb{E}_n X_{n+1}| = |Z_{n+1} - Z_n| \to 0$, finishing the proof. $\qquad \square$

**Corollary 3.** *Let $(X_t, \mathcal{F}_t)_{t \geq 1}$ be a sub-martingale such that $X_n \in [a, b]$ almost surely for some reals $a < b$. Let $Z_n = \sum_{t=0}^{n-1} X_{t+1} - \mathbb{E}_t[X_{t+1}]$ and assume that for any $n$, $\mathbb{E}[|Z_n|] < \infty$. Then, $X_{t+1} - \mathbb{E}_t[X_{t+1}] \to 0$ almost surely as $t \to \infty$.*

*Proof.* We use Lemma 13, hence we need to verify that the conditions of this result hold. Clearly, $\sup_{n \geq 1} \mathbb{E}[X_n^+] \leq b^+ < \infty$. Next, we have for any $n \geq 1$ that $|Z_n| \leq \sum_{t=0}^{n-1} |X_{t+1} - \mathbb{E}_t[X_{t+1}]| \leq n(b - a) < \infty$ since $\mathbb{E}_t[X_{t+1}] \in [a, b]$ also holds when $X_{t+1} \in [a, b]$. $\qquad \square$

**Lemma 14** (Extended Borel-Cantelli Lemma, Corollary 5.29 of [5])**.** *Let $(\mathcal{F}_n)_{n \geq 1}$ be a filtration, $A_n \in \mathcal{F}_n$. Then, almost surely,*

$$\{\omega \, : \, \omega \in A_n \text{ infinitely often }\} = \left\{\omega \, : \, \sum_{n=1}^{\infty} \mathbb{P}(A_n | \mathcal{F}_n)\right\}.$$

**Lemma 15** (Piecewise linear domination for sigmoid-like functions)**.** *Given $p \in (0, 1]$, define the following function,*

$$f_p(y) := \frac{e^y - 1}{e^y + \frac{1-p}{p}}. \tag{485}$$

*For any fixed $p \in (0, 1]$, and any fixed $\epsilon \in [0, 1]$, we have,*

$$(1 - \epsilon) \cdot p \cdot y \leq f_p(y) \leq (1 + \epsilon) \cdot p \cdot y, \quad \text{for all } y \in [0, \epsilon], \tag{486}$$
$$(1 + \epsilon) \cdot p \cdot y \leq f_p(y) \leq (1 - \epsilon) \cdot p \cdot y, \quad \text{for all } y \in [-\epsilon, 0]. \tag{487}$$

*Proof.* **First part.** For $y = 0$ or $\epsilon = 0$, Eqs. (486) and (487) hold trivially.

First, if $y = 0$, then we have $f_p(y) = p \cdot y = 0$, which means Eqs. (486) and (487) hold. Next, if $\epsilon = 0$, then $y = 0$ (since we prove for $|y| \leq \epsilon$) and Eqs. (486) and (487) again hold trivially.

We then prove for $\epsilon \in (0, 1]$ and for $y \neq 0$. Define the following function, for $p \in [0, 1]$,

$$g_p(y) := \frac{e^y - 1}{p \cdot y \cdot (e^y - 1) + y}, \quad \text{for all } y \neq 0. \tag{488}$$

**Second part.** Eq. (486). We prove for any fixed $p \in (0, 1]$, and any fixed $\epsilon \in (0, 1]$,

$$1 - \epsilon \leq g_p(y) \leq 1 + \epsilon, \text{ for all } y \in (0, \epsilon]. \tag{489}$$

First, for $p = 1$, and any fixed $\epsilon \in (0, 1]$, we have, for all $y \in (0, \epsilon]$,

$$g_1(y) = \frac{e^y - 1}{y \cdot e^y} \quad \text{(by Eq. (488))} \tag{490}$$

$$= \frac{1 - e^{-y}}{y} \tag{491}$$

$$\geq \frac{y - y^2}{y} \quad \left(e^{-y} \leq 1 - y + y^2, \text{ for all } y > 0\right) \tag{492}$$

$$= 1 - y \quad (y > 0) \tag{493}$$

$$\geq 1 - \epsilon. \quad (y \in (0, \epsilon]) \tag{494}$$

Second, for $p = 0$, and any fixed $\epsilon \in (0, 1]$, we have, for all $y \in (0, \epsilon]$,

$$g_0(y) = \frac{e^y - 1}{y} \quad \text{(by Eq. (488))} \tag{495}$$

$$\leq \frac{y + y^2}{y} \quad \left(e^y \leq 1 + y + y^2, \text{ for all } y \leq 1\right) \tag{496}$$

$$= 1 + y \quad (y > 0) \tag{497}$$

$$\leq 1 + \epsilon. \quad (y \in (0, \epsilon]) \tag{498}$$

Note that, for any $y > 0$, we have, $g_p(y)$ is monotonically decreasing over $p$, since

$$g_p(y)^{-1} = p \cdot y + \frac{y}{e^y - 1} \tag{499}$$

is monotonically increasing over $p$.

Therefore, we have, any fixed $p \in (0, 1]$, and any fixed $\epsilon \in (0, 1]$, for all $y \in (0, \epsilon]$,

$$1 - \epsilon \leq g_1(y) \quad \text{(by Eq. (490))} \tag{500}$$

$$\leq g_p(y) \quad (g_p(y) \text{ is monotonically decreasing over } p) \tag{501}$$

$$\leq g_0(y) \tag{502}$$

$$\leq 1 + \epsilon, \quad \text{(by Eq. (495))} \tag{503}$$

Note that,

$$f_p(y) = \frac{e^y - 1}{e^y + \frac{1-p}{p}} \qquad \text{(by Eq. (485))} \tag{504}$$

$$= \frac{e^y - 1}{p \cdot y \cdot (e^y - 1) + y} \cdot p \cdot y \qquad (p \in (0, 1], \epsilon \in (0, 1], \text{ and } y \in (0, \epsilon]) \tag{505}$$

$$= g_p(y) \cdot p \cdot y. \qquad \text{(by Eq. (488))} \tag{506}$$

Therefore, according to Eqs. (500) and (504), we have,

$$(1 - \epsilon) \cdot p \cdot y \leq f_p(y) \leq (1 + \epsilon) \cdot p \cdot y, \qquad (p \cdot y > 0) \tag{507}$$

which means any fixed $p \in (0, 1]$, and any fixed $\epsilon \in (0, 1]$, Eq. (486) holds for all $y \in (0, \epsilon]$.

**Second part.** Eq. (487). We prove for any fixed $p \in (0, 1]$, and any fixed $\epsilon \in (0, 1]$,

$$1 - \epsilon \leq g_p(y) \leq 1 + \epsilon, \text{ for all } y \in [-\epsilon, 0). \tag{508}$$

First, for $p = 1$, and any fixed $\epsilon \in (0, 1]$, we have, for all $y \in [-\epsilon, 0)$,

$$g_1(y) = \frac{e^y - 1}{y \cdot e^y} \qquad \text{(by Eq. (488))} \tag{509}$$

$$= \frac{1 - e^{-y}}{y} \tag{510}$$

$$\leq \frac{y - y^2}{y} \qquad \left( e^{-y} \leq 1 - y + y^2, \text{ for all } y \geq -1 \right) \tag{511}$$

$$= 1 - y \qquad (y < 0) \tag{512}$$

$$\leq 1 + \epsilon. \qquad (y \in [-\epsilon, 0)) \tag{513}$$

Second, for $p = 0$, and any fixed $\epsilon \in (0, 1]$, we have, for all $y \in [-\epsilon, 0)$,

$$g_0(y) = \frac{e^y - 1}{y} \qquad \text{(by Eq. (488))} \tag{514}$$

$$\geq \frac{y + y^2}{y} \qquad \left( e^y \leq 1 + y + y^2, \text{ for all } y \leq 1 \right) \tag{515}$$

$$= 1 + y \qquad (y < 0) \tag{516}$$

$$\geq 1 - \epsilon, \qquad (y \in [-\epsilon, 0)) \tag{517}$$

Note that, for any $y < 0$, we have, $g_p(y)$ is monotonically increasing over $p$, since

$$g_p(y)^{-1} = p \cdot y + \frac{y}{e^y - 1} \tag{518}$$

is monotonically decreasing over $p$.

Therefore, we have, any fixed $p \in (0, 1]$, and any fixed $\epsilon \in (0, 1]$, for all $y \in [-\epsilon, 0)$,

$$1 - \epsilon \leq g_0(y) \qquad \text{(by Eq. (514))} \tag{519}$$

$$\leq g_p(y) \qquad (g_p(y) \text{ is monotonically increasing over } p) \tag{520}$$

$$\leq g_1(y) \tag{521}$$

$$\leq 1 + \epsilon, \qquad \text{(by Eq. (509))} \tag{522}$$

Note that,

$$f_p(y) = \frac{e^y - 1}{e^y + \frac{1-p}{p}} \qquad \text{(by Eq. (485))} \tag{523}$$

$$= \frac{e^y - 1}{p \cdot y \cdot (e^y - 1) + y} \cdot p \cdot y \qquad (p \in (0, 1], \epsilon \in (0, 1], \text{ and } y \in [-\epsilon, 0)) \tag{524}$$

$$= g_p(y) \cdot p \cdot y. \qquad \text{(by Eq. (488))} \tag{525}$$

Therefore, according to Eqs. (519) and (523), we have,

$$(1 + \epsilon) \cdot p \cdot y \leq f_p(y) \leq (1 - \epsilon) \cdot p \cdot y, \qquad (p \cdot y < 0) \tag{526}$$

which means any fixed $p \in (0, 1]$, and any fixed $\epsilon \in (0, 1]$, Eq. (487) holds for all $y \in [-\epsilon, 0)$. $\quad\square$

**Lemma 16.** *Let $r \in [0,1]^K$ and $a^* := \arg\max_{a \in [K]} r(a)$ be the optimal action. Denote $\Delta := r(a^*) - \max_{a \neq a^*} r(a)$ as the reward gap of $r$. We have, for any policy $\pi$,*

$$\sum_{i=1}^{K} \pi(i)^2 \cdot \left| r(i) - \pi^\top r \right|^3 \geq \frac{\Delta}{K-1} \cdot \pi(a^*)^2 \cdot \left( r(a^*) - \pi^\top r \right)^2. \tag{527}$$

*Proof.* **First case.** If $\pi^\top r \leq \max_{a \neq a^*} r(a)$, then we have,

$$r(a^*) - \pi^\top r \geq r(a^*) - \max_{a \neq a^*} r(a) = \Delta. \tag{528}$$

Therefore, we have,

$$\sum_{i=1}^{K} \pi(i)^2 \cdot \left| r(i) - \pi^\top r \right|^3 \geq \pi(a^*)^2 \cdot \left| r(a^*) - \pi^\top r \right|^3 \qquad \text{(fewer terms)} \tag{529}$$

$$\geq \pi(a^*)^2 \cdot \left( r(a^*) - \pi^\top r \right)^2 \cdot \Delta \qquad \text{(by Eq. (528))} \tag{530}$$

$$\geq \frac{\Delta}{K-1} \cdot \pi(a^*)^2 \cdot \left( r(a^*) - \pi^\top r \right)^2. \qquad (K \geq 2) \tag{531}$$

**Second case.** If $\pi^\top r > \max_{a \neq a^*} r(a)$, then we have, for all $a \neq a^*$,

$$\pi^\top r - r(a) \geq \pi^\top r - \max_{a \neq a^*} r(a) > 0. \tag{532}$$

Therefore, we have,

$$\sum_{i=1}^{K} \pi(i)^2 \cdot \left| r(i) - \pi^\top r \right|^3 = \pi(a^*)^2 \cdot \left( r(a^*) - \pi^\top r \right)^3 + \sum_{a \neq a^*} \pi(a)^2 \cdot \left( \pi^\top r - r(a) \right)^3. \tag{533}$$

Note that,

$$\pi(a^*) \cdot \left( r(a^*) - \pi^\top r \right) = \underbrace{\sum_{i=1}^{K} \pi(i) \cdot \left( r(i) - \pi^\top r \right)}_{=0} - \sum_{a \neq a^*} \pi(a) \cdot \left( r(a) - \pi^\top r \right) \tag{534}$$

$$= \sum_{a \neq a^*} \pi(a) \cdot \left( \pi^\top r - r(a) \right). \tag{535}$$

Next, we have,

$$\sum_{a \neq a^*} \pi(a)^2 \cdot \left( \pi^\top r - r(a) \right)^3 \geq \left( \pi^\top r - \max_{a \neq a^*} r(a) \right) \cdot \sum_{a \neq a^*} \pi(a)^2 \cdot \left( \pi^\top r - r(a) \right)^2 \qquad \text{(by Eq. (532))} \tag{536}$$

$$\geq \frac{\pi^\top r - \max_{a \neq a^*} r(a)}{K-1} \cdot \left[ \sum_{a \neq a^*} \pi(a) \cdot \left( \pi^\top r - r(a) \right) \right]^2 \qquad \text{(by Cauchy–Schwarz)} \tag{537}$$

$$= \frac{\pi^\top r - \max_{a \neq a^*} r(a)}{K-1} \cdot \pi(a^*)^2 \cdot \left( r(a^*) - \pi^\top r \right)^2. \qquad \text{(by Eq. (534))} \tag{538}$$

Combining Eqs. (533) and (536), we have,

$$\sum_{i=1}^{K} \pi(i)^2 \cdot \left| r(i) - \pi^\top r \right|^3 \geq \pi(a^*)^2 \cdot \left( r(a^*) - \pi^\top r \right)^3 + \frac{\pi^\top r - \max_{a \neq a^*} r(a)}{K-1} \cdot \pi(a^*)^2 \cdot \left( r(a^*) - \pi^\top r \right)^2 \tag{539}$$

$$\geq \left[ \frac{r(a^*) - \pi^\top r}{K-1} + \frac{\pi^\top r - \max_{a \neq a^*} r(a)}{K-1} \right] \cdot \pi(a^*)^2 \cdot \left( r(a^*) - \pi^\top r \right)^2 \qquad (K \geq 2) \tag{540}$$

$$= \frac{r(a^*) - \max_{a \neq a^*} r(a)}{K-1} \cdot \pi(a^*)^2 \cdot \left( r(a^*) - \pi^\top r \right)^2 \tag{541}$$

$$= \frac{\Delta}{K-1} \cdot \pi(a^*)^2 \cdot \left( r(a^*) - \pi^\top r \right)^2. \tag{542}$$

Combining Eqs. (529) and (539) we finish the proofs. $\qquad\square$

**Lemma 17** (Performance difference lemma [12]). *For any policies $\pi$ and $\pi'$,*

$$V^{\pi'}(\rho) - V^{\pi}(\rho) = \frac{1}{1-\gamma} \cdot \sum_s d_\rho^{\pi'}(s) \cdot \sum_a (\pi'(a|s) - \pi(a|s)) \cdot Q^{\pi}(s,a) \tag{543}$$

$$= \frac{1}{1-\gamma} \cdot \sum_s d_\rho^{\pi'}(s) \cdot \sum_a \pi'(a|s) \cdot A^{\pi}(s,a). \tag{544}$$

*Proof.* According to the definition of value function,

$$V^{\pi'}(s) - V^{\pi}(s) = \sum_a \pi'(a|s) \cdot Q^{\pi'}(s,a) - \sum_a \pi(a|s) \cdot Q^{\pi}(s,a) \tag{545}$$

$$= \sum_a \pi'(a|s) \cdot \left( Q^{\pi'}(s,a) - Q^{\pi}(s,a) \right) + \sum_a (\pi'(a|s) - \pi(a|s)) \cdot Q^{\pi}(s,a) \tag{546}$$

$$= \sum_a (\pi'(a|s) - \pi(a|s)) \cdot Q^{\pi}(s,a) + \gamma \cdot \sum_a \pi'(a|s) \cdot \sum_{s'} P(s'|s,a) \cdot \left[ V^{\pi'}(s') - V^{\pi}(s') \right] \tag{547}$$

$$= \frac{1}{1-\gamma} \cdot \sum_{s'} d_s^{\pi'}(s') \cdot \sum_{a'} (\pi'(a'|s') - \pi(a'|s')) \cdot Q^{\pi}(s',a') \tag{548}$$

$$= \frac{1}{1-\gamma} \cdot \sum_{s'} d_s^{\pi'}(s') \cdot \sum_{a'} \pi'(a'|s') \cdot (Q^{\pi}(s',a') - V^{\pi}(s')) \tag{549}$$

$$= \frac{1}{1-\gamma} \cdot \sum_{s'} d_s^{\pi'}(s') \cdot \sum_{a'} \pi'(a'|s') \cdot A^{\pi}(s',a'). \qquad \square$$

**Lemma 18.** *Let $u_t \in (0,1)$ for all $t \geq 1$. The infinite product $\prod_{t=1}^{\infty}(1-u_t)$ converges to a positive value if and only if the series $\sum_{t=1}^{\infty} u_t$ converges to a finite value.*

*Proof.* See [21, Lemma 16]. We include a proof for completeness.

Define the following partial products and partial sums,

$$p_T := \prod_{t=1}^{T}(1-u_t), \tag{550}$$

$$s_T := \sum_{t=1}^{T} u_t. \tag{551}$$

Since $p_T$ is monotonically decreasing and non-negative, the infinite product converges to positive values, i.e.,

$$\prod_{t=1}^{\infty}(1-u_t) = \lim_{T\to\infty} \prod_{t=1}^{T}(1-u_t) = \lim_{T\to\infty} p_T > 0, \tag{552}$$

if and only if $p_T$ is lower bounded away from zero (boundedness convergence criterion for monotone sequence) [15, p. 80].

Similarly, since $s_T$ is monotonically increasing, the series converges to finite values, i.e.,

$$\sum_{t=1}^{\infty} u_t = \lim_{T\to\infty} \sum_{t=1}^{T} u_t = \lim_{T\to\infty} s_T < \infty, \tag{553}$$

if and only if $s_T$ is upper bounded.

**First part.** $\prod_{t=1}^{\infty}(1-u_t)$ converges to a positive value only if $\sum_{t=1}^{\infty} u_t$ converges to a finite value.

Suppose $\prod_{t=1}^{\infty}(1-u_t)$ converges to a positive value. We have, for all $T \geq 1$,

$$q_T \geq q > 0. \tag{554}$$

Then we have,

$$q \leq q_T \tag{555}$$

$$= \exp\left\{\log\left(\prod_{t=1}^{T}(1 - u_t)\right)\right\} \tag{556}$$

$$= \exp\left\{\sum_{t=1}^{T}\log(1 - u_t)\right\} \tag{557}$$

$$\leq \exp\left\{-\sum_{t=1}^{T}u_t\right\} \qquad (\log(1-x) < -x) \tag{558}$$

$$= \exp\{-s_T\}, \tag{559}$$

which implies that,

$$s_T \leq -\log q < \infty. \tag{560}$$

Therefore, we have $\sum_{t=1}^{\infty}u_t$ converges to a finite value.

**Second part.** $\prod_{t=1}^{\infty}(1 - u_t)$ converges to a positive value if $\sum_{t=1}^{\infty}u_t$ converges to a finite value.

Suppose $\sum_{t=1}^{\infty}u_t$ converges to a finite value. Then we have, $u_t \to 0$ as $t \to \infty$. There exists a finite number $t_0 \geq 1$, such that for all $t \geq t_0$, we have $u_t \leq 1/2$. Also, we have, for all $T \geq 1$,

$$s_T \leq s < \infty. \tag{561}$$

Then we have,

$$\prod_{t=t_0}^{T}(1 - u_t) = \exp\left\{\sum_{t=t_0}^{T}\log(1 - u_t)\right\} \tag{562}$$

$$\geq \exp\left\{-\sum_{t=t_0}^{T}2 \cdot u_t\right\} \qquad (-2 \cdot x \leq \log(1-x) \text{ for all } x \in [0, 1/2]) \tag{563}$$

$$= \exp\{-2 \cdot s_T\}, \tag{564}$$

which implies that, for all large enough $T \geq 1$,

$$q_T = \left(\prod_{t=1}^{t_0-1}(1 - u_t)\right) \cdot \left(\prod_{t=t_0}^{T}(1 - u_t)\right) \tag{565}$$

$$\geq \left(\prod_{t=1}^{t_0-1}(1 - u_t)\right) \cdot \exp\{-2 \cdot s_T\} \tag{566}$$

$$\geq \left(\prod_{t=1}^{t_0-1}(1 - u_t)\right) \cdot \exp\{-2 \cdot s\} \tag{567}$$

$$> 0. \tag{568}$$

Therefore, we have $\prod_{t=1}^{\infty}(1 - u_t)$ converges to a positive value. $\qquad\square$

**Lemma 19.** *Let $u_t \in (0, 1)$ for all $t \geq 1$. We have $\prod_{t=1}^{\infty}(1 - u_t) = \lim_{T\to\infty}\prod_{t=1}^{T}(1 - u_t) = 0$ if and only if the series $\sum_{t=1}^{\infty}u_t$ diverges to positive infinity.*

*Proof.* See [21, Lemma 17]. We include a proof for completeness.

**First part.** $\prod_{t=1}^{\infty}(1 - u_t)$ diverges to 0 only if $\sum_{t=1}^{\infty}u_t$ diverges to positive infinity.

Suppose $\prod_{t=1}^{\infty}(1 - u_t)$ diverges to 0. According to Lemma 18, $\sum_{t=1}^{\infty}u_t$ diverges. And since the partial sum $s_T := \sum_{t=1}^{T}u_t$ is monotonically increasing, we have $\sum_{t=1}^{\infty}u_t$ diverges to positive infinity.

**Second part.** $\prod_{t=1}^{\infty}(1-u_t)$ diverges to 0 if $\sum_{t=1}^{\infty}u_t$ diverges to a positive infinity.

Suppose $\sum_{t=1}^{\infty}u_t$ diverges to positive infinity. According to Lemma 18, $\prod_{t=1}^{\infty}(1-u_t)$ diverges. And since the partial product $q_T := \prod_{t=1}^{T}(1-u_t)$ is non-negative and monotonically decreasing, we have $\prod_{t=1}^{\infty}(1-u_t)$ diverges to 0. $\qquad\square$

**Lemma 20** (Smoothness). *Let $\pi_\theta = \mathrm{softmax}(\theta)$ and $\pi_{\theta'} = \mathrm{softmax}(\theta')$. For any $r \in (0,1]^K$, for any $\pi_\theta(a)$, we have $\theta \mapsto \pi_\theta(a)$ is $3/2$-smooth, i.e.,*

$$\left| \pi_{\theta'}(a) - \pi_\theta(a) - \left\langle \frac{d\pi_\theta(a)}{d\theta}, \theta' - \theta \right\rangle \right| \le \frac{3}{4} \cdot \|\theta' - \theta\|_2^2. \tag{569}$$

*Proof.* The proof is based on and improves [24, Lemma 2].

Let $S := S(r,\theta) \in \mathbb{R}^{K \times K}$ be the second derivative of the value map $\theta \mapsto \pi_\theta(a) = \pi_\theta^\top \mathbf{1}_a$, where

$$\mathbf{1}_a(i) = \begin{cases} 1, & \text{if } i = a, \\ 0, & \text{otherwise.} \end{cases} \tag{570}$$

By Taylor's theorem, it suffices to show that the spectral radius of $S$ (regardless of $r$ and $\theta$) is bounded by $3/2$. Now, by its definition we have

$$S = \frac{d}{d\theta}\left\{ \frac{d\pi_\theta^\top \mathbf{1}_a}{d\theta} \right\} \tag{571}$$

$$= \frac{d}{d\theta}\left\{ (\mathrm{diag}(\pi_\theta) - \pi_\theta\pi_\theta^\top)\mathbf{1}_a \right\}. \tag{572}$$

Continuing with our calculation fix $i, j \in [K]$. Then,

$$S_{i,j} = \frac{d\{\pi_\theta(i) \cdot (\mathbf{1}_a(i) - \pi_\theta^\top \mathbf{1}_a)\}}{d\theta(j)} \tag{573}$$

$$= \frac{d\pi_\theta(i)}{d\theta(j)} \cdot (\mathbf{1}_a(i) - \pi_\theta^\top \mathbf{1}_a) + \pi_\theta(i) \cdot \frac{d\{\mathbf{1}_a(i) - \pi_\theta^\top \mathbf{1}_a\}}{d\theta(j)} \tag{574}$$

$$= (\delta_{ij}\pi_\theta(j) - \pi_\theta(i)\pi_\theta(j)) \cdot (\mathbf{1}_a(i) - \pi_\theta^\top \mathbf{1}_a) - \pi_\theta(i) \cdot (\pi_\theta(j)\mathbf{1}_a(j) - \pi_\theta(j)\pi_\theta^\top \mathbf{1}_a) \tag{575}$$

$$= \delta_{ij}\pi_\theta(j) \cdot (\mathbf{1}_a(i) - \pi_\theta^\top \mathbf{1}_a) - \pi_\theta(i)\pi_\theta(j) \cdot (\mathbf{1}_a(i) - \pi_\theta^\top \mathbf{1}_a) - \pi_\theta(i)\pi_\theta(j) \cdot (\mathbf{1}_a(j) - \pi_\theta^\top \mathbf{1}_a), \tag{576}$$

where

$$\delta_{ij} = \begin{cases} 1, & \text{if } i = j, \\ 0, & \text{otherwise,} \end{cases} \tag{577}$$

is Kronecker's $\delta$-function. To show the bound on the spectral radius of $S$, pick $y \in \mathbb{R}^K$. Then,

$$\left| y^\top S y \right| = \left| \sum_{i=1}^{K} \sum_{j=1}^{K} S_{i,j} y(i) y(j) \right| \tag{578}$$

$$= \left| \sum_i \pi_\theta(i)(\mathbf{1}_a(i) - \pi_\theta^\top \mathbf{1}_a)y(i)^2 - 2\sum_i \pi_\theta(i)(\mathbf{1}_a(i) - \pi_\theta^\top \mathbf{1}_a)y(i) \sum_j \pi_\theta(j)y(j) \right| \tag{579}$$

$$= \left| \left( (\mathrm{diag}(\pi_\theta) - \pi_\theta\pi_\theta^\top)\mathbf{1}_a \right)^\top (y \odot y) - 2 \cdot \left( (\mathrm{diag}(\pi_\theta) - \pi_\theta\pi_\theta^\top)\mathbf{1}_a \right)^\top y \cdot \left( \pi_\theta^\top y \right) \right| \tag{580}$$

$$\le \left\| (\mathrm{diag}(\pi_\theta) - \pi_\theta\pi_\theta^\top)\mathbf{1}_a \right\|_\infty \cdot \|y \odot y\|_1 + 2 \cdot \left\| (\mathrm{diag}(\pi_\theta) - \pi_\theta\pi_\theta^\top)\mathbf{1}_a \right\|_1 \cdot \|y\|_\infty \cdot \|\pi_\theta\|_1 \cdot \|y\|_\infty \tag{581}$$

$$\le \left\| (\mathrm{diag}(\pi_\theta) - \pi_\theta\pi_\theta^\top)\mathbf{1}_a \right\|_\infty \cdot \|y\|_2^2 + 2 \cdot \left\| (\mathrm{diag}(\pi_\theta) - \pi_\theta\pi_\theta^\top)\mathbf{1}_a \right\|_1 \cdot \|y\|_2^2 \tag{582}$$

$$\le 3 \cdot \left\| (\mathrm{diag}(\pi_\theta) - \pi_\theta\pi_\theta^\top)\mathbf{1}_a \right\|_1 \cdot \|y\|_2^2, \tag{583}$$

where $\odot$ is Hadamard (component-wise) product, and the third last inequality uses Hölder's inequality together with the triangle inequality, and the second inequality uses $\|y \odot y\|_1 = \|y\|_2^2$, $\|\pi_\theta\|_1 = 1$, and $\|y\|_\infty \le \|y\|_2$. Next, we have,

$$\left\|(\mathrm{diag}(\pi_\theta) - \pi_\theta \pi_\theta^\top)\mathbf{1}_a\right\|_1 = \sum_i \pi_\theta(i) \cdot \left|\mathbf{1}_a(i) - \pi_\theta^\top \mathbf{1}_a\right| \tag{584}$$

$$= \pi_\theta(a) \cdot (1 - \pi_\theta(a)) + \pi_\theta(a) \cdot \sum_{i \ne a} \pi_\theta(i) \tag{585}$$

$$= 2 \cdot \pi_\theta(a) \cdot (1 - \pi_\theta(a)) \tag{586}$$

$$\le 1/2. \qquad (x \cdot (1-x) \le 1/4 \text{ for all } x \in [0,1]) \tag{587}$$

Therefore we have,

$$\left|y^\top S(r,\theta)y\right| \le 3 \cdot \left\|(\mathrm{diag}(\pi_\theta) - \pi_\theta \pi_\theta^\top)\mathbf{1}_a\right\|_1 \cdot \|y\|_2^2 \tag{588}$$

$$\le 3/2 \cdot \|y\|_2^2, \tag{589}$$

finishing the proof. $\qquad\square$