# OpenReview forum: "The Role of Baselines in Policy Gradient Optimization"
_NeurIPS.cc/2022/Conference — NeurIPS 2022 Accept_

### Official Review · Reviewer_24yz · 2022-07-06

**Rating:** 5
**Confidence:** 3
**Soundness:** 2 fair
**Presentation:** 1 poor
**Contribution:** 2 fair

**Summary:**

The authors consider the well-known policy gradient algorithm (PG) of reinforcement learning
and study the effect of the baselines, which is frequently used in practice with PG to improve the sample efficiency and the stability.
To this end, the authors study the asymptotic behavior of the exemplary algorithm, namely
the softmax policy + on-policy importance sampling + the natural policy gradient (NPG), with and without the baseline.
As a result, it is shown that NPG + baseline is sufficient to achieve the convergence to global optimum, which is not possible with the NPG only.
Moreover, the authors show that the convergence rate achieved with the baseline is O(1/t) in expectation, which is not improvable without compromising the global convergence property.
They also give intuitive explanation why baselines lead to global convergence while pointing out a flaw of the conventional explanation via variance reduction.
Finally, these theoretical results (the global convergence and the convergence rate) are confirmed with numerical simulations.


**Questions:**

Please address the above questions.

**Limitations:**

I think some aspects of the limitation are further addressed by answering the above questions.


**Strengths And Weaknesses:**

The strength of this work is that it gives one clear explanation on the effectiveness of the commonly-used baseline technique and thus promotes our understanding of reinforcement learning.
However, there are some concerns with the clarity of the technical details and the presentation style:

- In the stochastic NL inequalities, a variable learning rate (eq. 4) is used and its dependency on $\pi\_{\theta\_t}(a\_t)$ seems important there to reduce agressiveness of the updates. How can it be adopted by the fixed-learning-rate regime of Update 2?
- In Theorem 1 and 2, there seems no assumption on the learning rate $\eta$, but at least it should satisfy $\eta>0$. Is that all?
- With regard to Remark 1, it reads for me that Theorem 1 and 2 are only valid with the noiseless bandit feedback model. Is it true? If so, it should be clarified in the statement of the theorems.
- The problematic agressiveness of Update 1 seems to be largely attributed to the computation of the natural gradient rather than the absence of baseline. Could you discuss the benefits of baseline compared to just computing the naive softmax policy gradient?
- What exactly is your assumption on the distribution of $\hat{Q}$ in Algorithm 1? How can we estimate the Q function to satisfy it?
- The explanation with Section 3 focuses too much on narrative intuitions and seems to be disconnected with the theoretical results presented in Section 2. Organizing these explanations in a form of more structured proof sketch would be very helpful to make the paper more coherent as a whole. Specifically, how are those lemmas in Section 3 connected with the proof of Theorem 1 and 2?

---

> ### Author Response · Authors · 2022-08-02
> **Response to Reviewer 24yz: part 1**
>
> We would like to thank the reviewer for carefully reading the paper, checking the technical details, and their valuable comments. The main concerns are addressed below.
>
> > In the stochastic NL inequalities, a variable learning rate (eq. 4) is used and its dependency on $\pi_{\theta_t}(a_t)$ seems important there to reduce agressiveness of the updates. How can it be adopted by the fixed-learning-rate regime of Update 2?
>
> **First**, as mentioned in Remark 1, if the true mean reward ($Q$-value) is observed for sampled actions (the results can be generalized to whenever the estimator of reward/value of sampled action is accurate enough, such that the error is smaller than gaps between different actions' reward/value), then using constant learning rates $\eta \in \Theta(1)$ will also achieve similar progress, which is sufficient to establish the $O(1/t)$ rate for Update 2. This aspect of the results is shown in Sections E and F in the appendix.
>
> On the other hand, even with the true mean reward/value observable for sampled actions, Update 1 still fails because of being too aggressive (Proposition 1 and Lemma 6).
>
> The above results show that the only difference between using a baseline or not (learning rates are both constants) is the aggressiveness of resulting updates.
>
> **Second**, if noisy rewards/values are observed for sampled actions (Assumption 1), then we can construct a simple example (2-armed bandit) to show that, with any constant learning rate $\eta \in \Theta(1)$, Update 2 will not converge to $\pi^*$. Therefore, a diminishing learning rate like Eq. (4) is necessary for converging toward $\pi^*$ asymptotically.
>
> The reason is because the reward noise makes the progress negative whenever $\pi(a^*) \approx 1$, i.e., Lemma 1 establishing positive progress cannot hold when $\pi_t$ is close enough to $\pi^*$. As a result, $\pi_t \not\to \pi^*$ with $\eta \in \Theta(1)$ in Update 2, since Corollary 1 no longer holds, and $\pi_t$ will wander forever (it will get close enough to $\pi^*$, then move far away because of negative progress, then get close again due to positive progress).
>
> However, the above results hold for the phase of ''approaching $\pi^*$'', which is at the final stage of convergence and is different from ''avoid approaching sub-optimal deterministic policies'', which can happen in the early stage of optimization (as illustrated by the plateau in Figure 1(a)).
>
> **Third**, The root technical reason for using Eq. (4) arises from the non-linear softmax transform, which ''amplifies'' the noisy sampled reward signal. In particular, the expected progress after one update in a one-state MDPs is equal to Eq. (27) in the appendix, and if $x = r(i)$ (sampled reward $x$ is the true mean reward $r(i)$), then the progress is always non-negative ($( e^{c \cdot y} - 1) \cdot y \ge 0$ for all $c > 0$ and $y \in \mathbb{R}$). However, if the sampled reward is noisy, then $( e^{c \cdot x} - 1) \cdot y$ is not necessarily non-negative (since the sampled reward can be $x < 0$ while the true mean reward can be $y > 0$), which means Lemma 1 and Corollary 1 cannot hold and $\pi_t$ will wander because the exponential function amplifies the noise. To make Lemma 1 still hold, it is then necessary to use Lemma 11, which requires the learning rate to scale with the policy.
>
> **To summarize**, the learning rate in Eq. (4) for noisy observations is needed to ensure positive progress (to avoid the policy oscillating/not approaching $\pi^*$) because of the bad interaction between the non-linear softmax transform and the reward noise.  The results when observing the true mean reward/value of sampled actions make the case that it is the baseline (learning rates are both constants) that is the key to reducing the aggressiveness of updates.
>
> >In Theorem 1 and 2, there seems no assumption on the learning rate $\eta$, but at least it should satisfy $\eta > 0$. Is that all?
>
> Sorry for the oversight. In Theorem 1, $\eta$ is the same as in Lemma 1 (Eq. (4)), and in Theorem 2, $\eta$ is the same as in Lemma 3 (Eq. (13)).

---

> > ### Author Response · Authors · 2022-08-02
> > **Response to Reviewer 24yz: part 2**
> >
> > >With regard to Remark 1, it reads for me that Theorem 1 and 2 are only valid with the noiseless bandit feedback model. Is it true? If so, it should be clarified in the statement of the theorems.
> >
> > It is not true that "Theorem 1 and 2 are only valid with the noiseless bandit feedback model".
> >
> > **First**, as mentioned in our response to the first question above, Theorems 1 and 2 are valid for noisy observed rewards/values for sampled actions (but the noise is bounded as in Assumption 1). In this case, as also mentioned above, a diminishing learning rate like Eq. (4) is necessary to make $\pi_t$ not oscillate, since the non-linear softmax transform can interact badly with the reward noise.
> >
> > **Second**, for a noiseless bandit feedback model (observing the true mean reward for sampled actions) and MDPs (observing the current policy's true $Q$-value for sampled actions), similar results to Lemmas 1 and 3 are shown in Sections E and F in the appendix. In fact, those results are easier to prove for mentioned reasons ($( e^{c \cdot y} - 1) \cdot y \ge 0$ for all $c > 0$ and $y \in \mathbb{R}$), and they hold for constant learning rates of $\eta \in \Theta(1)$. The corresponding results to Lemmas 2 and 4, and Theorems 1 and 2 are much easier to argue for noiseless observations, mainly consisting of partial arguments from the noisy observation cases. We will include this in the revision for completeness.
> >
> > **To summarize**, Theorems 1 and 2 are valid for noisy observations, while for noiseless observations corresponding results are shown in Sections E and F in the appendix.
> >
> > >The problematic aggressiveness of Update 1 seems to be largely attributed to the computation of the natural gradient rather than the absence of baseline. Could you discuss the benefits of baseline compared to just computing the naive softmax policy gradient?
> >
> > The naive softmax policy gradient converges at a $O(1/\sqrt{t})$ rate if using on-policy stochastic PG. This is shown in two previous works that motivate this study (i.e., [3, 11] cited in the introduction). The analysis for the naive stochastic softmax policy gradient is standard (unbiased PG with bounded variance). Our results for using a baseline in NPG are faster $O(1/t)$ based on using new techniques (that have to cope with unbounded variance).
> >
> > >What exactly is your assumption on the distribution of $\hat{Q}$ in Algorithm 1? How can we estimate the Q function to satisfy it?
> >
> > Sorry for not making this clear. The assumption on $\hat{Q}$ in Algorithm 1 is similar to the assumption on reward distribution in Assumption 1, as shown in Eq. (236) in the proofs of Lemma 3. A Monte-Carlo estimator using on-policy trajectories will satisfy this assumption.  We will state this clearly as an assumption in the appendix.

---

> > > ### Author Response · Authors · 2022-08-02
> > > **Response to Reviewer 24yz: part 3**
> > >
> > > >The explanation with Section 3 focuses too much on narrative intuitions and seems to be disconnected with the theoretical results presented in Section 2. Organizing these explanations in a form of more structured proof sketch would be very helpful to make the paper more coherent as a whole. Specifically, how are those lemmas in Section 3 connected with the proof of Theorem 1 and 2?
> > >
> > > We would like to thank the reviewer for suggesting organizing the explanations in the form of a more structured proof sketch, which will improve the presentation. We discuss the proof sketch to explain why the lemmas in Section 3 are connected with the proof of Theorem 1 and 2 as follows.
> > >
> > > **First**, the proofs for Theorems 1 and 2 are based on "non-negative progress" (Lemmas 1 and 3) and "progress does not vanish before approaching a globally optimal policy" (Lemmas 2 and 4). If any one of these two does not hold, then global convergence is not achieved. For example, if progress is not always non-negative (Lemmas 1 and 3 are not true, like using a constant $\eta \in \Theta(1)$ for noisy rewards, as mentioned in the response to the first question), then the policy will wander (Corollary 1 does not hold). If Lemmas 2 and 4 do not hold, then one can only conclude that convergence will occur toward locally optimal policies (which include but are not restricted to globally optimal policies), and the identity of the specific policy converged to will depend on initialization.
> > >
> > > **Second**, the "bad sampling" behaviour of Lemma 5 can make both the above two parts not hold. This can be shown by a simple example. Consider a one-state MDP with true mean reward $r = (1.0, 0.5, 0.1)^\top$, and $\pi_1 = (0.1, 0.4, 0.5)^\top$ such that $\pi_1^\top r = 0.35$. According to Proposition 1 and Lemma 6, using on-policy NPG without a baseline, it is possible to sample one action forever using on-policy sampling. If action $3$ is sampled forever, after sufficient time (i.e., $\pi_t^\top r \to 0.1$ as $t \to \infty$), Lemma 1 can no longer hold, because the progress is negative or expected reward $\pi_t^\top r$ is decreasing ($\pi_1^\top r = 0.35 > 0.1 = r(3)$). If action $2$ is sampled forever (and then $\pi_t^\top r \to 0.5$ as $t \to \infty$), then Lemma 2 cannot hold (but Lemma 1 holds). In this case, progress is non-negative ($\pi_1^\top r = 0.35 < 0.5 = r(2)$), but will vanish before $\pi_t$ approaches $\pi^* = (1, 0, 0)^\top$, and $\pi_t$ will converge to a sub-optimal deterministic policy $(0, 1, 0)^\top$. It is worth mentioning that action $1$ (the optimal action) also has a chance to be sampled forever (because a globally optimal policy is also a locally optimal policy), but one cannot assert that convergence to a globally optimal policy will occur almost surely.
> > >
> > > **Third**, the "good sampling" behaviour of Lemma 7 is then necessary to make both the `non-negative progress'' (Lemmas 1 and 3) and "progress does not vanish before approaching a globally optimal policy'' (Lemmas 2 and 4) properties hold, and Lemma 8 shows that using a baseline will achieve this sampling behaviour. The intuition of why this is true is as follows. Using the current baseline $\pi_t^\top r$, the actions can be partitioned into "good actions" (with true mean reward $r(a_g) > \pi_t^\top r$) and "bad actions" (with true mean reward $r(a_b) \le \pi_t^\top r$), as discussed in Line 499 and 509 in the proof of Lemma 1 in the appendix. **(a)** If bad actions have a chance to be sampled forever, then the expected reward will be decreasing, and thus expected progress will be negative (Lemma 1 does not hold). Therefore, it is necessary to have a sampling behaviour like "good sampling" in Lemma 7 (rather than the "bad sampling" behaviour of Lemma 5) to make Lemma 1 hold. **(b)** If some good (but sub-optimal) actions have a chance to be sampled forever, then it is possible to converge to a sub-optimal deterministic policy. Therefore, "good sampling" in Lemma 7 again becomes necessary to ensure that the optimal action has a chance to be always sampled, ensuring global rather than local convergence.
> > >
> > > **To summarize**, the "bad sampling" behaviour of Lemma 5 will make both the "non-negative progress" (Lemmas 1 and 3) and "progress does not vanish before approaching a globally optimal policy" (Lemmas 2 and 4) properties not hold.  Moreover, the "good sampling" behaviour of Lemma 7 is necessary to make both of them hold. The two properties are both necessary to prove the global convergence results (Theorems 1 and 2). We will discuss and organize related sketch/intuition in the paper to make the connection between Section 3 and Section 2 more clear. Thank you for your suggestions.

---

> > > > ### Comment · Reviewer_24yz · 2022-08-08
> > > > **Thank you for your response**
> > > >
> > > > Thank you for your response. It clarifies a lot for me.
> > > > In particular, the comparison of the proposed method to the naive softmax policy gradient helps me (as a non-expert) understand the significance of the results presented in the paper.
> > > >
> > > > However, on the use of the $\pi(\theta_t)$-dependent learning rate in Theorem 1 (and 2),  it makes me wonder if the fast O(1/t) rate should be attributed to the baseline. I understand decaying learning rate is necessary and common practice in the stochastic feedback case, but the $\pi(\theta_t)$-dependent learning rate is not standard and arguably complicated. Also, as Eq. 4 contains $r$, which is unknown in the stochastic setting, Theorem 1 (and 2) in the stochastic setting seems not applicable to practically implementable algorithms.

---

> > > > > ### Author Response · Authors · 2022-08-08
> > > > > **Further response**
> > > > >
> > > > > Dear Reviewer 24yz,
> > > > >
> > > > > Thank you for checking our previous feedback and for asking further questions!
> > > > >
> > > > > >However, on the use of the $\pi_{\theta_t}$-dependent learning rate in Theorem 1 (and 2), it makes me wonder if the fast O(1/t) rate should be attributed to the baseline. I understand decaying learning rate is necessary and common practice in the stochastic feedback case, but the $\pi_{\theta_t}$-dependent learning rate is not standard and arguably complicated.
> > > > >
> > > > > As mentioned in our response to another reviewer C7pU, we also consider making the learning rate practical an important question. Currently, we have a preliminary observation that the learning rate in retrospect is $\eta \in O(1/t)$ as mentioned in Remark 1, which will make it independent across actions and not arguably complicated.
> > > > >
> > > > > The above arguments are used in the proofs. In Eq. (213) in the proofs for Theorem 1 in the appendix, we show that the learning rate $\eta $ decays at the same rate as the sub-optimality gap $r(a^*) - \pi_{\theta_t}^\top r$. Since we proved that $r(a^*) - \pi_{\theta_t}^\top r \in O(1/t)$, it turns out that $\eta$ itself is in $O(1/t)$. This evidence shows that using constants over $t$ will likely also work for the same algorithm and can achieve the same convergence results, and we left a better analysis for our future study.
> > > > >
> > > > > >Also, as Eq. 4 contains $r$, which is unknown in the stochastic setting, Theorem 1 (and 2) in the stochastic setting seems not applicable to practically implementable algorithms.
> > > > >
> > > > > As the reviewer mentioned, the current learning rate (e.g., in Eq. (4)) does rely on $r$ because it contains $| r(a_t) - \pi_{\theta_t}^\top r| $. Our preliminary observation here is using $1 - \pi_{\theta_t}(a_t)$ will likely also work and making the algorithm truly practical. The intuition is $| r(a_t) - \pi_{\theta_t}^\top r| = | \sum_{a \not= a_t} \pi_{\theta_t}(a) \cdot (r(a_t) - r(a)) | \le 1 - \pi_{\theta_t}(a_t)$ since $r \in (0, 1]^K$. On the other hand, $| r(a_t) - \pi_{\theta_t}^\top r| \ge \Delta \cdot (1  - \pi_{\theta_t}(a_t) )$ if $a_t = a^*$ where $\Delta$ is the reward gap of $r$. This essentially means $1 - \pi_{\theta_t}(a_t)$ is a good approximation of $| r(a_t) - \pi_{\theta_t}^\top r| $ if the later is non-zero.  Moreover, if $| r(a_t) - \pi_{\theta_t}^\top r| $ is zero, then this falls into Line 516, Case (c) of "indifferent action" in the proof of Lemma 1 in the appendix, which contributes $0$ to the expected progress. We left this as an important follow-up work in the near future as mentioned in the conclusion section.
> > > > >
> > > > > **To summarize**, **first**, the observation of $| r(a_t) - \pi_{\theta_t}^\top r| $ can be well approximated by $1 - \pi_{\theta_t}(a_t)$ if we are interested in calculating the expected progress will make the learning rate $\pi_{\theta_t}(a_t) \cdot \left( 1 - \pi_{\theta_t}(a_t) \right)$, which is practical, but as the reviewer mentioned, could be still not standard. **Second**, the results in retrospect that the term $\pi_{\theta_t}(a_t) \left( 1 - \pi_{\theta_t}(a_t) \right)$ decays at the same rate of sub-optimality gap $r(a^*) - \pi_{\theta_t}^\top r$ makes it possible to use constants over $t$ to achieve the same results. Those preliminary observations provide evidences that our work can inspire and be valuable for important future study.

---

> > > > > > ### Comment · Reviewer_24yz · 2022-08-08
> > > > > > **Thank you**
> > > > > >
> > > > > > Thank you for your reply and detailed explanations.
> > > > > > I think all of my concerns are addressed by the authors.
> > > > > > I will update my score accordingly, expecting the authors to update their manuscript to reflect my review.

---

### Official Review · Reviewer_C7pU · 2022-07-06

**Rating:** 6
**Confidence:** 3
**Soundness:** 3 good
**Presentation:** 2 fair
**Contribution:** 3 good

**Summary:**

*The Role of Baselines in Policy Gradient Optimization* is a submission that studies the impact of substracting baselines in the natural policy gradient updates. While in expectation (or with exact gradients), both updates are equivalent, the authors show that their stochastic updates behave differently, in particular with respect to the aggressivity of convergence.

**Questions:**

Please address my questions in the previous section.

**Limitations:**

No potential negative societal impact.

**Strengths And Weaknesses:**

This is a well rounded study on a phenomenon that has not been investigated before. While I think that the paper deserves to be published, I am bothered by several points (this explains my low recommendation):
* The claims made in the abstract and the introduction are too strong: the convergence to global optimality relies on Assumption 2 in MDPs in order to ensure exploration of all states. This is true even with exact gradients, when the updates are online PG [Laroche2021].
* Aggressiveness of updates is not a well defined concept and the reader has not a good grasp of what the authors mean before Section 3. I would rather call it speed of convergence that prevent a proper exploration of actions (exploration of states being handled by Assumption 2).
* Update 2 relies on the knowledge of $r$? Same for the learning rate? How is it a practical algorithm?
* I don't like the learning rate to be dependent on $\theta_t$. I don't feel that the update should be called NPG anymore. (this was already something that bothered me in the escort paper [Mei2020]) Could you justify your choice to keep calling it NPG?
* Lemma 1 would hold with $\eta$ being defined lesser than the right hand side of eq (4), wouldn't it?
* Converging faster than $O(1/t)$ in a stochastic setting is not possible, otherwise MAB could be solved with regret in $O(1)$. The authors spend more than 1 page and 4 lemmas on it. Could you please help me understand better why your arguments in 3.2-3.5 are necessary?
* I find it very surprising that introducing a baseline (basically IS on the advantage rather than the value) allows to obtain this dampening of the convergence as the authors claim. Isn't it rather granted by the adaptive learning rate? Wouldn't the same result be obtainable without baseline? (I don't really buy the explanation in Section 3.5)

---

> ### Author Response · Authors · 2022-08-02
> **Response to Reviewer C7pU: part 1**
>
> We would like to thank the reviewer for appreciating our work and for carefully checking the details. The main concerns are addressed as follows.
>
> >The claims made in the abstract and the introduction are too strong: the convergence to global optimality relies on Assumption 2 in MDPs in order to ensure exploration of all states. This is true even with exact gradients, when the updates are online PG [Laroche2021].
>
> Thank you for pointing the related work [Laroche2021] to us, which will be cited. As the reviewer mentioned, Assumption 2 is necessary "even with exact gradients", which suggests this is a "weakness" for almost all PG based methods. Our understanding is that the root reason is from the policy gradient theorem [Sutton1999], which contains expectation over $d_\mu^\pi$. If $d_\mu^\pi(s)$ is small for a valuable state (could be since $\mu$ is bad, or $\pi$ is bad, or both), then that state $s$ will not get enough update using PG methods. From this perspective, it seems this is not a problem of our specific setting, but a general problem that all methods using the PG theorem are facing.
>
> >Aggressiveness of updates is not a well defined concept and the reader has not a good grasp of what the authors mean before Section 3. I would rather call it speed of convergence that prevent a proper exploration of actions (exploration of states being handled by Assumption 2).
>
> Sorry for not making this clear. We use the definition of "committal rate" (Definition 2 in [11]) to be the formal quantitative characterization of "aggressiveness of updates". Fixing the sampling procedure to be sampling one action forever (rather than on-policy sampling $a_t \sim \pi_t(\cdot)$), the committal rate of an update on one action is how fast the repeatedly sampled action's probability is approaching $1$.
>
> The above "committal rate" definition from [11] only characterizes updates (since the sampling is always fixed to be sampling one action forever). It does characterize the intuition of "speed of convergence that prevent a proper exploration of actions" as the reviewer mentioned.
>
> Using this definition, Lemma 6 basically says that NPG without a baseline is aggressive/over-committal, since the repeatedly sampled action's probability is becoming $1 - O(e^{- c \cdot t})$ for some $c > 0$. This means the update itself is aggressive and the consequence is $\prod_{t=1}^{\infty}{ \pi_{t}(a) } > 0$, which means there is a positive chance of sampling one action $a$ forever if we use on-policy sampling with this aggressive update.
>
> On the other hand, Lemma 8 shows that NPG with baseline (Update 2) is not too aggressive since the repeatedly sampled action's probability is becoming $1 - \Omega(1/t)$, which makes it not possible to sample one action forever since  $\prod_{t=1}^{\infty}{ \pi_{t}(a) } = 0$.
>
> >Update 2 relies on the knowledge of $r$? Same for the learning rate? How is it a practical algorithm?
>
> $\hat{r}_t$ in Update 2 does not rely on the knowledge of $r$, since $\hat{r}_t$ is from Definition 1, which uses sampled rewards.
>
> The current learning rate (e.g., in Eq. (4)) does rely on $r$ because it contains $| r(a_t) - \pi_{\theta_t}^\top r| $. Our preliminary observation here is using $1 - \pi_{\theta_t}(a_t)$ will likely also work and making the algorithm truly practical. The intuition is $| r(a_t) - \pi_{\theta_t}^\top r| = | \sum_{a \not= a_t} \pi_{\theta_t}(a) \cdot (r(a_t) - r(a)) | \le 1 - \pi_{\theta_t}(a_t)$ since $r \in (0, 1]^K$. On the other hand, $| r(a_t) - \pi_{\theta_t}^\top r| \ge \Delta \cdot (1  - \pi_{\theta_t}(a_t) )$ if $a_t = a^*$ where $\Delta$ is the reward gap of $r$. This essentially means $1 - \pi_{\theta_t}(a_t)$ is a good approximation of $| r(a_t) - \pi_{\theta_t}^\top r| $ if the later is non-zero.  Moreover, if $| r(a_t) - \pi_{\theta_t}^\top r| $ is zero, then this falls into Line 516, Case (c) of "indifferent action" in the proof of Lemma 1 in the appendix, which contributes $0$ to the expected progress. We left this as an important follow-up work in the near future as mentioned in the conclusion section.

---

> > ### Author Response · Authors · 2022-08-02
> > **Response to Reviewer C7pU: part 2**
> >
> > >I don't like the learning rate to be dependent on $\theta_t$. I don't feel that the update should be called NPG anymore. (this was already something that bothered me in the escort paper [Mei2020]) Could you justify your choice to keep calling it NPG?
> >
> > The escort paper [Mei2020] is using learning rates depend on $\theta_t$ for a different reason. Their Lemma 2 shows that the smoothness coefficient depends on $\theta_t$, and thus the learning rate (1 over smoothness) also depends on $\theta_t$.
> >
> > We have a preliminary observation that the learning rate in retrospect is $O(1/t)$ as mentioned in Remark 1, which will make it independent across actions, making the algorithm the same as the standard NPG.
> >
> > The above arguments are used in the proofs. In Eq. (213) in the proofs for Theorem 1 in the appendix, we show that the learning rate $\eta$ decays at the same rate as the sub-optimality gap $r(a^*) - \pi_{\theta_t}^\top r$. Since we proved that $r(a^*) - \pi_{\theta_t}^\top r \in O(1/t)$, it turns out that $\eta$ itself is in $O(1/t)$. This evidence shows that using constants over $t$ will likely also work for the same algorithm and can achieve the same convergence results, and we left a better analysis for our future study.
> >
> > >Lemma 1 would hold with $\eta$ being defined lesser than the right hand side of eq (4), wouldn't it?
> >
> > The reviewer is right. We will clearly mention this point. The technical reason is we use Lemma 11 (e.g., $y \in [0, \epsilon]$ in Eq. (436)) to control the non-linear softmax transform of noisy sampled reward (consider $y$ as $\eta$ times sampled reward). And therefore any smaller enough $\eta$ would work.
> >
> > >Converging faster than $O(1/t)$ in a stochastic setting is not possible, otherwise MAB could be solved with regret in $O(1)$. The authors spend more than 1 page and 4 lemmas on it. Could you please help me understand better why your arguments in 3.2-3.5 are necessary?
> >
> > We totally agree with the reviewer on "faster than $O(1/t)$ in a stochastic setting is not possible". However, as mentioned in the introduction, one motivation of this work is to understand why using baseline would help convergence, especially when the variance of estimator is unbounded in the on-policy NPG setting here. Section 2 presented the results and new techniques but did not clarify the mechanism of using baseline behind those technical lemmas (Lemmas 1-4).
> >
> > As also asked by another reviewer 24yz, we discuss more of the proof sketch to explain why the lemmas in Section 3 are connected with the proof of Theorem 1 and 2, and therefore why arguments in 3.2-3.5 are necessary below.
> >
> > **First**, the proofs for Theorems 1 and 2 are based on "non-negative progress" (Lemmas 1 and 3) and "progress does not vanish before approaching a globally optimal policy" (Lemmas 2 and 4). If any one of these two properties does not hold, then global convergence is not achievable. For example, if progress is not always non-negative (i.e., if Lemmas 1 and 3 are not true, like using constant $\eta \in \Theta(1)$ for a noisy reward as mentioned in the response to the first question), then policy will wander (Corollary 1 does not hold). If Lemmas 2 and 4 do not hold, then one can only conclude that convergence is toward locally optimal policies (including but not restricted to globally optimal policies) and the specific policy to which convergence occurs will depend on initialization.
> >
> > **Second**, the "bad sampling" behaviour of Lemma 5 will possibly make both the above two parts not hold. This can be shown by a simple example. Consider a one-state MDP with true mean reward $r = (1.0, 0.5, 0.1)^\top$, and $\pi_1 = (0.1, 0.4, 0.5)^\top$ such that $\pi_1^\top r = 0.35$. According to Proposition 1 and Lemma 6, using on-policy NPG without a baseline, it is possible to sample one action forever using on-policy sampling. If action $3$ is sampled forever, then after a sufficiently large time (and then $\pi_t^\top r \to 0.1$ as $t \to \infty$) Lemma 1 cannot hold, because the progress is negative or the expected reward $\pi_t^\top r$ is decreasing ($\pi_1^\top r = 0.35 > 0.1 = r(3)$). If action $2$ is sampled forever (and then $\pi_t^\top r \to 0.5$ as $t \to \infty$), then Lemma 2 does not hold (but Lemma 1 holds). In this case, the progress is non-negative ($\pi_1^\top r = 0.35 < 0.5 = r(2)$), but it will vanish before $\pi_t$ approaches $\pi^* = (1, 0, 0)^\top$, and $\pi_t$ will converge to a sub-optimal deterministic policy $(0, 1, 0)^\top$. It is worth mentioning that action $1$ (the optimal action) also has a chance to be sampled forever, since a globally optimal policy is also locally optimal, but one cannot say this will happen almost surely.

---

> > > ### Author Response · Authors · 2022-08-02
> > > **Response to Reviewer C7pU: part 3**
> > >
> > > **Third**, the "good sampling" behaviour of Lemma 7 is then necessary to make both the `non-negative progress'' (Lemmas 1 and 3) and "progress does not vanish before approaching a globally optimal policy'' (Lemmas 2 and 4) hold, and Lemma 8 shows that using a baseline will achieve this sampling behaviour. The intuition of why this is true is as follows. Using the current baseline $\pi_t^\top r$, all the actions can be partitioned into "good actions" (with true mean reward $r(a_g) > \pi_t^\top r$) and "bad actions" (with true mean reward $r(a_b) \le \pi_t^\top r$), as discussed in Line 499 and 509 in the proofs for Lemma 1 in the appendix. **(a)** If bad actions have a chance to be sampled forever, then the expected reward will be decreasing, and thus expected progress will be negative (Lemma 1 does not hold). Therefore, it is necessary to have a sampling behaviour like "good sampling" in Lemma 7 (rather than "bad sampling" behaviour of Lemma 5) to make Lemma 1 hold. **(b)** If some good (but sub-optimal) actions have a chance to be sampled forever, then it is possible to converge to sub-optimal deterministic policies. Therefore, "good sampling" in Lemma 7 again becomes necessary to make optimal action has a chance to be always sampled, ensuring global rather than local convergence.
> > >
> > > **To summarize**, the "bad sampling" behaviour of Lemma 5 will make both the "non-negative progress" (Lemmas 1 and 3) and "progress does not vanish before approaching a globally optimal policy" (Lemmas 2 and 4) properties not hold. And the "good sampling" behaviour of Lemma 7 is necessary to make both of them hold. These two properties are both necessary to prove the global convergence results (Theorems 1 and 2). We will discuss and organize related sketch/intuition in the paper to make the connection between Section 3 and Section 2 more clear.
> > >
> > > >I find it very surprising that introducing a baseline (basically IS on the advantage rather than the value) allows to obtain this dampening of the convergence as the authors claim. Isn't it rather granted by the adaptive learning rate? Wouldn't the same result be obtainable without baseline? (I don't really buy the explanation in Section 3.5)
> > >
> > > Another reviewer 24yz asked a similar question. **First**, as mentioned in Remark 1, if true mean reward ($Q$-value) is observed for the sampled action (the results can be generalized to whenever the estimator of reward/value of sampled action is accurate enough / smaller than reward/value gaps between actions), then using constant learning rate $\eta \in \Theta(1)$ will also achieve similar progresses, which are sufficient to establish $O(1/t)$ rate for Update 2. This part of results is shown in Sections E and F in the appendix.
> > >
> > > On the other hand, even with true mean reward/value observable sampled action, Update 1 still fails because of being too aggressive (Proposition 1).
> > >
> > > The above results show that the only difference of using baseline or not (learning rates are both constants) makes the results of updates different.
> > >
> > > **Second**, if noisy rewards/values are observed for sampled actions (Assumption 1), then we can construct a simple example (2-armed bandit) to show that, with any constant learning rate $\eta \in \Theta(1)$, Update 2 will not converge to $\pi^*$. Therefore, a diminishing learning rate like Eq. (4) is necessary for converging toward $\pi^*$ asymptotically.
> > >
> > > The reason is because the reward noise makes the progress negative whenever $\pi(a^*) \approx 1$, i.e., Lemma 1 establishing positive progress cannot hold when $\pi_t$ is close enough to $\pi^*$. As a result, $\pi_t \not\to \pi^*$ with $\eta \in \Theta(1)$ in Update 2, since Corollary 1 no longer holds, and $\pi_t$ will wander forever (it will get close enough to $\pi^*$, then move far away because of negative progress, then get close again due to positive progress).
> > >
> > > However, the above results hold for the phase of ''approaching $\pi^*$'', which is at the final stage of convergence and is different from ''avoid approaching sub-optimal deterministic policies'', which can happen in the early stage of optimization (as illustrated by the plateau in Figure 1(a)).
> > >
> > > **Third**, in noisy observation settings, without using baseline, using $O(1/\sqrt{t})$ learning rate can obtain a $O(1/\sqrt{t})$ rate, but this is for standard softmax policy gradient in [21, 11].
> > >
> > > **To summarize**, when observing the true mean reward/value of sampled actions, baseline (learning rates are both constants) is the key to reducing the aggressiveness of updates. While for noisy observations, adaptive learning rate is needed to ensure positive progress (to avoid the policy oscillating/not approaching $\pi^*$) because of the bad interaction between the non-linear softmax transform and the reward noise. While in noisy setting the existing results are $O(1/\sqrt{t})$ rates for standard softmax PG rather than NPG.

---

> > > > ### Comment · Reviewer_C7pU · 2022-08-07
> > > > **Thank you for the response**
> > > >
> > > > This was very informative. I keep my weak accept recommendation.

---

### Official Review · Reviewer_aDcz · 2022-07-11

**Rating:** 6
**Confidence:** 3
**Soundness:** 3 good
**Presentation:** 4 excellent
**Contribution:** 4 excellent

**Summary:**

This work studies the role of baselines in natural policy gradients (NPG). Earlier it has been argued that baselines are effective for reducing variance of the updates. They first study the one-state MDP. They argue theoretically that NPG updates without baseline might converge to sub-optimal action and it is not so with the baseline. More importantly they provide results on a.s. global convergence rates for the baseline case. They then extend these results to general MDP. They argue that the variance maybe unbounded in the NPG with baseline case and therefore it is unfair to draw conclusions from a variance reduction perspective alone. The explain the vicious and virtuous cycle of update aggressiveness. Finally, they provide experiments in a simplified setting to evaluate their theoretical results.

**Questions:**

**Questions and suggestions for the authors**:
1. I think "the variance is unbounded" and therefore "Baselines Do Not Control Update Variance" is a leap I am still not convinced of after reading the main body of the paper. I was wondering if the authors could comment on how the two values of interest in Proposition 3 ($\frac{r(a)^2}{\pi_{\theta}(a)}$ and $\frac{(r(a) - \pi_{\theta_t}^Tr)^2}{\pi_{\theta}(a)}$) evolve through time $t$ (and not only the convergence)? Please point me to the result if it already dealt with in the paper or previous work. This might give us an idea of how baselines effect the variance through time.

2.  In the abstract you claim that for the NPG setting "finite variance is not necessary" (line 15)  and in Section 3.1 you claim "there exists at least one action such that $\frac{r(a)^2}{\pi_{\theta}(a)}$ and $\frac{(r(a) - \pi_{\theta_t}^Tr)^2}{\pi_{\theta}(a)}$ become unbounded" (line 244). Can you please explain in what case the variance is finite? My reading is that this necessitates variance being unbounded.

3. I am curious why the optima is stuck in a plateau in Fig 2(a)? Kakade [4] had asserted that NPG overcomes the plateau phenomena, in principle, effectively. I hope the authors can comment on this.

4. Can you present empirical results for a more general MDP case as well? That would make the argument stronger although I wouldn't reject the paper for this reason since the results for single state example serve as a good example.

5. Some additional background on NPG in section 2 and how it differs from vanilla PG would be helpful for the reader. Update 2 being NPG and where it comes from might not be apparent to most readers in the community.

6. How is committal behavior different from the standard exploration exploitation problem? I am curious if some of the effects of the baseline are similar to that of adding noise (or policy gradient version of $\epsilon$-greedy exploration) to the policy?

Minor: Title mismatch between the submission and paper.

[4] A Natural Policy Gradient, Sham M. Kakade, NIPS.

**Limitations:**

Apart from the limitations mentioned above I don't see any limitations in terms of addressing the ethical guidelines.

**Strengths And Weaknesses:**

**Strengths**:
1. The paper is very well written and goes through various concepts very carefully despite it being dealing with topics that are very complicated.
2. The new perspective on update aggressiveness is both intuitive and technically sound from what I understand. The clear dichotomy between the "vicious circle" and "virtuous circle" of updates is deeply insightful and very well presented!
3. Their experiment on convergence rate illustrates the practical outcome of Theorem 1 very clearly.

**Weaknesses**:
1. My primary concern is that the authors have not very well positioned their work in light of past work, to the best of my knowledge. This hinders my understanding of how exactly their analysis is novel and stands apart from previous work. For example, Lemma 2 by Bhatnagar et al [1] is a well known result on how the value function is the minimum variance baseline (to the best of my knowledge) which is also for the more general linear setting. How is this different from your results or how is the setting different? My best guess is that lines 237-239 hint at this but I am unsure how this works in light of previous results. Also, the difference is that your work considers on-policy NPG? Also please see other citations [2,3].

2. My second concern is the lack of explanation of the constants in the convergence rate results. The constants $C$ in Theorems 1 and 2 are not explained. Are they the same or similar? I see that the constant does not depend on $t$ but as a reader I would like to know what this constant does depend on? What properties of the MDP effect these constants? Therefore, in my opinion, a corresponding empirical study of how this convergence rate changes with the MDP would also be warranted for a complete understanding of the theoretical results in addition to the explanations.

**Overall**: I am currently reluctant to reject the paper and would recommend an accept if my concerns above and below are addressed. This is primarily because the results formalise "committal behavior" and open up avenues for future research which could impact the practice of RL.

[1] Natural actor-critic algorithms, Shalabh Bhatnagar and Richard S. Sutton and Mohammad Ghavamzadeh and Mark Lee, 2009

[2] Variance Reduction Techniques for Gradient Estimates in Reinforcement Learning, Evan Greensmith and Peter L. Bartlett and Jonathan Baxter, 2004

[3] Variance Reduction for Policy Gradient with Action-Dependent Factorized Baselines, Cathy Wu and Aravind Rajeswaran and Yan Duan and Vikash Kumar and Alexandre M. Bayen and Sham M. Kakade and Igor Mordatch and P. Abbeel, 2018

---

> ### Author Response · Authors · 2022-08-02
> **Response to Reviewer aDcz: part 1**
>
> We would like to thank the reviewer for appreciating our contributions and for very carefully reading our paper. After reading the review, we think the main concerns are not from technical weaknesses, and they can be addressed by making clarifications as follows.
>
> >My primary concern is that the authors have not very well positioned their work in light of past work, to the best of my knowledge. This hinders my understanding of how exactly their analysis is novel and stands apart from previous work. For example, Lemma 2 by Bhatnagar et al [1] is a well known result on how the value function is the minimum variance baseline (to the best of my knowledge) which is also for the more general linear setting. How is this different from your results or how is the setting different? My best guess is that lines 237-239 hint at this but I am unsure how this works in light of previous results. Also, the difference is that your work considers on-policy NPG? Also please see other citations [2,3].
>
> We apologize for not making this point clear enough. We now discuss the difference between past work and our novelty as follows.
>
> We thank the reviewer for mentioning the well known result of Lemma 2 by Bhatnagar et al [1] and related works of [2] and [3], which will be cited. On the other hand, existing work ([1-3] as mentioned, as well as more recent works of [15, 22, 9, 21] already cited in the main paper) analyzed and focused on two settings: (i) true gradient setting; (ii) unbiased stochastic gradient with **bounded variance** of PG estimators.
>
> In practice, bounded variance is achieved by either sampling multiple (non-constant number of) data depending on the behavior policies ([9] cited in the main paper) or assuming good behavior policies are available. For example, in Algorithm 1 in Agarwal et al. ([1] cited in the main paper), it is assumed that the behaviour policy covers all actions, which can guarantee bounded variance of estimators from sampled data.
>
> The bounded variance setting works by controlling the noise of estimator and make the stochastic gradient "close enough" to true gradient updates, and its standard results look like "sub-optimality gap $\le O(1/\sqrt{t}) + $ variance" (e.g., Thm. 6.2 in Agarwal et al. [1] in the paper), which requires the variance term to be uniformly bounded by constants across the whole probability simplex. Otherwise, the upper bound will be vacuous (sub-optimality gap $\le \infty$).
>
> Consider on-policy setting we studied, where in each iteration one action is sampled (results also apply to other constant batch size of $\Theta(1)$), the variance of NPG estimator is unbounded as shown in Proposition 3. The above results of bounded variance are not applicable.
>
> To summarize, **first**, existing results to our knowledge are like "sub-optimality gap $\le O(1/\sqrt{t}) + $ variance", which requires variance of PG estimators to be uniformly bounded by constants, which is not applicable in our on-policy NPG setting, and therefore we proposed novel techniques (Lemmas 1-4) to do analyses. **Second**, the on-policy setting we considered is also more practical (comparing to either sampling multiple data or assuming away the difficulty by using good behavior policies). Losing the bounded variance properties created difficulties for analysis, making not only our results but also the techniques necessarily novel.
>
> >My second concern is the lack of explanation of the constants in the convergence rate results. The constants $C$ in Theorems 1 and 2 are not explained. Are they the same or similar? I see that the constant does not depend on $t$ but as a reader I would like to know what this constant does depend on? What properties of the MDP effect these constants? Therefore, in my opinion, a corresponding empirical study of how this convergence rate changes with the MDP would also be warranted for a complete understanding of the theoretical results in addition to the explanations.
>
> This is also an important question to us. Currently, we do not have a quantitative characterization for constants $C$ in Theorems 1 and 2, and some evidences and intuitive understandings are as follows.
>
> **First**, the constants $C$ in Theorems 1 and 2 are similar but not the same, since general MDPs contain multiple states, where state distribution will have an effect.
>
> **Second**, the constant $C$ in Theorem 1 depends on problems (including reward gap of $r$, and noise level of reward distribution) and initialization. Figure 1(a) used adversarial initializations ($\pi_{\theta_1}(a^*) \approx 0.006$ as shown in Section D in the appendix), and Figure 1(c) used uniform initializations. The constants $C$ for those two initializations are highly different, since 1(a) contains a long plateau (thus large $C$ constants) while 1(c) does not.

---

> > ### Author Response · Authors · 2022-08-02
> > **Response to Reviewer aDcz: part 2**
> >
> > The fact that the constants $C$ depend on initialization also makes them depend on "chance" from stochasticity/randomness due to on-policy sampling $a_t \sim \pi_{\theta_t}(\cdot)$. If sub-optimal actions are sampled and updated for multiple times simply because of bad luck, then the situation after updating becomes equivalent to starting from a bad initialization (where the optimal action has a very small chance to be sampled).
> >
> > **Third**, there are only qualitative/intuitive understandings of these constants. For example, in Section D in the appendix, the noise level/reward range of the sampled reward is $3$, and increasing this range to larger values like $5$ or $10$ will make it more difficult for the same algorithm to converge, as we observed. However, how exactly the $C$ quantity depends on the reward range is still unclear (because of it also depends on initializations as mentioned, and thus it is not obvious to us how to only investigate its dependence on every individual factor like the reward range).
> >
> > **Fourth**, for general MDPs, there is related work on standard softmax PG using true gradients ([10] cited in the paper), showing that $C$ can depend exponentially on the number of states in the worst case, which is very pessimistic. However, the results are not directly applicable here, both because the setting (stochastic) and the method (NPG) are different. Here the long plateau (therefore a large constant $C$) in Figure 1(a) arises simply from the behaviour policy (optimal action just has small chance to be sampled and learned initially) rather than from a small softmax true policy gradient, as in [10].
> >
> > >I think "the variance is unbounded" and therefore "Baselines Do Not Control Update Variance" is a leap I am still not convinced of after reading the main body of the paper. I was wondering if the authors could comment on how the two values of interest in Proposition 3 ($\frac{r(a)^2}{\pi_{\theta_t}(a)}$ and $\frac{ ( r(a) - \pi_{\theta_t}^\top r )^2 }{ \pi_{\theta_t}(a) }$) evolve through time $t$ (and not only the convergence)? Please point me to the result if it already dealt with in the paper or previous work. This might give us an idea of how baselines effect the variance through time.
> >
> > In this on-policy NPG setting, we have both $\frac{r(a)^2}{\pi_{\theta_t}(a)} \to \infty$ (without baseline) and $\frac{ ( r(a) - \pi_{\theta_t}^\top r )^2 }{ \pi_{\theta_t}(a) } \to \infty $ (with baseline) as $t \to \infty$. However, the big difference is $\frac{r(a)^2}{\pi_{\theta_t}(a)} \to \infty$ (without baseline) is achieved by approaching any deterministic policy (including sub-optimal and optimal deterministic policies) depending on initializations and stochasticity from sampling, while $\frac{ ( r(a) - \pi_{\theta_t}^\top r )^2 }{ \pi_{\theta_t}(a) } \to \infty $ (with baseline) is achieved by approaching the global optimal deterministic policy with probability 1 (Lemma 2).
> >
> > Consider $r = (1.0, 0.5, 0.1)^\top$, for any near deterministic policies like $\pi = (\epsilon, 1 - 2 \epsilon, \epsilon)$ (sub-optimal) or $\pi = (1 - 2 \epsilon, \epsilon, \epsilon )$ (optimal), it is easy to calculate $\frac{r(3)^2}{\pi(3)} = \frac{0.1^2}{\epsilon} $ and $\frac{ ( r(3) - \pi^\top r )^2 }{ \pi(3) }  = \frac{ ( 0.1 - \pi^\top r )^2 }{ \epsilon }$ (where $\pi^\top r \approx 1.0$ or $\pi^\top r \approx 0.5$ depending on which policy of the above two is used for calculation). The two quantities both blow up as $\epsilon \to 0$, which happens for any deterministic policy (including sub-optimal and optimal deterministic policies). Therefore, the two quantities both $\to \infty$ as $t \to \infty$. This confirms again our statements in the paper that the variance of estimators itself is not sufficient to clarify why and how baselines would help global convergence.
> >
> > >In the abstract you claim that for the NPG setting "finite variance is not necessary" (line 15) and in Section 3.1 you claim "there exists at least one action such that $\frac{r(a)^2}{\pi_{\theta_t}(a)}$ and $\frac{ ( r(a) - \pi_{\theta_t}^\top r )^2 }{ \pi_{\theta_t}(a) }$ become unbounded" (line 244). Can you please explain in what case the variance is finite? My reading is that this necessitates variance being unbounded.
> >
> > The reviewer is right that if there are no additional assumptions, then the two quantities are necessarily unbounded. However, as mentioned above (e.g., Algorithm 1 in Agarwal et al. [1]), assuming the behaviour policy is covering all action spaces will make the ratio bounded (thus assuming away the difficulty here), and using multiple sampling depending on $\pi_t$ ([12] cited in the paper) also makes the variance bounded (but then requires a non-constant number of samples, and is not applicable to the constant sampling size $1$ considered here, or any other constant batch sizes).

---

> > > ### Author Response · Authors · 2022-08-02
> > > **Response to Reviewer aDcz: part 3**
> > >
> > > >I am curious why the optima is stuck in a plateau in Fig 2(a)? Kakade [4] had asserted that NPG overcomes the plateau phenomena, in principle, effectively. I hope the authors can comment on this.
> > >
> > > This is because the behaviour policy is bad. As shown in Section D, we have $\pi_{\theta_1}(a^*) \approx 0.006$, which means the optimal action initially only has less than a $1$ percent chance of being sampled using on-policy sampling $a_t \sim \pi_{\theta_t}(\cdot)$. This is a challenge in the on-policy stochastic setting, and it is not conflicting with the NPG results mentioned by the reviewer, which hold in the true gradient setting as well as in the stochastic gradient setting **with bounded variance** (one can show that the stochastic gradient is then "close enough" to the true gradient).
> > >
> > > In this case, the plateau no longer arises because PG itself is not good (since we are using NPG, and thus is different from [10] cited in the main paper), but because the behaviour policy itself is bad. Using other designs to overcome this challenge in the on-policy setting (such as entropy regularization and other baseline designs mentioned in the conclusion) would be interesting future work.
> > >
> > > >Can you present empirical results for a more general MDP case as well? That would make the argument stronger although I wouldn't reject the paper for this reason since the results for single state example serve as a good example.
> > >
> > > The results on a tree MDP with depth $4$ and branching factor $4$ (total number of states $S = 85$) has been added to Section D in the appendix. As we observed, the results are aligned with the theory and very similar to Figure 1.
> > >
> > > >Some additional background on NPG in section 2 and how it differs from vanilla PG would be helpful for the reader. Update 2 being NPG and where it comes from might not be apparent to most readers in the community.
> > >
> > > Sorry for missing the background on NPG. The cited related papers Kakade [6] and Agarwal et al. [1] will be discussed in more detail, explaining where the updates come from.
> > >
> > > >How is committal behavior different from the standard exploration exploitation problem? I am curious if some of the effects of the baseline are similar to that of adding noise (or policy gradient version of $\epsilon$-greedy exploration) to the policy?
> > >
> > > The committal behavior is different from the standard exploration-exploitation problem even in the simplest setting of a noiseless bandit model (observing true mean reward for sampled actions, but actions still need to be sampled $a_t \sim \pi_t(\cdot)$).
> > >
> > > In this setting, there is no exploration-exploitation problem, since the reward is fixed (not adversarial/changing) and observed without noise (not stochastic bandits, no need to estimate reward).
> > >
> > > However, as shown in Proposition 1 and Lemma 6, committal behaviour in on-policy NPG without a baseline can be problematic (converging toward sub-optimal deterministic policies with positive constant probabilities).  Meanwhile, using a baseline guarantees convergence, since it prevents over-committal behaviours / reduces aggressiveness of the updates (Lemma 8 and Eq. (16)).
> > >
> > > Regarding the last question of relating the baseline to adding noise to the policy or $\epsilon$-greedy exploration, our understanding is the reviewer is asking in general whether adding a baseline is similar/related to other existing exploration methods. This is also curious to us (as mentioned in the conclusion section), and our high-level intuitive understanding is that a baseline does maintain some level of "automatic exploration" (by maintaining every action's probability decaying no faster than $O(1/t)$ thus every action has a sufficient chance to be sampled in the long run).  However, we do not have a clearer answer since our understanding of the effect of the baseline is still fresh.  This question requires further investigation.

---

> > > > ### Comment · Reviewer_aDcz · 2022-08-09
> > > > **Thank you for the detailed response and changing my rating to a 6**
> > > >
> > > > My apologies for the late response. Thank you very much for the detailed response. I am changing my rating to a 6.
> > > >
> > > > I would only recommend a couple of things for the revisions of the paper:
> > > > 1. I believe some more background information would really be helpful. I do realise you cite Agarwal et al, Kakade, and Bhatnagar et al it would be more informative if you gave background in the paper.
> > > >
> > > > 2. I would point the reader to relevant section in the appendix in the main body (especially Section D). I was unable to find a reference but I think this would help with presentation.

---

> > > > > ### Author Response · Authors · 2022-08-09
> > > > > **Thank you for the suggestions**
> > > > >
> > > > > Dear Reviewer aDcz,
> > > > >
> > > > > Thank you for reconsidering the rating!
> > > > >
> > > > > As you suggested, we will give more background for NPG and point relevant appendix sections in subsequent versions.

---

> ### Author Response · Authors · 2022-08-09
> **Thank you for checking response**
>
> Dear Reviewer aDcz,
>
> Thank you for checking our previous response! The following is one more thing we could say to hopefully better clarify one of your major concerns from review in retrospect.
>
> >My second concern is the lack of explanation of the constants in the convergence rate results. The constants $C$ in Theorems 1 and 2 are not explained. Are they the same or similar? I see that the constant does not depend on $t$ but as a reader I would like to know what this constant does depend on? What properties of the MDP effect these constants? Therefore, in my opinion, a corresponding empirical study of how this convergence rate changes with the MDP would also be warranted for a complete understanding of the theoretical results in addition to the explanations.
>
> To our understanding, an ideal result as the reviewer suggested for the constants $C$ would look like (just for explaining things): $C = \frac{ R_{\max}  }{ \Delta } \cdot \text{dist}( \pi_{\theta_1}, \pi^* ) $ (in Theorem 1), and the larger the sample reward range $R_{\max}$ and the smaller the reward gap $\Delta$ are, the more difficult the problem is. On the other hand, the larger the distance between $\pi_{\theta_1}$ and $\pi^*$ is, the worse the initialization is, and thus the longer time the same algorithm would take to converge.
>
> The above potential result of $C$ is definitely desirable once achieved. However, this result would require the update to have "constant amount of progress comparing to current sub-optimality" (this will be explained shortly), which is unfortunately not true for softmax on-policy stochastic NPG we studied.
>
> Consider $(\pi^* - \pi_{\theta_t})^\top r \le \frac{C}{t}$, which required us to establish the following relationship between progress and sub-optimality gap, $\pi_{\theta_{t+1}}^\top r - \pi_{\theta_{t}}^\top r \ge \frac{1}{C} \cdot \left[ (\pi^* - \pi_{\theta_t})^\top r \right]^2 $. If $C = \frac{ R_{\max}  }{ \Delta } \cdot \text{dist}( \pi_{\theta_1}, \pi^* ) $ as mentioned before, then this essentially means comparing to $\left[ (\pi^* - \pi_{\theta_t})^\top r \right]^2 $, the progress $\pi_{\theta_{t+1}}^\top r - \pi_{\theta_{t}}^\top r $ is always a "constant amount" of $\frac{1}{C} = \frac{ \Delta }{ R_{\max}  } \cdot \frac{1}{  \text{dist}( \pi_{\theta_1}, \pi^* ) }$.
>
> Starting from an initial policy $\pi_{\theta_1}$, the (expected) progress of on-policy NPG is proportional to $\frac{\Delta}{R_{\max}^2} \cdot \pi_{\theta_t}(a^*) \cdot \left[ (\pi^* - \pi_{\theta_t})^\top r \right]^2$ (Lemma 1), and a simple example of 2-armed bandit can show that the inequality is tight in the worst case (progress $\le \pi_{\theta_t}(a^*)$ is also true for at least one initialization).
>
> Now if we have $\pi_{\theta_{t+1}}(a^*) \ge \pi_{\theta_{t}}(a^*)$ for on-policy stochastic NPG, then we will have $\pi_{\theta_{t+1}}^\top r - \pi_{\theta_{t}}^\top r \ge \frac{\Delta}{R_{\max}^2} \cdot \pi_{\theta_{1}}(a^*) \cdot \left[ (\pi^* - \pi_{\theta_t})^\top r \right]^2$, which means $\frac{1}{C} = \frac{\Delta}{R_{\max}^2} \cdot \pi_{\theta_{1}}(a^*)$, and thus the final result would look like a convergence rate of $\frac{C}{t}$ with $C = \frac{R_{\max}^2}{\Delta} \cdot \frac{1}{ \pi_{\theta_{1}}(a^*) }$.
>
> However, this is not the reality of on-policy stochastic NPG method, because of "$\pi_{\theta_{t+1}}(a^*) \ge \pi_{\theta_{t}}(a^*)$" is not true. As verified in Section D (Figure 3(c)), the quantity $\pi_{\theta_{t}}(a^*)$ can decrease to $10^{-12}$ and it is clearly not monotonically increasing (its initial value $\pi_{\theta_1}(a^*) = 0.07$). And for this reason, to our understanding as explained here, getting a simple dependence like $C \propto \frac{1}{ \pi_{\theta_{1}}(a^*) }$ seems unlikely to happen for this method.

---

### Official Review · Reviewer_6WdT · 2022-07-12

**Rating:** 5
**Confidence:** 3
**Soundness:** 3 good
**Presentation:** 3 good
**Contribution:** 3 good

**Summary:**

This paper discusses the effect of baselines in stochastic policy gradient optimization, i.e., the subtracted term in the advantage function. Previous approaches have demonstrated that by subtracting a state value baseline would stabilize the learning of RL empirically, however, few researches focus on explaining this theoretically. Considering natural policy gradient (NPS), the paper demonstrated that using a state value baseline can prevent the probability of the optimal action from vanishing almost for sure, and the NPG can converge to a globally optimal policy at rate of O(1/t). The paper further discusses more on the variance reduction effect of using baselines for stochastic gradients. Results show that the PG estimator is unbounded neither with nor without a baseline, and the key effect of baseline is ensuring convergence and reduce the aggressiveness of updates. Some simulation results in simple one-state MDP toy example are provided to support the proposed theories.

**Questions:**

Are there any connections between the theories in this paper and existing variance reducation baselines, e.g., the input-dependent baseline [1]? Could the used techniques to applied to explain such baselines in the effect of variance reduction and global convergence?

[1] Mao H, Venkatakrishnan S B, Schwarzkopf M, et al. Variance reduction for reinforcement learning in input-driven environments[J]. arXiv preprint arXiv:1807.02264, 2018.


**Limitations:**

Insufficient empirical studies of stochastic NPG with baseline.

**Strengths And Weaknesses:**

Pros:
The proposed discussion and analyses indeed provide me new insights for the effect of baselines in stochastic PG. I think the used techniques in analyzing the global optimality convergence for stochastic PG is informative for future research. Moreover, the relationship between using value baselines and variance reduction is discussed and the results tell that there might not exist directly.


Cons:
The experimental results are far insufficient and general MDPs are not studied for Algorithm 1, which, I think, is the most critical part for RL applications. The only concern I have is the practical performance that whether on-policy stochastic NPG matches the theoretical convergence results.

---

> ### Author Response · Authors · 2022-08-02
> **Response to Reviewer 6WdT**
>
> We would like to thank the reviewer for appreciating our work and we address the main concerns as follows.
>
> >Cons: The experimental results are far insufficient and general MDPs are not studied for Algorithm 1, which, I think, is the most critical part for RL applications. The only concern I have is the practical performance that whether on-policy stochastic NPG matches the theoretical convergence results.
>
> The results on a tree MDP with depth $4$ and branch factor $4$ (total number of states $S = \sum_{i=0}^{3}{4^i} = 85$) is added into Section D in the appendix. As we observed, the results are aligned with the theory and similar to Figure 1.
>
> >Are there any connections between the theories in this paper and existing variance reducation baselines, e.g., the input-dependent baseline [1]? Could the used techniques to applied to explain such baselines in the effect of variance reduction and global convergence?
>
> We thank the reviewer for pointing Mao el al. [1] to us. To our knowledge, existing results on stochastic PG are using bounded variance assumptions (or calculate/show that the variance of estimator is bounded), including papers mentioned by another reviewer aDcz as well as [15, 22, 9, 21] already cited in the main paper. The standard results have the form of "sub-optimality gap $\le O(1/\sqrt{t}) + $ variance" (e.g., Thm. 6.2 in Agarwal et al. [1] cited in the paper), which requires the variance term to be uniformly bounded by constants across the whole probability simplex. Otherwise, the upper bound will be vacuous (sub-optimality gap $\le \infty$).
>
> Given bounded variance, the term "variance reduction" is then meaningful. However, in the on-policy NPG setting studied in this work, as calculated in Proposition 3, the variance is unbounded and so our results are not covered by these studies. As also discussed in the response to the first question from Reviewer aDcz, the variance terms $\frac{r(a)^2}{\pi_{\theta_t}(a)}$ (without baseline) and $\frac{ ( r(a) - \pi_{\theta_t}^\top r )^2 }{ \pi_{\theta_t}(a) }$ (with baseline) both approach $\infty$ as $t \to \infty$. Therefore, from this observation, the magnitude of the variance of estimators seems to be insufficient to determine convergence behaviour in this setting. It remains unclear to us how our result can be directly applied to variance reduction settings when additional assumptions of bounded variance are imposed on the estimators.

---

> > ### Comment · Reviewer_6WdT · 2022-08-08
> > **Response to Authors**
> >
> > Thank you for your detailed responses. The empirical results on a tree MDP provide better support for the theories, although I would expect a more complex or real-world task instead of simulations. The explanations on the connection to variance reduction methods are insightful and I think this would be discussed more in the revision, although it is still unclear that how to apply the theory to variance reduction currently. I would keep my rating as 5.

---

### Meta-Review · Area_Chair_wwgW · 2022-08-27

**Recommendation:** Accept
**Confidence:** Less certain

**Metareview:**

The paper studies the effect of subtracting baselines in natural policy gradient methods for reinforcement learning. The authors offer a new perspective on update aggressiveness that is both intuitive and technically sound. All the reviewers appreciated the exposition around "vicious circle" and "virtuous circle" of updates, and how this is different from variance reduction. There were concerns that the empirical evaluations are limited, but the authors pointed to experiments in depth-4 tree MDPs to provide additional evidence.

There were several clarifying questions from the reviews that the authors addressed comprehensively during the feedback phase. Please include these clarifications in a paper revision to strengthen the current exposition.

**Award:**

No

---

### Decision · Program_Chairs · 2022-09-14

Accept